# Robust Imitation via Mirror Descent Inverse Reinforcement Learning

**Dong-Sig Han, Hyunseo Kim, Hyundo Lee, Je-Hwan Ryu, Byoung-Tak Zhang**
Artificial Intelligence Institute, Seoul National University
{dshan, hskim, hdlee, jhryu, btzhang}@bi.snu.ac.kr

## Abstract

Recently, adversarial imitation learning has shown a scalable reward acquisition method for inverse reinforcement learning (IRL) problems. However, estimated reward signals often become uncertain and fail to train a reliable statistical model since the existing methods tend to solve hard optimization problems directly. Inspired by a first-order optimization method called mirror descent, this paper proposes to predict a sequence of reward functions, which are iterative solutions for a constrained convex problem. IRL solutions derived by mirror descent are tolerant to the uncertainty incurred by target density estimation since the amount of reward learning is regulated with respect to local geometric constraints. We prove that the proposed mirror descent update rule ensures robust minimization of a Bregman divergence in terms of a rigorous regret bound of $\mathcal{O}(1/T)$ for step sizes $\{\eta_t\}_{t=1}^{T}$. Our IRL method was applied on top of an adversarial framework, and it outperformed existing adversarial methods in an extensive suite of benchmarks.

## 1 Introduction

One crucial requirement of practical imitation learning methods is *robustness*, often described as learning expert behavior for a finite number of demonstrations, overcoming various realistic challenges [1]. In real-world problems such as motor control tasks, the demonstration size can be insufficient to create a precise model of an expert [2], and even in some cases, demonstrations can be noisy or suboptimal to solve the problem [3]. For such challenging scenarios, imitation learning algorithms inevitably struggle with unreliable statistical models; thus, the way of handling the uncertainty of estimated cost functions dramatically affects imitation performance. Therefore, a thorough analysis of addressing these issues is required to construct a robust algorithm.

Inverse reinforcement learning (IRL) is an algorithm for learning ground-truth rewards from expert demonstrations where the expert acts optimally with respect to an unknown reward function [4, 5]. Traditional IRL studies solve the imitation problem based on iterative algorithms [6, 7], alternating between the reward estimation process and a reinforcement learning (RL) [8] algorithm. In contrast, newer studies of adversarial imitation learning (AIL) [9, 10] rather suggest learning reward functions of a certain form "directly," by using adversarial learning objectives [11] and nonlinear discriminative neural networks [10]. Compared to classical approaches, the AIL methods have shown great success on control benchmarks in terms of scalability for challenging control tasks [12].

Technically, it is well known that AIL formulates a divergence minimization problem with its discriminative signals, which incorporates fine-tuned estimations of the target densities [13]. Through the lens of differential geometries, the limitation of AIL naturally comes from the implication that minimizing the divergence does not guarantee unbiased progression due to constraints of the underlying space [14]. In order to ensure further stability, we argue that an IRL algorithm's progress needs to be regulated, yielding gradual updates with respect to local geometries of policy distributions.

36th Conference on Neural Information Processing Systems (NeurIPS 2022).

We claim that there are two issues leading to unconstrained policy updates: ① a statistical divergence often cannot be accurately obtained for challenging problems, and ② an immediate divergence between agent and expert densities does not guarantee unbiased learning directions. Our approach is connected to a collection of optimization processes called mirror descent (MD) [15]. For a sequence of parameters $\{w_t\}_{t=1}^{T}$ and a convex function $\Omega$, an MD update for a cost function $F_t$ is derived as

$$\nabla\Omega(w_{t+1}) = \nabla\Omega(w_t) - \eta_t \nabla F_t(w_t). \tag{1}$$

In the equation, the gradient $\nabla\Omega(\cdot)$ creates a transformation that links a parametric space to its dual space. Theoretically, MD is a first-order method for solving constrained problems, which enjoys rigorous regret bounds for various geometries [16, 17] including probability spaces. Thus, applying MD to the reward estimation process can be efficient in terms of the number of learning phases.

In this paper, we derive an MD update rule in IRL upon a postulate of nonstationary estimations of the expert density, resulting in convergent reward acquisition even for challenging problems. Compared to MD algorithms in optimization studies, our methodology draws a sequence of functions on an alternative space induced by a reward operator $\Psi_\Omega$ (Definition 1). To this end, we propose an AIL algorithm called mirror descent adversarial inverse reinforcement learning (MD-AIRL). Our empirical evidence showed that MD-AIRL outperforms the regularized adversarial IRL (RAIRL) [18] methods. For example, MD-AIRL showed higher performance in 30 distinct cases among 32 different configurations in challenging MuJoCo [19] benchmarks, and it also clearly showed higher tolerance to suboptimal data. All of these results are strongly aligned with our theoretical analyses.

Table 1: A technical overview. Traditional IRL methods lack scalability, and RAIRL does not guarantee convergence of its solution for realistic cases. MD-AIRL combines desirable properties.

| Method | Reference | Scalability | Rewards | Bregman divergence | Iterative solutions | Convergence analyses |
|---|---|---|---|---|---|---|
| BC (1991) | [20] | ✓ | ✗ | ✗ | ✗ | ✓ |
| MM-IRL (2004) | [6] | ✗ | ✓ | ✗ | ✓ | ✓ |
| GAIL (2016) | [9] | ✓ | ✓ | ✗ | ✗ | ✗ |
| RAIRL (2021) | [18] | ✓ | ✓ | ✓ | ✗ | ✗ |
| **MD-AIRL (ours)** | – | ✓ | ✓ | ✓ | ✓ | ✓ |

**Our contributions.** Our work is complementary to previous IRL studies; the theoretical and technical contributions are built upon a novel perspective of considering iterative RL and IRL algorithms as a combined optimization process with dual aspects. Comparing MD-AIRL and RAIRL, both are highly generalized algorithms in terms of a variety of choices of divergence functions. Tab. 1 shows that MD-AIRL brings beneficial results in realistic situations of limited time and data, since our approach is more aligned with earlier theoretical IRL studies providing formalized reward learning schemes and convergence guarantees. In summary, we list our main contributions below:

- Instead of a monolithic estimation process of a global solution in AIL, we derive a sequence of reward functions that provides iterative local objectives (Section 4).
- We formally prove that rewards derived by an MD update rule guarantee the robust performance of divergence minimization along with a rigorous regret bound (Section 5).
- We propose a novel adversarial algorithm that is motivated by mirror descent, which is tolerant of unreliable discriminative signals of the AIL framework (Sections 6 and 7).

## 2   Related Works

**Mirror descent.**   We are interested in a family of statistical divergences called the Bregman divergence [21]. The divergence generalizes constrained optimization problems such as least squares [22, 23], and it also has been applied in various subfields of machine learning [24, 25]. In differential geometries, the Bregman divergence is a first-order approximation for a metric tensor and satisfies metric-like properties [14, 26]. MD is also closely related to optimization methods regarding non-Euclidean geometries with a discretization of steps such as natural gradients [27, 28]. In the primal space, training with the infinitesimal limit of MD steps corresponds to a Riemannian gradient flow [29, 30]. In the RL domain, MD has been recently studied for policy optimization [31–33]. In this paper, we focus on learning with suboptimal representations of policy, and our distinct goal is to draw a robust reward learning scheme based on MD for the IRL problem.

**Imitation learning.** As a statistical model for the information geometry [34], energy-based policies (i.e., Boltzmann distributions) appeared in early IRL studies, such as Bayesian IRL, natural gradient

IRL, and maximum likelihood IRL [35–37] for modeling expert distribution to parameterized functions. Notably, MaxEnt IRL [7, 38] is one of the representative classical IRL algorithms based on an information-theoretic perspective toward IRL solutions. Also, discriminators of AIL are trained by logistic regression; thus, the logit score of the discriminator defines an energy function that approximates the truth data density for the expert distribution [39]. Other statistical entropies have also been applied to AIL, such as the Tsallis entropy [40]. On the one hand, our approach is closely related to RAIRL [18], which defined its AIL objective using the Bregman divergence. On the other hand, this work further employs the Bregman divergence to derive iterative MD updates for reward functions, resulting in theoretically pleasing properties while retaining the scalability of AIL.

**Learning theory.** There have been considerable achievements in dealing with temporal costs $\{F_t\}_{t=1}^{\infty}$, often referred to as *online learning* [41]. The most ordinary approach is stochastic gradient descent (SGD): $w_{t+1} = w_t - \eta_t \nabla F_t(w_t)$. In particular, SGD is a desirable algorithm when the parameter $w_t$ resides in the Euclidean space since it ensures unbiased minimization of the expected cost. Apparently, policies appear in geometries of probabilities; thus, an incurred gradient may not be the direction of the steepest descent due to geometric constraints [27, 34]. An online form of MD in Eq. (1) is analogous to SGD for non-Euclidean spaces, where each local metric is specified by a Bregman divergence [30]. Our theoretical findings and proofs follow the results of online mirror descent (OMD) that appeared in previous literature for general aspects [15, 16, 42, 28, 17, 30]. Our analyses extend existing theoretical results to IRL; at the same time, they are also highly general to cover various online imitation learning problems which require making decisions sequentially.

## 3 Background

For sets $X$ and $Y$, let $Y^X$ be a set of functions from $X$ to $Y$ and $\Delta_X$ ($\Delta_X^Y$) be a set of (conditional) probabilities over $X$ (conditioned on $Y$). We consider an MDP defined as a tuple $(\mathcal{S}, \mathcal{A}, P, r, \gamma)$ with the state space $\mathcal{S}$, the action space $\mathcal{A}$, the Markovian transition kernel $P \in \Delta_{\mathcal{S}}^{\mathcal{S} \times \mathcal{A}}$, the reward function $r \in \mathbb{R}^{S \times A}$ and the discount factor $\gamma \in [0, 1)$. Let a function $\Omega : \Delta_{\mathcal{A}} \to \mathbb{R}$ be strongly convex. Using $\Omega$, the Bregman divergence is defined as

$$D_{\Omega}(\pi^s \| \hat{\pi}^s) := \Omega(\pi^s) - \Omega(\hat{\pi}^s) - \langle \nabla\Omega(\hat{\pi}^s), \ \pi^s - \hat{\pi}^s \rangle_{\mathcal{A}},$$

where $\pi^s$ and $\hat{\pi}^s$ denote arbitrary policies for a given state $s$. For a representative divergence, one can consider the popular Kullback-Leibler (KL) divergence. The KL divergence is a Bregman divergence when $\Omega$ is specified as the negative Shannon entropy: $\Omega(\pi^s) = \sum_a \pi^s(a) \ln \pi^s(a)$.

Regularization of the policy distribution with respect to convex $\Omega$ brings distinct properties to the learning agent [43, 44]. The objective of regularized RL is to find $\pi \in \Pi$ that maximizes the expected value of discounted cumulative returns along with a causal convex regularizer $\Omega$, i.e.,

$$J_{\Omega}(\pi, r) := \mathbb{E}_{\pi}\Big[\sum\nolimits_{i=0}^{\infty} \gamma^i \big\{ r(s_i, a_i) - \Omega\big(\pi(\cdot|s_i)\big) \big\} \Big], \tag{2}$$

where the subscript $\pi$ on the expectation indicates that each action is sampled by $\pi(\cdot|s_i)$ for the given MDP. In this setup, a regularized RL algorithm finds a unique solution in a subset of the conditional probability space denoted as $\Pi := [\Pi^s]_{s \in \mathcal{S}} \subset \Delta_{\mathcal{A}}^{\mathcal{S}}$ constrained by the parameterization of a policy.

The objective of IRL is to find a function $r_E$ that rationalizes the behavior of an expert policy $\pi_E$. For an inner product $\langle \cdot, \cdot \rangle_{\mathcal{A}}$, consider $\Omega^*$, the Legendre-Fenchel transform (convex conjugate) of $\Omega$:

$$\forall q^s \in \mathbb{R}^{\mathcal{A}}, \quad \Omega^*\big(q^s\big) = \max_{\pi^s \in \Delta_{\mathcal{A}}} \langle \pi^s, \ q^s \rangle_{\mathcal{A}} - \Omega\big(\pi^s\big), \tag{3}$$

where $q^s$ and $\pi^s$ denote the shorthand notation of $q(s, \cdot)$ and $\pi(\cdot|s)$. Differentiating both sides with respect to $q^s$, the gradient of conjugate $\nabla\Omega^*$ maps $q^s$ to a policy distribution. One fundamental property in *regularized* IRL [43] is that $\pi_E$ is the maximizing argument of $\Omega^*$ for $q_E$, where $q_E$ is the regularized state-action value function $q_E(s, a) = \mathbb{E}_{\pi_E}[\sum_{i=0}^{\infty} \gamma^i \{ r_E(s, a) - \Omega(\pi_E^s) \} | s_0 = s, a_0 = a]$. Note that the problem is ill-posed, and every $\hat{r}_E$ that makes its value function $\hat{q}_E$ satisfy $\pi_E^s = \nabla\Omega^*(\hat{q}_E^s) \ \forall s \in \mathcal{S}$ is a valid solution. Addressing this issue, Jeon et al. [18] proposed a reward operator $\Psi_{\Omega} : \Delta_{\mathcal{A}}^{\mathcal{S}} \to \mathbb{R}^{\mathcal{S} \times \mathcal{A}}$, providing a unique IRL solution by $\Psi_{\Omega}(\pi_E)$.

**Definition 1** (Regularized reward operators)**.** Define the regularized reward operator $\Psi_{\Omega}$ as $\psi_{\pi}(s, a) := \Omega'(s, a; \pi) - \langle \pi^s, \nabla\Omega(\pi^s) \rangle_{\mathcal{A}} + \Omega(\pi^s)$, for $\Omega'(s, \cdot; \pi) := \nabla\Omega(\pi^s) = \big[ \nabla_p \Omega(p) \big]_{p=\pi(\cdot|s)}$.

The reward function $\psi_E := \Psi_{\Omega}(\pi_E)$ replaces its state-action value function, since the sum of composite Bregman divergences derived from Eq. (2) allows reward learning in a greedy manner [18].

## 4 RL-IRL as a Proximal Method

**Associated reward functions.** We consider the RL-IRL processes as a sequential algorithm with local constraints and define sequences $\{\pi_t\}_{t=1}^{\infty}$ and $\{\psi_t\}_{t=1}^{\infty}$ that denote policies and associated reward functions, respectively. The associated reward functions are in a space $\Psi_\Omega(\Pi)$, which is an alternative space of the dual space, defined by the regularized reward operator $\Psi_\Omega$. Formally, we provide Lemma 1, which shows a bijective relation between the operators $\nabla\Omega^*$ and $\Psi_\Omega$ in the set $\Pi$. The proof is in Appendix A.

**Lemma 1** (Natural isomorphism). *Let $\psi \in \Psi_\Omega(\Pi)$ for $\Psi_\Omega(X) \coloneqq \{\psi \mid \psi(s,a) = \psi_\pi(s,a), \ \forall s \in \mathcal{S}, a \in \mathcal{A}, \pi \in X\}$. Then, $\nabla\Omega^*(\psi)$ is unique and for every $\pi = \nabla\Omega^*(\psi)$, $\pi \in \Pi$.*

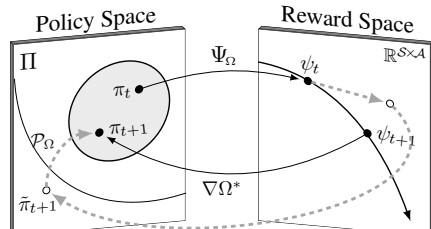

Figure 1: A schematic illustration. MD is locally constrained by a divergence (gray area), i.e., $D_\Omega(\cdot \| \pi_t)$. An MD update is performed for the reward function $\psi_t$ in an associated reward space of defined by $\Psi_\Omega$, and $\pi_{t+1}$ is achieved in the desired space of $\Pi$ by applying $\nabla\Omega^*$ for the function $\psi_{t+1}$. The gray dashed lines provide another interpretation of MD with $\tilde{\pi}_{t+1}$ and the projection operator $\mathcal{P}_\Omega$.

Fig. 1 illustrates that there is a unique $\psi_t$ for $\pi_t$ in every time step. Note that $\Psi_\Omega(\pi_t)$ is different from $\nabla\Omega(\pi_t)$; it is shifted by a vector $\mathbf{1}c$ with a constant $c = \Omega(\pi_t^s) - \langle \pi_t^s, \nabla\Omega(\pi_t^s)\rangle_\mathcal{A}$. Since the underlying space is a probability simplex, the operator $\nabla\Omega^*$ reconstructs the original point for both $\Psi_\Omega$ and $\nabla\Omega$, as the distributivity [43] $\Omega^*(y + \mathbf{1}c) = \Omega^*(y) + c$ holds (so $\nabla\Omega^*(y + \mathbf{1}c) = \nabla\Omega^*(y)$). An alternative interpretation is of considering a projection (gray dashed line in Fig. 1). Suppose that a policy $\pi_t$ is updated to $\tilde{\pi}_{t+1} \in \mathbb{R}^{\mathcal{S}\times\mathcal{A}}$. The Bregman projection operator $\mathcal{P}_\Omega$ is applied that locates the subsequent update $\pi_{t+1}$ to the "feasible" region, i.e., $\mathcal{P}_\Omega(\tilde{\pi}_{t+1}) \coloneqq \mathrm{argmin}_{\pi\in\Pi}[D_\Omega(\pi^s \| \tilde{\pi}_{t+1}^s)]_{s\in\mathcal{S}}$.

Consequently, one can consider an updated reward function $\psi_{t+1}$ as a projected target of MD associated with an alternative parameterization of $\Pi$. For instance, the parameters of $\psi_t$ can construct a softmax policy for a discrete space, or a Gaussian policy for a continuous space. Using the reward function $\psi_{t+1}$, an arbitrary regularized RL process maximizing Eq. (2) at the $t$-th step [18]

$$J_\Omega(\pi, \psi_{t+1}) = -\mathbb{E}_\pi\Big[\sum_{i=0}^{\infty} \gamma^i D_\Omega\Big(\pi(\cdot|s_i)\Big\|\pi_{t+1}(\cdot|s_i)\Big)\Big] \tag{4}$$

becomes finding the next iteration $\pi_{t+1} = \nabla\Omega^*(\psi_{t+1})$ by maximizing the expected cumulative return. The equation shows that a regularized RL algorithm with the regularizer $\Omega$ forms a cumulative sum of Bregman divergences; thus, the policy $\pi_{t+1}$ is uniquely achieved by the property of divergence.

**Online imitation learning.** Our setup starts from the apparent yet vital premise that an imitation learning algorithm does not retain the global target $\pi_E$ during training. That is, it is fundamentally uncertain to model global objectives (such as $J_\Omega(\pi, \psi_E)$), which are not attainable for both RL and IRL. Instead, we hypothesize on the existence of a random process $\{\bar{\pi}_{E,t}\}_{t=1}^{\infty}$ where each estimation $\bar{\pi}_{E,t}$ resides in a closed, convex neighborhood of $\pi_E$, generated by an arbitrary estimation algorithm. Substituting $\psi_E$ to $\psi_{\bar{\pi}_{E,t}} = \Psi_\Omega(\bar{\pi}_{E,t})$, the nonstationary objective $J_\Omega(\pi, \psi_{\bar{\pi}_{E,t}})$ forms a temporal cost:

$$F_t(\pi) = \mathbb{E}_\pi\Big[\sum_{i=0}^{\infty} \gamma^i D_\Omega\big(\pi(\cdot|s_i)\big\|\bar{\pi}_{E,t}(\cdot|s_i)\big)\Big]. \tag{5}$$

For the sake of better understanding, we considered an actual experiment depicted in Fig. 2. Suppose that the policies of the learning agent and the expert follow multivariate Gaussian distributions at

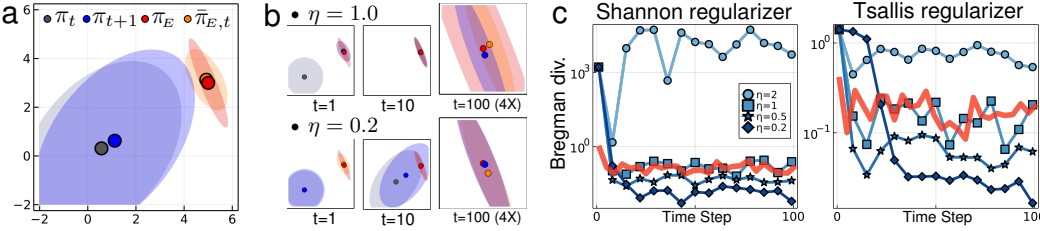

Figure 2: (a) A policy $\pi_t$ learns from MD updates for temporal costs $D_\Omega(\cdot\|\bar{\pi}_{E,t})$. (b) The updates of $\pi_t$ vary by $\eta$, and the distance between $\pi_t$ and $\pi_E$ can be closer than the distance between $\bar{\pi}_{E,t}$ and $\pi_E$ when $t$ is sufficiently large and the $\eta$ is effectively low. (c) Two plots show $D_\Omega(\pi_t, \|\pi_E)$ associated with entropic regularizers for four different $\eta$ (10 trials), with the red baselines $D_\Omega(\bar{\pi}_{E,t}\|\pi_E)$.

$\mathcal{N}([0, 0]^\intercal, \mathrm{I})$ and $\mathcal{N}([5, 3]^\intercal, \Sigma_E)$ with $|\Sigma_E| < 1$. Let a (suboptimal) reference policy $\bar{\pi}_{E,t}$ be independently fitted with a maximum likelihood estimator with a relatively high learning rate, starting from $\bar{\pi}_{E,1} = \pi_1$. The policy $\pi_t$ was trained by a cost function $D_\Omega(\cdot \| \bar{\pi}_{E,t})$ using the MD update rule in Eq. (1). In Fig. 2, we first observed that choosing a high step size constant $\eta$ accelerated the training speed mainly in the early phase. The results also showed that the performance of MD ($D_\Omega(\pi_t \| \pi_E)$) outperformed that of referenced maximum likelihood estimation ($D_\Omega(\bar{\pi}_{E,t} \| \pi_E)$) by choosing an effectively low step size. This empirical evidence suggests that there are clear advantages in formalization of the training steps and scheduling the step sizes, especially for unreliable statistical model $\bar{\pi}_{E,t}$.

**MD update rules.** As a result of these findings, we formulate subsequent MD steps with a regularized reward function. Let $w_t$ be a parameter on a set $\mathcal{W}$ and $F_t : \mathcal{W} \to \mathbb{R}$ be a convex cost function from a class of functions $\mathcal{F}$ at the $t$-th step. Replacing the L2 proximity term of proximal gradient descent with a Bregman divergence, the proximal form of the MD update for Eq. (1) is written as [45]

$$\underset{w \in \mathcal{W}}{\text{minimize}} \; \langle \nabla F_t(w_t), \, w - w_t \rangle_\mathcal{W} + \alpha_t D_\Omega(w \| w_t), \tag{6}$$

where $\alpha_t := 1/\eta_t$ denotes an inverse of the current step size $\eta_t$ [46]. Plugging each divergence of the cumulative cost $F_t$ to Eq. (6), the MD-IRL update for the subsequent reward function $\psi_{t+1} = \Psi_\Omega(\pi_{t+1})$ is derived by solving a problem

$$\underset{\pi^s \in \Pi^s}{\text{minimize}} \; \langle \underbrace{\nabla D_\Omega(\pi_t^s \| \bar{\pi}_{E,t}^s)}_{\nabla \Omega(\pi_t^s) - \nabla \Omega(\bar{\pi}_{E,t}^s)}, \, \pi^s - \pi_t^s \rangle_\mathcal{A} + \alpha_t D_\Omega(\pi^s \| \pi_t^s)$$

$$\iff \quad \underset{\pi^s \in \Pi^s}{\text{minimize}} \; D_\Omega(\pi^s \| \bar{\pi}_{E,t}^s) - D_\Omega(\pi^s \| \pi_t^s) + \alpha_t D_\Omega(\pi^s \| \pi_t^s)$$

$$\iff \quad \underset{\pi^s \in \Pi^s}{\text{minimize}} \; \eta_t \underbrace{D_\Omega(\pi^s \| \bar{\pi}_{E,t}^s)}_{\text{estimated expert}} + (1 - \eta_t) \underbrace{D_\Omega(\pi^s \| \pi_t^s)}_{\text{learning agent}} \qquad \forall s \in \mathcal{S}, \tag{7}$$

where the gradient of $D_\Omega$ is taken with respect to its first argument $\pi_t^s$. Note that solving the optimization Eq. (7) requires interaction between $\pi_t$ and the dynamics of the given environment in order to minimize $F_t$; thus, the corresponding RL process plays an essential role in sequential learning by the induced the value measures. At a glance, the objective is analogous to finding an interpolation at each iteration where the point is controlled $\eta_t$. Fig. 3 shows that the uncertainty of $\pi_t$ (blue region) gets minimal regardless of persisting uncertainty of $\bar{\pi}_{E,t}$ (red region).

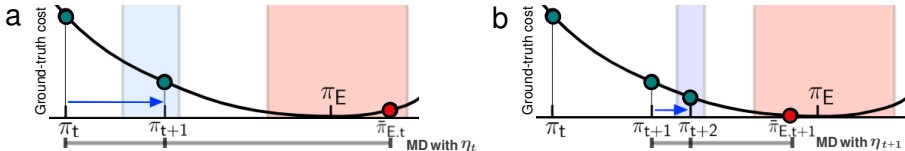

Figure 3: Illustrations of MD at the (a) $t$-th iteration and (b) $(t+1)$-th iteration where $\eta_t > \eta_{t+1}$. $\{\bar{\pi}_{E,t}\}_{t=1}^\infty$ is an arbitrary estimation process attained from a neighborhood of $\pi_E$ with respect to a norm. The MD update is taken inside the interval of $\pi_t$ and $\bar{\pi}_{E,t}$ using Eq. (7).

## 5 Convergence Analyses

In this section, we present our theoretical results. The main goals of the following arguments are to address ① the convergence of MD updates for various cases and ② the necessity of scheduling the amount of learning. Suppose that state instances of $s_i^{(t)} \in \tau_t$ cover the entire $\mathcal{S}$ by executing the policy $\pi_t$ in an infinite horizon. From this assumption, we define a temporal cost function at the time step $t$:

$$f(\pi_t, \tau_t) := \sum_{i=0}^\infty \gamma^i D_\Omega(\pi_t(\cdot \mid s_i^{(t)}) \| \bar{\pi}_{E,t}(\cdot \mid s_i^{(t)})), \tag{8}$$

that involves $\pi_t$, and additionally a trajectory $\tau_t$ as inputs. We refer to the global objective as finding a unique fixed point $\pi_* \in \Pi$ that minimizes a total cost $F(\pi) := \mathbb{E}[f(\pi, \tau_t)]$, where the expectation is taken over trajectories of entire steps, i.e., $\lim_{t \to \infty} \mathbb{E}_{\tau_{1:t}}[f(\pi, \tau_t)]$. Taking the (stepwise) gradient for each $\pi(\cdot | s)$, an optimal policy $\pi_*$ is found by $\mathbb{E}[\nabla \Omega(\pi_*(\cdot | s)) - \nabla \Omega(\bar{\pi}_{E,t}(\cdot | s))] = 0$ when $t \to \infty$; hence, $\nabla \Omega(\pi_*(\cdot | s)) = \lim_{t \to \infty} \mathbb{E}[\nabla \Omega(\bar{\pi}_{E,t}(\cdot | s))]$. Introducing the optimal policy $\pi_*$ allows not only the specific situation when ① $\pi_E = \pi_* \in \Pi$ and the estimation algorithm of $\bar{\pi}_{E,t}$ is actually convergent with $t \to \infty$, but also more general situations where ② $\pi_E \notin \Pi$ or the estimated expert policy $\bar{\pi}_{E,t}$ does not converge; the algorithm finds convergence to a fixed point by scheduling updates.

We state two conditions of $\{\eta_t\}_{t=1}^{\infty}$ to guarantee convergence justified in Theorems 1 and 2.
- Convergent sequence & divergent series:

$$\lim_{t\to\infty} \eta_t = 0 \qquad \text{and} \qquad \sum_{t=1}^{\infty} \eta_t = \infty. \tag{9}$$

- Divergent series & convergent series of squared terms:

$$\sum_{t=1}^{\infty} \eta_t = \infty \qquad \text{and} \qquad \sum_{t=1}^{\infty} \eta_t^2 < \infty. \tag{10}$$

Let us assume Lipschitz continuity of $\nabla\Omega$ and boundedness of $D_\Omega$ in a Banach space. In some $\Omega$, these two assumptions do not necessarily hold for extreme cases in $\Delta_{\mathcal{A}}^{\mathcal{S}}$, e.g., a distribution that $\pi(a|s) = 0$ for some entries. Nevertheless, these outliers can be left out if the parametrization is constrained to satisfy the assumptions. For example, one can either ① prevent a policy from having non-zero entries of probabilities for a discrete policy or ② prevent a policy from having too low entropy for a continuous policy, by enforcing certain constraints on its parametric representations.

Theorem 1 argues that the sequence $\{\eta_t\}_{t=1}^{\infty}$ shall diverge for its series; therefore, Eq. (9) is satisfied.

**Theorem 1** (Stepsize considerations). *Let $\Omega$ be strongly convex, $\nabla\Omega$ be Lipschitz continuous, and the associated Bregman divergence $D_\Omega$ is bounded. Assume a general condition of the problem that $\inf_{\pi\in\Pi}\mathbb{E}[f(\pi,\tau_t)] > 0$. Then we get $\lim_{T\to\infty}\mathbb{E}_{\tau_{1:T}}\left[\sum_{i=0}^{\infty} D_\Omega\left(\pi_*(\cdot|s_i)\big\|\pi_T(\cdot|s_i)\right)\right] = 0$ if and only if Eq. (9) is satisfied.*
*(a) If $\lim_{t\to\infty}\eta_t = 0$, then $T\in\mathbb{N}$, $n<T$, and $c>0$ exist such that $\mathbb{E}_{\tau_{1:T}}\left[f_T(\pi_T,\tau_T)\right] \geq \frac{c}{T-n}$.*
*(b) If the step size is in the form of $\eta_t = \frac{4}{t+1}$, then $\mathbb{E}_{\tau_{1:T}}\left[\sum_{i=0}^{\infty} D_\Omega\left(\pi_*(\cdot|s_i)\big\|\pi_T(\cdot|s_i)\right)\right] = \mathcal{O}(1/T)$.*

Next, we present Theorem 2, which addresses the convergence in a specific case when $\pi_E$ resides in $\Pi$. Additionally, the theorem addresses the bounds of the performance for fixed size update $\eta_t \equiv \eta_1$.

**Theorem 2** (Optimal cases). *Let $\Omega$ be strongly convex, $\nabla\Omega$ be Lipschitz continuous, and the associated Bregman divergences be bounded. Assume $\pi_1 \neq \pi_E$ and $\inf_{\pi\in\Pi}\mathbb{E}[f(\pi,\tau_t)] = 0$. Then, $\mathbb{E}[f(\pi_t,\tau_t)] = 0$ if and only if $\sum_{t=1}^{\infty}\eta_t = \infty$. If $\eta_t \equiv \eta_1$, then there exist $c_1, c_2 \in (0,1)$ such that $c_1^{T-1}\cdot A_1 \leq A_T \leq c_2^{T-1}\cdot A_1$, for $A_t = \sup_{s\in\mathcal{S}}\mathbb{E}_{\tau_{1:t}}\left[D_\Omega(\pi_E^s\|\pi_t^s)\right]$.*

Lastly, Proposition 1 provides the sufficient condition for the almost certain convergence of the algorithm by imposing the stronger condition of step size in Eq. (10). The proofs are in Appendix A.

**Proposition 1** (General cases). *Assume that $\pi_E \notin \Pi$, hence $\inf_{\pi\in\Pi}\mathbb{E}[f(\pi,\tau_t)] > 0$. If the step sizes satisfies Eq. (10), then $\lim_{t\to\infty}\sum_{i=0}^{\infty}\gamma^i D_\Omega\left(\pi_*(\cdot|s_i)\big\|\pi_t(\cdot|s_i)\right)$ converges to 0 almost surely.*

**Regrets.** For a sequence of state trajectories $\{\tau_t\}_{t\in\mathbb{N}}$, let us define a regret at the $t$-th iteration as

$$\frac{1}{t}\sum_{i=1}^{t} f(\pi_i,\tau_i) - \inf_{\pi\in\Pi}\left\{\frac{1}{t}\sum_{j=1}^{t} f(\pi,\tau_j)\right\}. \tag{11}$$

In the optimal case of $\inf_{\pi\in\Pi}\mathbb{E}[f(\pi,\tau_t)] = 0$, the cost $f$ inherits the property of Bregman divergence so that the infimum is achieved by 0 at $\pi_E$. In this case, the regret is bounded to $\mathcal{O}(1/T)$ by the theorems. By Proposition 1, the MD updates converge for the case of $\inf_{\pi\in\Pi}\mathbb{E}[f(\pi,\tau_t)] > 0$ when the step sizes abide by Eq. (10). Thus, the regret is bounded to $\mathcal{O}(1/T)$ even for the general case.

## 6 Algorithm: MD-IRL on an Adversarial Framework

In this section, we propose MD-AIRL, a novel AIL algorithm which trains a parameterized reward function with adversarial learning and the MD update rule. Neural network parameters $\theta$, $\phi$, and $\nu$ are newly presented representing agent policy, reward, and expert policy functions respectively.

**Dual discriminators.** In order to bridge the gap between theory and practice, we propose a novel discriminative architecture, motivated by GAN studies regarding multiple discriminators [47, 48]. Basically, the proposed discriminators separate two concepts in AIL: matching overall state densities and imitating specific behavior. Given a learning agent policy $\pi_\theta$, an estimation policy $\pi_\nu$, and a discriminative neural network for states $d_\xi : \mathcal{S} \to \mathbb{R}$, the two discriminators are defined as

$$D_\nu(s,a;\theta,\xi) = \sigma\left(\log\{\pi_\nu(a|s)/\pi_\theta(a|s)\} + d_\xi(s)\right) \quad \text{and} \quad D_\xi(s) = \sigma\left(d_\xi(s)\right), \quad \forall\, s\in\mathcal{S},\, a\in\mathcal{A},$$

where $\sigma(\cdot)$ denotes the sigmoid function. The discriminators are trained using binary logistic regression losses with respect to mini-batch adversarial samples:

$$\text{maximize } \mathcal{J}_{d_\xi} = \mathbb{E}_{\pi_E}\left[\log D_\xi(s)\right] + \mathbb{E}_{\pi_\theta}\left[\log(1 - D_\xi(s))\right], \tag{12}$$

$$\text{maximize } \mathcal{J}_{\pi_\nu} = \mathbb{E}_{\pi_E}\left[\log D_\nu(s,a)\right] + \mathbb{E}_{\pi_\theta}\left[\log(1 - D_\nu(s,a))\right], \tag{13}$$

**Algorithm 1** Mirror Descent Adversarial Inverse Reinforcement Learning.

1: **Input:** trajectories $\{\tau_t^*\}_{t=1}^T$, an agent $\pi_\theta$, a reference policy $\pi_\nu$, a neural network $d_\xi : \mathcal{S} \to \mathbb{R}$, a regularized reward function $\psi_\phi \in \Psi_\Omega(\Pi)$, $\alpha_1, \alpha_T$, and $\lambda$.
2: **for** $t \leftarrow 1$ to $T$ **do**
3:     $\alpha_t \leftarrow \alpha_1 + (t-1)(\alpha_T - \alpha_1/T - 1)$   and then   $\eta_t \leftarrow 1/\alpha_t$.
4:     Optimize $d_\xi$ and $\pi_\nu$ via binary logistic regression for $D_\xi$ and $D_\nu$.
5:     Optimize $\psi_\phi$ with the objective in Eq. (14) using both $\tau_t^*$ and $\tau_t$.
6:     Train $\pi_\theta$ via RL to maximize $\psi_\phi^\lambda(s,a)$ with regularizer $\lambda\Omega(\cdot)$.
7: **Output:** $\pi_\theta, \psi_\phi^\lambda$.

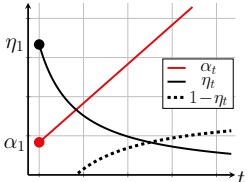

Figure 4: A sequence example of $(\alpha_t, \eta_t)$.

where $d_\xi$ and $\pi_\theta$ are not trained for learning $D_\nu$. Let $\rho_\pi \in \Delta_\mathcal{S}$ denote the state visitation density of $\pi$, which is defined as $\rho_\pi(s) := (1-\gamma)\mathbb{E}_\pi\big[\sum_{i=0}^\infty \gamma^i \mathbb{I}\{s_i = s\}\big]$, where $\mathbb{I}\{\cdot\}$ is an indicator function. The convergence of functions in an ideal case is found at $\pi_\nu = \pi_E$ and $d_\xi(s) = \log\{\rho_{\pi_E}(s)/\rho_{\pi_\theta}(s)\}$.

**Learning with MD-based rewards.** Based on the MD solution for a regularized reward function, we focus on developing an MD-based learning objective. Let $\psi_\phi \in \Psi(\Pi)$ denote a parameterized regularized reward function, and $\pi_\phi$ denotes a corresponding policy from $\phi$. Note that the transformation between $\psi_\phi$ and $\pi_\phi$ can be performed with shared $\phi$ without the additional computational costs under specific parameterizations [18]. Using a step size $\eta_t$, the RL agent $\pi_\theta$, and the estimated expert policy $\pi_\nu$, we define the objective of $\phi$ as a direct interpretation of the update rule of Eq. (7):

$$\text{minimize } \mathcal{L}_{\psi_\phi} = \mathbb{E}_{s \sim \bar{\tau}_t}\big[\eta_t\, D_\Omega\big(\pi_\phi(\cdot|s)\big\|\pi_\nu(\cdot|s)\big) + (1-\eta_t)D_\Omega\big(\pi_\phi(\cdot|s)\big\|\pi_\theta(\cdot|s)\big)\big], \quad (14)$$

where the trajectory $\bar{\tau}_t$ denotes sample states using both agent and expert trajectories. As shown in Fig. 4, $\eta_t$ is adjusted by linearly increasing $\alpha_t$, which originated from the analyses in Section 5.

Another important consideration is the way of handling covariate shifts [49] since it is likely that state densities between the expert and the agent are misaligned. Thus, we define the IRL reward function as linear combinations of $\psi_\phi$ and the state density discriminative signal:

$$\psi_\phi^\lambda(s,a) = \lambda\,\psi_\phi(s,a) + d_\xi(s), \quad (15)$$

with a coefficient $\lambda \in \mathbb{R}^+$. Utilizing an arbitrary regularized RL algorithm with a regularizer $\lambda\Omega(\cdot)$, the reward learning regarding agent policy $\pi_\theta$ is decomposed into the following:

$$\mathbb{E}_{\pi_\theta}\big[\psi_\phi^\lambda(s,a) - \lambda\Omega\big(\pi_\theta(\cdot|s)\big)\big] = \lambda\mathbb{E}_{\pi_\theta}\big[\psi_\phi(s,a) - \Omega\big(\pi_\theta(\cdot|s)\big)\big] - D_{\text{KL}}\big(\rho_{\pi_\theta}\big\|\rho_{\pi_E}\big)$$
$$= -\lambda\mathbb{E}_{\pi_\theta}\big[D_\Omega\big(\pi_\theta(\cdot|s)\big\|\pi_\phi(\cdot|s)\big)\big] - D_{\text{KL}}\big(\rho_{\pi_\theta}\big\|\rho_{\pi_E}\big).$$

Minimizing the first term of $\mathbb{E}_{\pi_\theta}\big[D_\Omega(\pi_\theta^s\|\pi_\phi^s)\big]$ represents learning with the MD formulation. Minimizing the second term $D_{\text{KL}}(\rho_{\pi_\theta}\|\rho_{\pi_E})$ plays an auxiliary role in facilitating the supports of state visitation densities to be correctly matched. With the hyperparameter $\lambda$, we report that learning the second term is helpful when the state densities are heavily misaligned in certain benchmarks. Algorithm 1 summarizes the entire procedure. We defer additional details to Appendices B and C.

## 7 Experimental Results

The aim of our experiments was to identify whether MD-AIRL facilitates robustness for various $\Omega$ while retaining the scalability of AIL. The comparative method was RAIRL with density-based models (RAIRL-DBM) which contained comparable expressiveness as MD-AIRL. For RL, we used RAC [44], which is a generalization of the SAC algorithm [50]. We considered a class of regularizers $\Omega(p) = -\mathbb{E}_{x \sim p}[\varphi(p(x))]$ with ① Shannon $(\varphi(x) = \log(x))$, ② Tsallis $(\varphi(x;q) = \frac{1}{q-1}(x^{q-1} - 1)$, $q = 2$ by default), ③ exp $(\varphi(x) = e - e^x)$, ④ cos $(\varphi(x) = \cos(\frac{\pi}{2}x))$, and ⑤ sin $(\varphi(x) = 1 - \sin(\frac{\pi}{2}x))$.

### 7.1 Large scale multiarmed bandits

To measure the performance of IRL, we first considered multiarmed bandit problems, where the cardinality of action spaces varies largely. Learning the optimal distribution of $\pi_E$ becomes challenging as the cardinality of the space $|\mathcal{A}|$ increases, because the frequency of each sample becomes sparse due to the curse of dimensionality. The stateless expert distribution $\pi_E$ was generated by the parameters of softmax distribution $\pi_E(a) = \exp(z_a)/\sum_i \exp(z_i)$, where the logits $z_i$ were randomly initialized. We set the action size to $|\mathcal{A}| = 10^2, 10^3, 10^4$ and restricted each sample size of 16.

Table 2: The training results of $|\mathcal{A}| \cdot D_\Omega(\pi_T \| \pi_E)$ with five types of regularization (five runs with different seeds).

|  | $|\mathcal{A}| = 10^2$ | | $|\mathcal{A}| = 10^3$ | | $|\mathcal{A}| = 10^4$ | |
| --- | --- | --- | --- | --- | --- | --- |
| Method | RAIRL | MD-AIRL | RAIRL | MD-AIRL | RAIRL | MD-AIRL |
| Shannon | $2.55 \pm 1.59$ | $\mathbf{2.28 \pm 1.20}$ | $140.3 \pm 87.5$ | $\mathbf{125.3 \pm 61}$ | - | - |
| Tsallis | $0.21 \pm 0.13$ | $\mathbf{0.11 \pm 0.04}$ | $0.55 \pm 0.13$ | $\mathbf{0.24 \pm 0.03}$ | $4.95 \pm 2.3$ | $\mathbf{4.21 \pm 0.2}$ |
| exp | $0.27 \pm 0.17$ | $\mathbf{0.13 \pm 0.06}$ | $0.55 \pm 0.12$ | $\mathbf{0.23 \pm 0.03}$ | $5.06 \pm 2.4$ | $4.97 \pm 0.7$ |
| cos | $0.05 \pm 0.04$ | $\mathbf{0.02 \pm 0.01}$ | $0.03 \pm 0.02$ | $\mathbf{0.01 \pm 0.01}$ | $0.21 \pm 0.6$ | $0.05 \pm 0.1$ |
| sin | $0.34 \pm 0.25$ | $\mathbf{0.12 \pm 0.04}$ | $3.82 \pm 3.46$ | $\mathbf{1.07 \pm 0.75}$ | $8.12 \pm 3.8$ | $7.59 \pm 1.0$ |

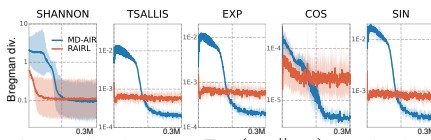

Figure 5: The cost $D_\Omega(\pi_T \| \pi_E)$ on the log-scale at $|\mathcal{A}| = 10^3$. The shade represents $95\%$ confidence interval.

Tab. 2 shows that MD-AIRL achieved overall lower Bregman divergence on average when three different cardinalities and five regularizers were considered. Fig. 5 shows that the Bregman divergence was large for MD-AIRL at the early training phase, because we chose the initial step size $\eta_1$ to be greater than 1 ($\alpha_1 = 0.5$). MD-AIRL exceeded the discriminative performance of RAIRL after certain steps, while the progression of RAIRL mostly stopped at local minima. MD-AIRL outperformed RAIRL in four cases by choosing an effectively low step size at $\eta_T$ to be less than 1 ($\alpha_T = 2$). These results match the properties of MD algorithms and our convergence analyses. Therefore, we argue that a constrained update rule with appropriate step sizes is necessary for robust reward acquisition and imitation for situations when the total number of data samples is limited.

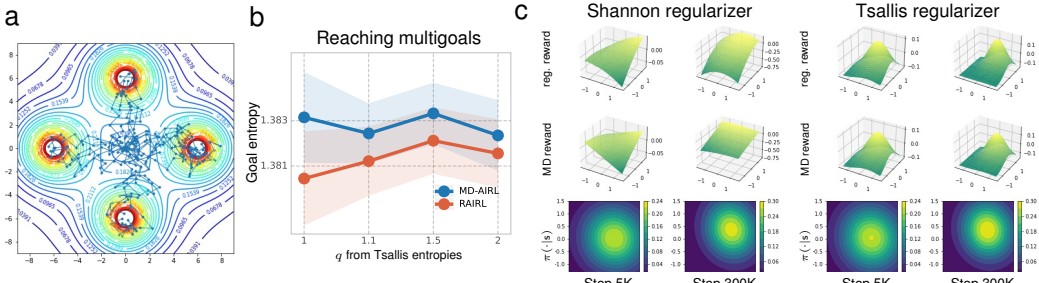

Figure 6: (a) Visualization of trajectories trained by MD-AIRL, and the ground-truth reward surface. (b) The entropies for the probabilities of achieving four goals. The x-axis indicates the $q$ value from the Tsallis regularizers (the Shannon regularizer is considered by $q = 1$ [51]). (c) The top and middle of each column show induced reward surfaces. The bottom shows the agent policy.

## 7.2 A continuous multigoal environment

We then considered a multigoal environment. In this environment, an agent is a two-dimensional point mass initialized at the origin, and the four goals are located in the four cardinal directions. The objective of imitation learning is to go toward each direction evenly as possible where the expert model was trained by the SAC algorithm. To draw informative reward surfaces regarding stochastic actions, we considered the multivariate Gaussian distribution policies parameterized with full covariance matrices instead of conventional diagonal Gaussian policies (see Appendices B and C).

Fig. 6 (a) shows trajectories generated by the trained agent. Fig. 6 (b) shows that MD-AIRL achieved higher entropy for reaching the multiple goals. Fig. 6 (c) shows reward surfaces with regularizers, which were calculated by $\psi_\phi(s, a) + \varphi(\pi_\theta(a|s))$ for each point of $a \in \mathcal{A}$ and $s = (5, -1)$. During the training, the MD reward was similar to the estimated ground truth using adversarial training. However, the surface of MD-AIRL became flatter than the ground-truth estimation when $\pi_t$ was sufficiently close to the expert behavior. As a result, we claim that a drastic change in the target distribution, which is one of the typical characteristics of adversarial frameworks, is prevented. We argue that these characteristics mitigate overfitting caused by unreliable discriminative signals.

## 7.3 A continuous control benchmark: MuJoCo

Lastly, we validated MD-AIRL on the MuJoCo continuous control benchmark suite. We assumed full covariance Gaussian policies for both learner's policy $\pi$ and expert policy $\pi_E$. We used the hyperbolized environment assumption [18] where the action constraint is incorporated into the dynamics as a part of the environment using hyperbolic tangent activation.

**Sample efficiency.** For each task, we considered two different numbers of episodes collected by an expert policy. In Fig. 7, the performance of MD-AIRL, RAIRL, and behavior cloning (bc) algorithms [20] is shown with the expert and random agent performance. MD-AIRL was able

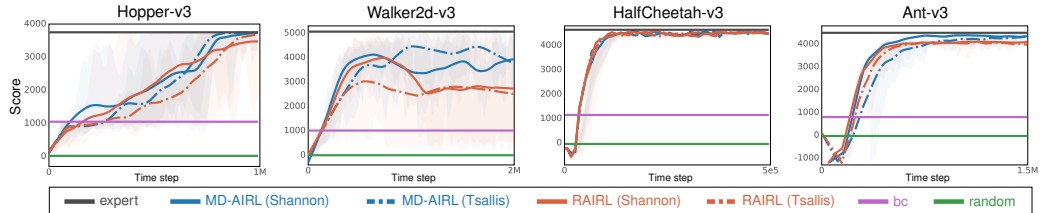

Figure 7: Average scores for 5 runs with two different regularizers (Shannon and Tsallis regularizer). The agent and IRL reward functions were trained with 4 episodes of expert demonstrations.

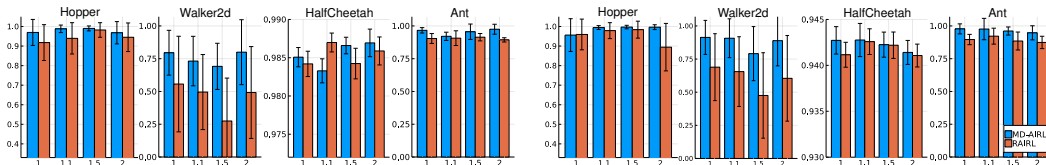

Figure 8: Scores on the last $10^5$ steps in a total of 32 different settings. The $x$-axis indicates the $q$ value of the Tsallis regularizers. The scores are rescaled by considering the expert performance as 1, and the error bars represent standard deviations. Left: 4 demonstrations. Right: 100 demonstrations.

to achieve consistent performance throughout the tasks and demonstration size. On the training curves, MD-AIRL showed high tolerance to the scarcity of data compared to RAIRL for 4 expert demonstrations. The plots in Fig. 8 indicate that MD-AIRL showed higher average scores with lower variance compared to RAIRL, across 30 distinct cases among 32 configurations we have tested. MD-AIRL inherits the scalability of AIL, and it is highly stable with respect to limited sample sizes.

Table 3: Scores on noisy demonstrations. The values of $\varepsilon$ represents scales of the Gaussian noises.

| | Method | $\varepsilon = 0.01$ | $\varepsilon = 0.5$ | | Method | $\varepsilon = 0.01$ | $\varepsilon = 0.5$ |
|---|---|---|---|---|---|---|---|
| **Hopper** | RAIRL (Shannon) | $3636.03 \pm 391.09$ | $3573.74 \pm 508.14$ | **HalfCheetah** | RAIRL (Shannon) | $4354.15 \pm 63.83$ | $4216.99 \pm 661.17$ |
| | MD-AIRL (Shannon) | $\mathbf{3669.25 \pm 177.78}$ | $\mathbf{3653.31 \pm 267.87}$ | | MD-AIRL (Shannon) | $\mathbf{4373.17 \pm 68.12}$ | $\mathbf{4337.18 \pm 106.40}$ |
| | RAIRL (Tsallis) | $3671.12 \pm 322.32$ | $3576.17 \pm 515.75$ | | RAIRL (Tsallis) | $4364.13 \pm 68.09$ | $4216.67 \pm 248.08$ |
| | MD-AIRL (Tsallis) | $\mathbf{3730.14 \pm 63.09}$ | $\mathbf{3701.24 \pm 205.68}$ | | MD-AIRL (Tsallis) | $\mathbf{4388.87 \pm 73.19}$ | $\mathbf{4247.44 \pm 266.73}$ |
| **Walker2d** | RAIRL (Shannon) | $2856.56 \pm 939.9$ | $2451.00 \pm 1392.6$ | **Ant** | RAIRL (Shannon) | $4493.74 \pm 383.04$ | $3777.78 \pm 505.78$ |
| | MD-AIRL (Shannon) | $\mathbf{3386.38 \pm 953.59}$ | $\mathbf{3252.65 \pm 1395.7}$ | | MD-AIRL (Shannon) | $\mathbf{4658.29 \pm 201.37}$ | $\mathbf{4284.38 \pm 329.79}$ |
| | RAIRL (Tsallis) | $2731.84 \pm 1058.7$ | $2435.10 \pm 1555.2$ | | RAIRL (Tsallis) | $4359.62 \pm 168.46$ | $3660.22 \pm 508.54$ |
| | MD-AIRL (Tsallis) | $\mathbf{3624.00 \pm 992.63}$ | $\mathbf{3093.54 \pm 963.96}$ | | MD-AIRL (Tsallis) | $\mathbf{4705.25 \pm 130.53}$ | $\mathbf{4127.37 \pm 457.25}$ |

**Noisy demonstrations.** Tab. 3 shows the results of imitation learning experiments for 100 expert demonstrations with two levels of Gaussian additive noises, resulting in suboptimal demonstrations. MD-AIRL is highly tolerant to noisy data, consistently achieving higher performance. The experiment is closely related to the general case in the theory; the results suggest that the characteristics of MD-AIRL are in alignment with our analyses of the MD reward learning scheme.

We present a detailed analysis of the noisy demonstration experiments (Fig. 9). Let the Bregman divergence between agent and ground-truth expert policies be the error, and we measured these errors by increasing the given noise level for the expert trajectories. Fig. 9 shows a general tendency that MD-AIRL has lower errors than RAIRL. With Tab. 3 and Fig. 9, we were able to find the evident correlation between average Bregman divergence and performance since imitation learning convergence when the divergence is zero. Thus, this is another piece of empirical evidence that verifies our theoretical claims.

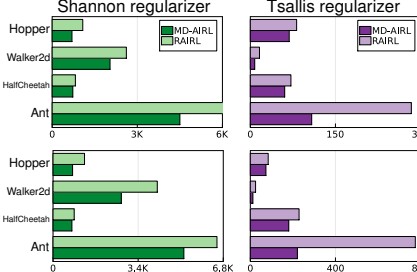

Figure 9: Divergences after imitation learning. Top: $\varepsilon = 0.01$. Bottom: $\varepsilon = 0.5$.

# 8 Conclusions and Discussion

In this paper, we presented MD-AIRL, a practical AIL algorithm designed to solve the imitation learning problem in the real world. We proved that the proposed method has clear advantages over previous AIL methods in terms of robustness. We verified MD-AIRL in a variety of situations, including high-dimensional spaces, limited samples, and imperfect demonstrations. The empirical

evidence showed that MD-AIRL outperforms previous methods on various benchmarks. We conclude that the rich foundation of optimization theories shows a promising direction for AIL studies.

Considering RL and IRL with geometric perspectives is vital for solving real-world problems. Although our work covers various imitation learning problems with the Bregman divergence, this does not include some other problems when the proximity term is of other statistical divergence families, such as the f-divergence [52]. If the relationship between these classes of divergences is studied in more detail, it is expected to proceed with applications to various subfields of machine learning. The assumptions on $\Omega$ in our analyses are usually justified by enforcing a specific policy space, but some outliers might have substantial meaning for certain tasks. Therefore, extensive analyses on these parameterizations remain as future works. The "impurity" of the MD-AIRL reward function compared to $\Psi_\Omega(\Pi)$ can be regarded as a limitation. To fully resolve this problem, all data must be treated as on-policy samples, which might require a sophisticated sampling mechanism.

**Societal impacts.** The evolution of imitation learning algorithms is expected to bring a structural shift in the labor market. The negative impact could be mitigated by diversification, unification, and redefinition of routine and manual jobs. The results of our work can be abused as a tool for analyzing individual data. Therefore, we stress that certain acts should be carefully regulated, such as collecting a substantial amount of individuals' data and aggressively tracking personal identity.

## Acknowledgments

The authors would like to thank the anonymous reviewers, Woosuk Choi, Jaein Kim, and Min Whoo Lee for their helpful discussion and comments. This work was partly supported by the IITP (2022-0-00951-LBA/25%, 2022-0-00953-PICA/25%, 2015-0-00310-SW.StarLab/10%, 2021-0-02068-AIHub/10%, 2021-0-01343-GSAI/10%, 2019-0-01371-BabyMind/10%) grant funded by the Korean government, and the CARAI (UD190031RD/10%) grant funded by the DAPA and ADD.

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
