# Robust Imitation via Mirror Descent Inverse Reinforcement Learning

## A Proofs

We denote the entire set of conditional distributions as $\Delta_{\mathcal{A}}^{\mathcal{S}}$, which is a vector space formed by a collection of $|\mathcal{S}|$ elements of unit $(|\mathcal{A}| - 1)$-simplexes: $\Delta_{\mathcal{A}} = \left\{ x_1 e_1 + \cdots + x_{|\mathcal{A}|} e_{|\mathcal{A}|} \mid \sum_{i=1}^{|\mathcal{A}|} x_i = 1 \text{ and } x_i \geq 0 \text{ for } i \in \mathcal{A} \right\}$. A trainable policy space is a subset of the entire conditional probability denoted as $\Pi := [\Pi^s]_{s \in \mathcal{S}} \subset \Delta_{\mathcal{A}}^{\mathcal{S}}$. We assume that $\Pi^s$ is a member of a specific Banach space called $L^p$ space $(\mathbb{R}^{\mathcal{A}}, \|\cdot\|)$, where $\|\cdot\|$ is a p-norm on $\mathcal{A}$. The dual space of $L^p$ space for $1 < p < \infty$ is $L^q$ space $(\mathbb{R}^{\mathcal{A}}, \|\cdot\|_*)$, where $\|\cdot\|_*$ is defined as a q-norm $(1/p + 1/q = 1)$. Here, the condition $1 < p \leq 2$ is assumed for existence and convergence properties in the dual $L^q$ space.

We begin with the following preliminary definitions: the Lipschitz continuity and martingales.

**Definition 2** (Lipschitz constants). Given two metric spaces $(X, d_X)$ and $(Y, d_Y)$ where $d_X$ denotes the metric on set $X$ and $d_Y$ is the metric on set $Y$, a function $f : X \to Y$ is called Lipschitz continuous if there exists a real constant $k \geq 0$ such that, for all $x_1$ and $x_2$ in $X$,

$$d_Y\big(f(x_1), f(x_2)\big) \leq k \cdot d_X(x_1, x_2). \tag{16}$$

In particular, a function $f$ is called Lipschitz continuous if there exists a constant $k \geq 0$ such that,

$$\big\|f(x_1) - f(x_2)\big\|_* \leq k \|x_1 - x_2\|, \quad \forall x_1, x_2 \tag{17}$$

where norms $\|\cdot\|$ and $\|\cdot\|_*$ are endowed with spaces $X$ and $Y$ respectively. For the smallest $L$ that substitutes $k$, $L$ is called the Lipschitz constant and $f$ is called a $L$-Lipschitz continuous function.

**Definition 3** (Discrete-time martingales). If a stochastic process $\{Z_t\}_{t \geq 1}$ satisfies $\mathbb{E}[|Z_n|] < \infty$ and

① $\mathbb{E}[Z_{n+1} | X_1, \ldots, X_n] \leq Z_n$, ② $\mathbb{E}[Z_{n+1} | X_1, \ldots, X_n] = Z_n$, ③ $\mathbb{E}[Z_{n+1} | X_1, \ldots, X_n] \geq Z_n$,

the stochastic process $\{Z_t\}_{t \geq 1}$ is called ① a submartingale, ② a martingale, and ③ a supermartingale, with respect to filtration $\{X_t\}_{t \geq 1}$.

The following arguments and proofs follow the results that appeared in previous literature for general aspects [15, 16, 42, 28, 17, 30]. Our analyses extend existing theoretical results to imitation learning and IRL; they are also highly general to cover various online methods for sequential decision problems.

### A.1 Proof of Lemma 1

*Proof of Lemma 1.* The conjugate operator of $\psi_\pi^s$ satisfies the following identity (Lemma 1 of [18])

$$\begin{aligned}
\Omega^*(\psi_\pi^s) &= \max_{\tilde{\pi}^s \in \Delta_{\mathcal{A}}} \langle \tilde{\pi}^s, \psi_\pi^s \rangle_{\mathcal{A}} - \Omega(\tilde{\pi}^s) \\
&= \max_{\tilde{\pi}^s \in \Delta_{\mathcal{A}}} \langle \tilde{\pi}^s, \nabla\Omega(\pi^s) \rangle_{\mathcal{A}} - \langle \pi^s, \nabla\Omega(\pi^s) \rangle_{\mathcal{A}} + \Omega(\pi^s) - \Omega(\tilde{\pi}^s) \\
&= \min_{\tilde{\pi}^s \in \Delta_{\mathcal{A}}} \Omega(\tilde{\pi}^s) - \Omega(\pi^s) - \langle \nabla\Omega(\pi^s), \tilde{\pi}^s - \pi^s \rangle_{\mathcal{A}} \\
&= \min_{\tilde{\pi}^s \in \Delta_{\mathcal{A}}} D_\Omega\big(\tilde{\pi}^s \| \pi^s\big),
\end{aligned}$$

for every state $s \in \mathcal{S}$. By the property of Bregman divergence and the convexity of $D_\Omega(\tilde{\pi}^s \| \pi^s)$ with respect to $\tilde{\pi}^s$, the optimal condition is obtained by the unique maximizing argument $\tilde{\pi}(\cdot|s) = \pi(\cdot|s)$. By taking gradient to both sides with respect to $\psi_\pi^s$ we yield $\pi^s = \nabla\Omega^*(\psi_\pi^s)$.

If there is another $\tilde{\pi} \in \Delta_{\mathcal{A}}^{\mathcal{S}}$ that makes $\psi_{\tilde{\pi}} = \psi_\pi$, this contradicts the property of unique maximizing arguments for conjugates since $\pi \in \Pi$ and $\Pi \subset \Delta_{\mathcal{A}}^{\mathcal{S}}$. Therefore, $\psi_\pi$ is uniquely defined for each $\pi$ and $\nabla\Omega^*(\psi^s) \in \Pi^s$ for all $s \in \mathcal{S}$. $\qquad\square$

## A.2 Proof of Theorem 1

Consider the unique fixed point of $\pi_*$ as the solution of $\inf_{\pi \in \Pi} \mathbb{E}[f(\pi, \tau_t)]$ where the expectation indicates that we consider all outputs with respect to $\tau_t$ for $t \to \infty$ i.e., $\lim_{t \to \infty} \mathbb{E}_{\tau_{1:t}}[f(\pi, \tau_t)]$. By equating derivatives to zero, we write the condition of fixed point $\pi_*$ as $\nabla\Omega(\pi_*) = \lim_{t \to \infty} \mathbb{E}[\nabla\Omega(\bar{\pi}_{E,t})]$. This assumption is useful since this paper provides some general results, the case of $\inf_{\pi \in \Pi} \mathbb{E}[f(\pi, \tau_t)] > 0$ in particular, which means that the estimates $\{\bar{\pi}_{E,t}\}_{t=1}^{\infty}$ do not converge to the fixed point of $\pi_*$, hence $\lim_{t \to \infty} \mathbb{E}_{\tau_{1:t}}[\|\pi_* - \bar{\pi}_{E,t}\|] \neq 0$. As a result, MD-based imitation learning algorithms allow many challenging settings, such as scarcity of data or imperfect demonstrations.

We first introduce a fundamental relationship regarding cumulative gradients in our online MD setting.

**Lemma 2.** *Let $\{\pi_t\}_{t=1}^{\infty}$, $\{\bar{\pi}_{E,t}\}_{t=1}^{\infty}$, and $\{\eta_t\}_{t=1}^{\infty}$ be policy, estimate, and step size sequences, respectively. The subsequent policy $\pi_{t+1}$ in Eq. (7) is obtained by an RL algorithm using the derivation of $\psi_{t+1}$ in Eq. (7), resulting to the following equation:*

$$\pi_{t+1}(\cdot|s) = \underset{\pi^s \in \Pi^s}{\operatorname{argmin}} \, \eta_t D_\Omega\big(\pi^s \big\| \bar{\pi}_{E,t}^s\big) + (1-\eta_t) D_\Omega\big(\pi^s \big\| \pi_t^s\big) \quad \forall s \in \mathcal{S}. \tag{18}$$

*We have for $t \in \mathbb{N}$,*

$$\eta_t\Big(\nabla\Omega\big(\pi_t^s\big) - \nabla\Omega\big(\bar{\pi}_{E,t}^s\big)\Big) = \nabla\Omega\big(\pi_t^s\big) - \nabla\Omega\big(\pi_{t+1}^s\big) \quad \forall s \in \mathcal{S}. \tag{19}$$

*Proof of Lemma 2.* Since the optimization problem is convex with respect to each $\pi^s$, we equate the derivatives at $\pi_{t+1}$ to zero:

$$\eta_t\Big(\nabla\Omega\big(\pi_{t+1}^s\big) - \nabla\Omega\big(\bar{\pi}_{E,t}^s\big)\Big) + (1-\eta_t)\Big(\nabla\Omega(\pi_{t+1}^s) - \nabla\Omega\big(\pi_t^s\big)\Big) = 0, \quad \forall \, s \in \mathcal{S}.$$

Then, we derive Eq. (19) as

$$\begin{aligned}
&\eta_t\Big(\nabla\Omega\big(\pi_{t+1}^s\big) - \nabla\Omega\big(\bar{\pi}_{E,t}^s\big)\Big) + (1-\eta_t)\Big(\nabla\Omega\big(\pi_{t+1}^s\big) - \nabla\Omega\big(\pi_t^s\big)\Big) = 0 \\
\Leftrightarrow \quad &\nabla\Omega\big(\pi_{t+1}^s\big) - \eta_t\nabla\Omega\big(\bar{\pi}_{E,t}^s\big) - (1-\eta_t)\nabla\Omega\big(\pi_t^s\big) = 0 \\
\Leftrightarrow \quad &\nabla\Omega\big(\pi_t^s\big) - \nabla\Omega\big(\pi_{t+1}^s\big) = \eta_t\Big(\nabla\Omega\big(\pi_t^s\big) - \nabla\Omega\big(\bar{\pi}_{E,t}^s\big)\Big) \quad \forall \, s \in \mathcal{S}.
\end{aligned}$$

Therefore, the proof is complete. $\qquad\square$

Lemma 2 indicates that the distances between dual maps are equivalent to $\eta_t\big\|\nabla\Omega(\bar{\pi}_{E,t}^s) - \nabla\Omega(\pi_t^s)\big\|_*$. Therefore, when the step size converges as $\lim_{t \to \infty} \eta_t = 0$, the convergence in the dual space is induced as $\lim_{t \to \infty}\big\|\nabla\Omega(\pi_t^s) - \nabla\Omega(\pi_{t+1}^s)\big\|_* = 0$; thus, the convergence of associated reward functions for every state in Section 4 is reasonable when $\Omega$ is strongly smooth.

In the following lemmas (Lemmas 3-5), we omit the given state for simplicity since they hold for $\forall s \in \mathcal{S}$, hence one can write distributions $\pi_a = \pi_a^s$, $\pi_b = \pi_b^s$, and $\pi_c = \pi_c^s$ for a arbitrary given state $s$. First, we reintroduce the three-point identity as follows.

**Lemma 3** (Three-point identity)**.** *Let $\pi_a$, $\pi_b$, and $\pi_c$ be any policy distributions with a given state. We have the following identity:*

$$\big\langle \nabla\Omega(\pi_a) - \nabla\Omega(\pi_b), \, \pi_c - \pi_b \big\rangle_{\mathcal{A}} = D_\Omega\big(\pi_c \big\| \pi_b\big) - D_\Omega\big(\pi_c \big\| \pi_a\big) + D_\Omega\big(\pi_b \big\| \pi_a\big)$$

*Proof of Lemma 3.* This can be derived using the definition of divergence as follows.

$$\begin{aligned}
D_\Omega(\pi_c\|\pi_b) - D_\Omega(\pi_c\|\pi_a) + D_\Omega(\pi_b\|\pi_a) &= \Omega(\pi_c) - \Omega(\pi_b) - \big\langle \nabla\Omega(\pi_b), \, \pi_c - \pi_b \big\rangle_{\mathcal{A}} \\
&\quad - \Omega(\pi_c) + \Omega(\pi_a) + \big\langle \nabla\Omega(\pi_a), \, \pi_c - \pi_a \big\rangle_{\mathcal{A}} \\
&\quad + \Omega(\pi_b) - \Omega(\pi_a) - \big\langle \nabla\Omega(\pi_a), \, \pi_b - \pi_a \big\rangle_{\mathcal{A}} \\
&= \big\langle \nabla\Omega(\pi_a) - \nabla\Omega(\pi_b), \, \pi_c - \pi_b \big\rangle_{\mathcal{A}}.
\end{aligned}$$

Therefore, the proof is complete. $\qquad\square$

Then, we introduce two identities in Lemmas 4 and 5 that are later used to address the progress of mirror descent updates in terms of Bregman divergences.

**Lemma 4.** *Let $\pi_a$, $\pi_b$, and $\pi_c$ be any policy distributions with a given state. The following identity holds.*

$$D_\Omega\big(\pi_c\big\|\pi_b\big) - D_\Omega\big(\pi_c\big\|\pi_a\big) = D_\Omega\big(\pi_a\big\|\pi_b\big) + \big\langle \nabla\Omega(\pi_a) - \nabla\Omega(\pi_b),\ \pi_c - \pi_a \big\rangle_{\mathcal{A}} \qquad (20)$$

*Proof of Lemma 4.* By Lemma 3, we have

$$D_\Omega\big(\pi_c\big\|\pi_b\big) - D_\Omega\big(\pi_c\big\|\pi_a\big) = -D_\Omega\big(\pi_b\big\|\pi_a\big) + \big\langle \nabla\Omega(\pi_a) - \nabla\Omega(\pi_b),\ \pi_c - \pi_b \big\rangle_{\mathcal{A}}.$$

Utilizing an identity of two Bregman divergences for arbitrary $(\pi, \tilde{\pi})$:

$$D_\Omega(\pi\|\tilde{\pi}) + D_\Omega(\tilde{\pi}\|\pi) = \big\langle \nabla\Omega(\pi) - \nabla\Omega(\tilde{\pi}),\ \pi - \tilde{\pi} \big\rangle_{\mathcal{A}}, \qquad (21)$$

we separate $\pi_c - \pi_b$ into $\pi_c - \pi_a$ and $\pi_a - \pi_b$ and write the rest of the derivation as follows.

$$D_\Omega\big(\pi_c\big\|\pi_b\big) - D_\Omega\big(\pi_c\big\|\pi_a\big)$$
$$= \underbrace{-D_\Omega\big(\pi_b\big\|\pi_a\big) + \big\langle \nabla\Omega(\pi_a) - \nabla\Omega(\pi_b),\ \pi_a - \pi_b \big\rangle_{\mathcal{A}}}_{\text{Eq. (21)}} + \big\langle \nabla\Omega(\pi_a) - \nabla\Omega(\pi_b),\ \pi_c - \pi_a \big\rangle_{\mathcal{A}}$$
$$= D_\Omega\big(\pi_a\big\|\pi_b\big) + \big\langle \nabla\Omega(\pi_a) - \nabla\Omega(\pi_b),\ \pi_c - \pi_a \big\rangle_{\mathcal{A}}$$

Therefore, we achieve the desired identity. $\qquad\qquad\square$

**Lemma 5.** *Let $\pi_a$, $\pi_b$, and $\pi_c$ be any policy distributions with a given state. The following identity holds.*

$$D_\Omega\big(\pi_b\big\|\pi_a\big) - D_\Omega\big(\pi_c\big\|\pi_a\big) = -\big\langle \nabla\Omega(\pi_c) - \nabla\Omega(\pi_a),\ \pi_c - \pi_b \big\rangle_{\mathcal{A}} + D_\Omega\big(\pi_b\big\|\pi_c\big) \qquad (22)$$

*Proof of Lemma 5.* By Lemma 3, we have

$$D_\Omega\big(\pi_b\big\|\pi_a\big) - D_\Omega\big(\pi_c\big\|\pi_a\big) = -D_\Omega\big(\pi_c\big\|\pi_b\big) + \big\langle \nabla\Omega(\pi_a) - \nabla\Omega(\pi_b),\ \pi_c - \pi_b \big\rangle_{\mathcal{A}}.$$

We separate $\nabla\Omega(\pi_a) - \nabla\Omega(\pi_b)$ into $\nabla\Omega(\pi_a) - \nabla\Omega(\pi_c)$ and $\nabla\Omega(\pi_c) - \nabla\Omega(\pi_b)$ and write the rest of the derivation as follows.

$$D_\Omega\big(\pi_b\big\|\pi_a\big) - D_\Omega\big(\pi_c\big\|\pi_a\big)$$
$$= \underbrace{-D_\Omega\big(\pi_c\big\|\pi_b\big) + \big\langle \nabla\Omega(\pi_c) - \nabla\Omega(\pi_b),\ \pi_c - \pi_b \big\rangle_{\mathcal{A}}}_{\text{Eq. (21)}} + \big\langle \nabla\Omega(\pi_a) - \nabla\Omega(\pi_c),\ \pi_c - \pi_b \big\rangle_{\mathcal{A}}$$
$$= D_\Omega\big(\pi_b\big\|\pi_c\big) + \big\langle \nabla\Omega(\pi_a) - \nabla\Omega(\pi_c),\ \pi_c - \pi_b \big\rangle_{\mathcal{A}}$$

Therefore, we achieve the desired identity. $\qquad\qquad\square$

Combining above lemmas, we show a key argument to prove Theorem 1 in the following lemma.

**Lemma 6.** *Assume $\inf_{\pi\in\Pi} \mathbb{E}[f(\pi, \tau_t)] > 0$. Assume that $\Omega$ is $\omega$-strongly convex and $\nabla\Omega$ is $L$-Lipschitz continuous for $\omega \geq 0$ and $L \geq 0$. If $\lim_{t\to\infty} \mathbb{E}_{\tau_{1:t}}\big[\sum_{i=0}^{\infty} \gamma^i D_\Omega\big(\pi_t(\cdot|s_i)\big\|\pi_E(\cdot|s_i)\big)\big] = 0$ for $\pi_E \in \Pi$, then $\{\eta_t\}_{t=1}^{\infty}$ satisfies Eq. (9). Furthermore, if $\Omega$ is strongly smooth, then Theorem 1 (a) holds with some constants $n \in \mathbb{N}$ and $c > 0$.*

*Proof of Lemma 6.* First, we show the condition of $\lim_{t\to\infty} \eta_t = 0$. Assuming all states are decomposable[1], the condition $\lim_{t\to\infty} \mathbb{E}_{\tau_{1:t}}\big[\sum_{i=0}^{\infty} \gamma^i D_\Omega\big(\pi_t(\cdot|s_i)\big\|\pi_E(\cdot|s_i)\big)\big] = 0$ implies $\lim_{t\to\infty} \mathbb{E}_{\tau_{1:t}}\big[\|\pi_t - \pi_E\|\big] = 0$, where $\|\cdot\|$ is the *matrix* norm induced by the p-norm on $\mathcal{A}$. Then, our aim is to show that the gradient of the strong convex function for $\pi_t$ converges to $\nabla\Omega(\pi_E)$, i.e.

$$\lim_{t\to\infty} \mathbb{E}_{\tau_{1:t}}\big[\big\|\nabla\Omega(\pi_t) - \nabla\Omega(\pi_E)\big\|_*\big] = 0, \qquad (23)$$

where $\|\cdot\|_*$ is the matrix norm induced by the q-norm and $\nabla\Omega(\pi)$ is a shorthand notation for $[\nabla\Omega(\pi_t^s)]_{s\in\mathcal{S}}$. To prove this argument, we use the continuity of $\nabla\Omega$ at $\pi_E$; this means for any $\varepsilon > 0$, there exists some $0 < \delta \leq 1$ such that $\|\nabla\Omega(\pi) - \nabla\Omega(\pi_E)\|_* < \varepsilon$ whenever $\|\pi - \pi_E\| < \delta$.

---

[1] The decomposability condition; Definition B.1 of Fu et al. [10]

When $\|\pi - \pi_E\| \geq \delta$, we apply the $L$-Lipschitz continuity assumption to find

$$\left\|\nabla\Omega(\pi) - \nabla\Omega(\pi_E)\right\|_* \leq L\|\pi - \pi_E\|, \tag{24}$$

where $\|\cdot\|_*$ is a matrix norm induced by the $q$-norm. Combining Eq. (23) and Eq. (24), we know that

$$\mathbb{E}_{\tau_{1:t}}\left[\left\|\nabla\Omega(\pi_t) - \nabla\Omega(\pi_E)\right\|_*\right] \leq \varepsilon + L \cdot \mathbb{E}_{\tau_{1:t}}\left[\|\pi_t - \pi_E\|\right]. \tag{25}$$

Since $\lim_{t\to\infty}\mathbb{E}_{\tau_{1:t}}\left[\|\pi_E - \pi_t\|\right] = 0$ ensures the existence of some $n \in \mathbb{N}$ such that for $t > n$, it holds that $\mathbb{E}_{\tau_{1:t}}\left[\|\pi_E - \pi_t\|\right] < \varepsilon/L$. Applying this inequality to Eq. (25), we have $\mathbb{E}_{\tau_{1:t}}\left[\left\|\nabla\Omega(\pi_t) - \nabla\Omega(\pi_E)\right\|_*\right] < 2\varepsilon$ for some $t > n$.

For temporal estimations, let us define the infimum of the expectation throughout the time as

$$\ell = \inf_{\pi\in\Delta_{\mathcal{A}}^{\mathcal{S}}} \mathbb{E}\left[\left\|\nabla\Omega(\pi_t) - \nabla\Omega(\bar{\pi}_{E,t})\right\|_*\right] > 0.$$

From Lemma 2, we have $\eta_t\left(\nabla\Omega(\pi_t^s) - \nabla\Omega(\bar{\pi}_{E,t}^s)\right) = \nabla\Omega(\pi_t^s) - \nabla\Omega(\pi_{t+1}^s)$ for every $s$. Taking the expectations, for every state $s$, the following inequality holds:

$$\eta_t\ell \leq \eta_t\mathbb{E}_{\tau_{1:t+1}}\left[\left\|\nabla\Omega(\pi_{t+1}^s) - \nabla\Omega(\bar{\pi}_{E,t}^s)\right\|_*\right] = \mathbb{E}_{\tau_{1:t+1}}\left[\left\|\nabla\Omega(\pi_t^s) - \nabla\Omega(\pi_{t+1}^s)\right\|_*\right] \quad \forall s \in \mathcal{S}.$$

Hence the convergence of the point $[\nabla\Omega(\pi_t^s)]_{s\in\mathcal{S}}$ is confirmed by taking the limit: $\lim_{t\to\infty}\eta_t = 0$.

Next, we show $\sum_{t=1}^{\infty}\eta_t = \infty$. By the $\omega$-strong convexity by the $L$-Lipschitz continuity of $\Omega$, we can find inequalities as

$$\left\langle\nabla\Omega(\pi^s) - \nabla\Omega(\tilde{\pi}^s),\, \pi^s - \tilde{\pi}^s\right\rangle_{\mathcal{A}} \leq L\|\pi^s - \tilde{\pi}^s\|^2 \leq \frac{2L}{\omega}D_\Omega(\pi^s\|\tilde{\pi}^s) \qquad \forall s \in \mathcal{S}. \tag{26}$$

We note that $\|\pi_{t+1}^s - \bar{\pi}_{E,t}^s\| \leq \|\pi_t^s - \bar{\pi}_{E,t}^s\|$ so that there is a constant $\varepsilon$ that satisfies $\mathbb{E}[\|\pi_{t+1}^s - \bar{\pi}_{E,t+1}^s\|] \geq \mathbb{E}[\|\pi_{t+1}^s - \bar{\pi}_{E,t}^s\|] + \varepsilon$. Therefore, taking expectations in Eq. (22) (and setting $\pi_a = \bar{\pi}_{E,t}^s$, $\pi_b = \pi_{t+1}^s$, and $\pi_c = \pi_t^s$) from Lemma 5, for the strongly convex $\Omega$, we can find

$$\begin{aligned}
\mathbb{E}_{\tau_{1:t+1}}\left[D_\Omega(\pi_{t+1}^s\|\bar{\pi}_{E,t+1}^s)\right] &\geq \mathbb{E}_{\tau_{1:t+1}}\left[D_\Omega(\pi_{t+1}^s\|\bar{\pi}_{E,t}^s)\right] + \varepsilon' \\
&\geq (1-a\eta_t)\,\mathbb{E}_{\tau_{1:t}}\left[D_\Omega(\pi_t^s\|\bar{\pi}_{E,t}^s)\right] + \mathbb{E}_{\tau_{1:t+1}}\left[D_\Omega(\pi_{t+1}^s\|\pi_t^s)\right] + \varepsilon' \qquad \textit{Eq. (22)} \\
&\geq (1-a\eta_t)\,\mathbb{E}_{\tau_{1:t}}\left[D_\Omega(\pi_t^s\|\bar{\pi}_{E,t}^s)\right] + \varepsilon'' \qquad \forall s \in \mathcal{S}, \tag{27}
\end{aligned}$$

for some $t$ and $0 < \varepsilon' < \varepsilon''$ when for $\lim_{t\to\infty}\eta_t = 0$. The positive constant $a := 2L/\omega$ is derived by the inequalities in Eq. (26).

Since $\lim_{t\to\infty}\eta_t = 0$, we can also find a constant $n \in \mathbb{N}$ such that $\eta_t \leq (3a)^{-1}$ for $t \geq n$. Applying the inequality $1 - x > \exp(-2x)$ for $x \in (0, 1/3]$, we derive another inequality

$$\mathbb{E}_{\tau_{1:t+1}}\left[D_\Omega(\pi_{t+1}^s\|\bar{\pi}_{E,t+1}^s)\right] \geq \exp(-2a\eta_t)\,\mathbb{E}_{\tau_{1:t}}\left[D_\Omega(\pi_t^s\|\bar{\pi}_{E,t}^s)\right], \quad \forall\, t \geq n \quad \forall s \in \mathcal{S}. \tag{28}$$

Applying this for $t = T-1, \ldots, n$ yields

$$\begin{aligned}
\mathbb{E}_{\tau_{1:T}}\left[D_\Omega\left(\pi_T^s\|\bar{\pi}_{E,T}^s\right)\right] &\geq \left(\prod_{t=n+1}^{T}\exp(-2a\eta_t)\right)\mathbb{E}_{\tau_{1:n}}\left[D_\Omega\left(\pi_n^s\|\bar{\pi}_{E,n}^s\right)\right] \\
&= \exp\left(-2a\cdot\sum_{t=n+1}^{T}\eta_t\right)\mathbb{E}_{\tau_{1:n}}\left[D_\Omega\left(\pi_n^s\|\bar{\pi}_{E,n}^s\right)\right]. \tag{29}
\end{aligned}$$

Using Eq. (29), we conclude $\mathbb{E}_{\tau_{1:n}}\left[D_\Omega(\pi_n^s\|\bar{\pi}_{E,n}^s)\right] > 0$ for some $s$. Otherwise, we have that

$$\mathbb{E}_{\tau_{1:n}}\left[D_\Omega(\pi_n^s\|\bar{\pi}_{E,n}^s)\right] = \mathbb{E}_{\tau_{1:n+1}}\left[D_\Omega(\pi_{n+1}^s\|\bar{\pi}_{E,n+1}^s)\right] = 0 \quad \forall s \in \mathcal{S},$$

which leads to $\mathbb{E}_{\tau_{1:n}}\left[\|\pi_n - \bar{\pi}_{E,n}\|^2\right] = \mathbb{E}_{\tau_{1:n+1}}\left[\|\pi_{n+1} - \bar{\pi}_{E,n+1}\|^2\right] = 0$, according to Eq. (28). This implies $\pi_n = \bar{\pi}_{E,n} = \pi_{n+1}$ almost surely, leading to $\mathbb{E}[f(\pi_t, \tau_t)] = 0$. Essentially, this is a contradiction to the previous assumption $\inf_{\pi\in\Delta_{\mathcal{A}}^{\mathcal{S}}}\mathbb{E}[f(\pi, \tau_t)] > 0$; thus, $\mathbb{E}_{\tau_{1:n+1}}\left[D_\Omega(\pi_{n+1}\|\bar{\pi}_{E,n+1})\right] > 0$. Let us assume the ideal case in which the estimation process learns the exact $\pi_E$ in $t \to \infty$. To satisfy the limit $\lim_{T\to\infty}\mathbb{E}_{\tau_{1:T}}\left[D_\Omega(\pi_T\|\bar{\pi}_{E,T})\right] = 0$ we see from Eq. (29) that $\sum_{t=1}^{\infty}\eta_t = \infty$.

Now, we show that Theorem 1 (a) holds. Since $\Omega$ is $\omega$-strongly convex, so basically $\Omega^*$ is $(\omega^{-1})$-strongly smooth with respect to $\|\cdot\|_*$. On the other hand, the $L$-Lipschitz continuity of $\nabla\Omega$ implies $L$-strong smoothness of $\Omega$; thus, $\Omega^*$ is $L$-strongly convex.

Since $\lim_{t\to\infty}\left\|\nabla\Omega(\pi_t^s) - \nabla\Omega(\pi_{t+1}^s)\right\|_* = 0$ for $\forall t \geq n$ and $\forall s \in \mathcal{S}$, the condition $\eta_t \leq (3a)^{-1}$ induces

$$\mathbb{E}_{\tau_{1:t+1}}\left[D_\Omega(\pi_{t+1}^s\|\bar\pi_{E,t+1}^s)\right]$$
$$\geq (1-a\eta_t)\,\mathbb{E}_{\tau_{1:t}}\left[D(\pi_t^s\|\bar\pi_{E,t}^s)\right] + (2L)^{-1}\mathbb{E}_{\tau_{1:t+1}}\left[\left\|\nabla\Omega(\pi_t) - \nabla\Omega(\pi_{t+1})\right\|_*^2\right],$$
$$\geq (1-a\eta_t)\,\mathbb{E}_{\tau_{1:t}}\left[D_\Omega(\pi_t^s\|\bar\pi_{E,t}^s)\right] + (2L)^{-1}\eta_t^2\,\mathbb{E}_{\tau_{1:t+1}}\left[\left\|\nabla\Omega(\pi_t) - \nabla\Omega(\bar\pi_{E,t})\right\|_*\right]. \quad \textit{Lemma 2}$$

Using the Cauchy-Schwarz inequality, we obtain a lower bound of the last term as

$$\mathbb{E}_{\tau_{1:t}}\left[\left\|\nabla\Omega(\pi_t) - \nabla\Omega(\bar\pi_{E,t})\right\|_*^2\right] \geq \left\{\mathbb{E}_{\tau_{1:t}}\left[\left\|\nabla\Omega(\pi_t) - \nabla\Omega(\bar\pi_{E,t})\right\|_*\right]\right\}^2 \geq \ell^2.$$

Thus, we obtain the final inequality for all $s \in \mathcal{S}$ as

$$\mathbb{E}_{\tau_{1:t+1}}\left[D_\Omega(\pi_{t+1}^s\|\bar\pi_{E,t+1}^s)\right] \geq (1 - a\eta_t)\,\mathbb{E}_{\tau_{1:t}}\left[D_\Omega(\pi_t^s\|\bar\pi_{E,t}^s)\right] + (2L)^{-1}(\eta_t\ell)^2, \quad \forall\, t \geq n.$$

Applying this inequality from $t = T \geq n+1$ to $t = n+1$, we achieve

$$\mathbb{E}_{\tau_{1:T+1}}\left[D_\Omega(\pi_{T+1}^s\|\bar\pi_{E,T+1}^s)\right] \geq \mathbb{E}_{\tau_{1:n}}\left[D_\Omega(\pi_n^s\|\bar\pi_{E,n}^s)\right] \prod_{t=n+1}^{T}(1 - a\eta_t)$$
$$+ (2L)^{-1}\ell^2 \sum_{t=n+1}^{T}\eta_t^2 \prod_{k=t+1}^{T}(1 - a\eta_k)$$
$$\geq (2L)^{-1}\ell^2 \sum_{t=n+1}^{T}\eta_t^2 \prod_{k=t+1}^{T}(1 - a\eta_k).$$

By the Cauchy-Schwarz inequality and our bound $0 < 1 - a\eta_k \leq 1$ for $k \geq n$, we have

$$\sum_{t=n+1}^{T}\eta_t \prod_{k=t+1}^{T}(1 - a\eta_k) \leq \left\{\sum_{t=n+1}^{T}\eta_t^2 \prod_{k=t+1}^{T}(1 - a\eta_k)\right\}^{1/2}(T - n)^{1/2}.$$

Hence

$$\sum_{t=n+1}^{T}\eta_t^2 \prod_{k=t+1}^{T}(1 - a\eta_k) \geq \frac{1}{a^2(T-n)}\left(\sum_{t=n+1}^{T}a\eta_t \prod_{k=t+1}^{T}(1 - a\eta_k)\right)^2$$
$$= \frac{1}{a^2(T-n)}\left(\sum_{t=n+1}^{T}(1 - (1 - a\eta_t))\prod_{k=t+1}^{T}(1 - a\eta_k)\right)^2$$
$$= \frac{1}{a^2(T-n)}\left(\sum_{t=n+1}^{T}\left[\prod_{k=t+1}^{T}(1 - a\eta_k) - \prod_{k=t}^{T}(1 - a\eta_k)\right]\right)^2$$
$$\geq \frac{1}{a^2(T-n)}\left(\sum_{t=n+1}^{T}1 - \prod_{k=t}^{T}(1 - a\eta_k)\right)^2$$
$$\geq \frac{1}{a^2(T-n)}\left(1 - (1 - a\eta_{n+1})\right)^2 = \frac{\eta_{n+1}^2}{T-n}$$

Therefore, we obtain the lower bound of

$$\mathbb{E}_{\tau_{1:T}}\left[D_\Omega(\pi_{T+1}\|\bar\pi_{E,T+1})\right] \geq \frac{\eta_{n+1}^2(2L)^{-1}\ell^2}{T-n}.$$

Since the Bregman divergence is assumed to be bounded for all states, the sequence $\{\gamma^i D_\Omega(\pi_t^{s_i}\|\pi_t^{s_i})\}$ will converge as $i \to \infty$. Applying the monotone convergence theorem, we can interchange expectation and summation, which yields

$$
\begin{aligned}
\mathbb{E}_{\tau_{1:T}}\left[\sum_{i=0}^{\infty}\gamma^i D_\Omega\Big(\pi_T(\,\cdot\,|s_i)\,\big\|\,\bar\pi_{E,T}(\,\cdot\,|s_i)\Big)\right] &= \sum_{i=0}^{\infty}\mathbb{E}_{\tau_{1:T}}\Big[\gamma^i D_\Omega\big(\pi_T(\,\cdot\,|s_i)\,\big\|\,\bar\pi_{E,T}(\cdot|s_i)\big)\Big] \\
&= \sum_{i=0}^{\infty}\gamma^i \mathbb{E}_{\tau_{1:T}}\Big[D_\Omega\big(\pi_T(\cdot|s_i)\,\big\|\,\bar\pi_{E,T}(\cdot|s_i)\big)\Big] \\
&\geq \frac{\eta_{n+1}^2(2L-2L\gamma)^{-1}\ell^2}{T-n}, \quad \forall\, T \geq n.
\end{aligned}
$$

This verifies Theorem 1 (a) with the constant $c = \eta_{n+1}^2(2L-2L\gamma)^{-1}\ell^2$. $\qquad\square$

Lastly, we show convergence to a unique fixed point of $\pi_*$ using the particular form of $\eta_t$ in Eq. (9).

**Lemma 7.** *If $\{\eta_t\}_{t=1}^{\infty}$ satisfies Eq. (9), $\lim_{t\to\infty}\mathbb{E}_{\tau_{1:t}}[\sum_{i=0}^{\infty}\gamma^i D_\Omega(\pi_*^s\|\pi_t^s)] = 0$. Furthermore, if the step size takes the form $\eta_t = \frac{4}{t+1}$, then $\mathbb{E}_{\tau_{1:T}}[\sum_{i=0}^{\infty}\gamma^i D_\Omega(\pi_*^{s_i}\|\pi_T^{s_i})] = \mathcal{O}(1/T)$.*

*Proof of Lemma 7.* According to Lemma 4 and the fundamental identity of Bregman divergence for the convex conjugate $\Omega^*$, the one-step progress regarding $\bar\pi_{E,t}^s$ can be written as

$$
\begin{aligned}
D_\Omega(\pi_*^s\|\pi_{t+1}^s) - D_\Omega(\pi_*^s\|\pi_t^s) &= \big\langle \nabla\Omega(\pi_t^s)-\nabla\Omega(\pi_{t+1}^s),\ \pi_*^s-\pi_t^s\big\rangle_{\mathcal{A}} + D_\Omega(\pi_t^s\|\pi_{t+1}^s) \\
&= \eta_t\big\langle \nabla\Omega(\pi_t^s)-\nabla\Omega(\bar\pi_{E,t}^s),\ \pi_*^s-\pi_t^s\big\rangle_{\mathcal{A}} + D_{\Omega^*}\big(\nabla\Omega(\pi_{t+1}^s)\big\|\nabla\Omega(\pi_t^s)\big),
\end{aligned}
\tag{30}
$$

for all $s \in \mathcal{S}$. As $\omega$-strong convexity of $\Omega$ implies the $(\omega^{-1})$-strong smoothness of $\Omega^*$, we have

$$
D_{\Omega^*}\big(\nabla\Omega(\pi_{t+1}^s)\big\|\nabla\Omega(\pi_t^s)\big) \leq \frac{1}{2\omega}\big\|\nabla\Omega(\pi_{t+1}^s)-\nabla\Omega(\pi_t^s)\big\|_*^2 = \frac{\eta_t^2}{2\omega}\big\|\nabla\Omega(\bar\pi_{E,t}^s)-\nabla\Omega(\pi_t^s)\big\|_*^2 \tag{31}
$$

Then, we bound $\|\nabla\Omega(\bar\pi_{E,t}^s)-\nabla\Omega(\pi_t^s)\|_*^2$ by $2\|\nabla\Omega(\pi_t^s)-\nabla\Omega(\pi_*^s)\|_*^2 + 2\|\nabla\Omega(\pi_*^s)-\nabla\Omega(\bar\pi_{E,t}^s)\|_*^2$, following the work of Lei and Zhou [17]. Since $\nabla\Omega$ is cocoercive with $\frac{1}{L}$ by the Lipschitz continuity of $\nabla\Omega$, we obtain

$$
\big\|\nabla\Omega(\pi_t^s)-\nabla\Omega(\pi_*^s)\big\|_*^2 \leq L\big\langle\nabla\Omega(\pi_*^s)-\nabla\Omega(\pi_t^s),\pi_*^s-\pi_t^s\big\rangle
$$

thus, using Eq. (30), we get

$$
\begin{aligned}
D_\Omega(\pi_*^s\|\pi_{t+1}^s) - D_\Omega(\pi_*^s\|\pi_t^s) &\leq \eta_t\big\langle\nabla\Omega(\pi_*^s)-\nabla\Omega(\bar\pi_{E,t}^s),\pi_*^s-\pi_t^s\big\rangle \\
&- \left(1-\frac{\eta_t L}{\omega}\right)\eta_t\big\langle\nabla\Omega(\pi_*^s)-\nabla\Omega(\pi_t^s),\pi_*^s-\pi_t^s\big\rangle + \frac{\eta_t^2}{\omega}\big(\|\nabla\Omega(\pi_*^s)-\nabla\Omega(\bar\pi_{E,t}^s)\|_*^2\big).
\end{aligned}
\tag{32}
$$

By taking expectation, it follows that there exists $n \in \mathbb{N}$ such that $\eta_t \leq \frac{\omega}{2L}$ for $t \geq n$ holds

$$
\begin{aligned}
\mathbb{E}_{\tau_{1:t+1}}\big[D_\Omega(\pi_*^s\|\pi_{t+1}^s)\big] &\leq \mathbb{E}_{\tau_{1:t}}\left[D_\Omega\big(\pi_*^s\big\|\pi_t^s\big) - \frac{\eta_t}{2}D_\Omega\big(\pi_*^s\big\|\pi_t^s\big) + \frac{\eta_t^2}{\omega}\big\|\nabla\Omega(\pi_*^s)-\nabla\Omega(\bar\pi_{E,t}^s)\big\|_*^2\right], \\
&\leq \mathbb{E}_{\tau_{1:t}}\left[D_\Omega\big(\pi_*^s\big\|\pi_t^s\big) - \frac{\eta_t}{2}D_\Omega\big(\pi_*^s\big\|\pi_t^s\big)\right] + z\eta_t^2,
\end{aligned}
\tag{33}
$$

where $z$ is the constant $z = \frac{1}{\omega}\mathbb{E}[\|\nabla\Omega(\pi_*)-\Omega(\bar\pi_{E,t})\|_*^2]$. Let $\{A_t\}_{t=1}^{\infty}$ denote a sequence of $A_t = \sup_{s\in\mathcal{S}}\mathbb{E}_{\tau_{1:t}}\big[D_\Omega(\pi_*^s\|\pi_t^s)\big]$. Then we have

$$
A_{t+1} \leq \left(1-\frac{\eta_t}{2}\right)A_t + z\eta_t^2, \quad \forall t \geq n. \tag{34}
$$

For a constant $h > 0$, we claim that $A_{t_1} < h$ for some $t_1 > n'$. Assume that this is not true, and we find some $t_2 \geq t_1$ such that $A_t > h, \forall t \geq t_2$. Since $\lim_{t\to\infty}\eta_t = 0$, there are some $t > t_3 > t_2$ that $\eta_t \leq \frac{h}{4b}$. However, Eq. (34) tells us that for $t \geq t_3$,

$$
A_{t+1} \leq \left(1-\frac{\eta_t}{2}\right)A_t + z\eta_t^2 \leq A_{t_3} - \frac{h}{4}\sum_{k=t_\gamma'}^{t}\eta_k \to -\infty \quad (\text{as } t \to \infty).
$$

This is a contradiction, which verifies $A_t < h$ for $t > n'$. Since $\lim_{t \to \infty} \eta_t = 0$, we can find some $\eta_t$ that makes $A_t$ monotonically decreasing. Then, we can conclude that the nonnegative sequence $\{A_t\}_{t=1}^{\infty}$ converges by iteratively applying the upper bounds.

We now prove Theorem 1 (b) under the consideration of the condition $\eta_t = \frac{4}{t+1}$. The estimate becomes

$$A_{t+1} \leq \left(1 - \frac{2}{t+1}\right) A_t + \frac{16z}{(t+1)^2}, \quad \forall t \geq n.$$

It follows the recurrence relation is

$$t(t+1)A_{t+1} \leq (t-1)tA_t + 16z, \quad \forall t \geq n.$$

Iteratively applying this relation, we obtain the general form of inequality.

$$(T-1)TA_T \leq (n-1)nA_n + 16z(T-n), \quad \forall T \geq n,$$

therefore we obtain the inequality as follows:

$$\mathbb{E}_{\tau_{1:T}}\left[D_\Omega(\pi_*^s \| \pi_T^s)\right] \leq \frac{(n-1)n\mathbb{E}_{\tau_{1:n}}\left[D_\Omega(\pi_*^s \| \pi_n^s)\right]}{(T-1)T} + \frac{16z}{T}, \quad \forall T \geq n, \quad \forall s \in \mathcal{S}.$$

By applying the monotone convergence theorem, we can interchange expectation and summation, which yields similar result to formulation from Proof of Lemma 6

$$\mathbb{E}_{\tau_{1:T}}\left[\sum_{i=1}^{\infty} \gamma^i D_\Omega\Big(\pi_*(\cdot \,|\, s_i) \,\big\|\, \pi_T(\cdot \,|\, s_i)\Big)\right] = \mathcal{O}\left(\frac{1}{T}\right).$$

Therefore, the proof is complete. $\qquad \square$

### A.3  Proof of Theorem 2

*Necessity.* First, we rewrite the inequality in Eq. (27) as

$$\mathbb{E}_{\tau_{1:t+1}}\left[D_\Omega\left(\pi_{t+1}^s \big\| \bar{\pi}_{E,t+1}^s\right)\right] \geq (1 - 2L\omega^{-1}\eta_t)\,\mathbb{E}_{\tau_{1:t}}\left[D_\Omega\left(\pi_t^s \big\| \bar{\pi}_{E,t}^s\right)\right], \quad \forall s \in \mathcal{S}. \tag{35}$$

Since we assume that $\eta_t$ converges to 0 from previous arguments, consider the step size sequence $0 < \eta_t \leq \frac{\omega}{(2+\kappa)L}$ for $\kappa > 0$ and $t \geq n$ where $\forall n \in \mathbb{N}$. Denote a constant $\tilde{a} = \frac{2+\kappa}{2} \log \frac{2+\kappa}{\kappa}$ and apply the elementary inequality

$$1 - x \geq \exp(-\tilde{a}x), \quad \text{such that } 0 < x \leq \frac{2}{2+\kappa}$$

From Eq. (35), it can be obtained that

$$\mathbb{E}_{\tau_{1:t+1}}\left[D_\Omega\left(\pi_{t+1}^s \big\| \bar{\pi}_{E,t+1}^s\right)\right] \geq \exp(-2\tilde{a}L\omega^{-1}\eta_t)\mathbb{E}_{\tau_{1:t}}\left[D_\Omega(\pi_t^s \big\| \bar{\pi}_{E,t}^s)\right].$$

Applying this inequality iteratively for $t = n, \ldots, T-1$ gives

$$\mathbb{E}_{\tau_{1:T}}\left[D_\Omega\left(\pi_T^s \big\| \bar{\pi}_{E,T}^s\right)\right] \geq \mathbb{E}_{\tau_{1:n}}\left[D_\Omega\left(\pi_n^s \big\| \bar{\pi}_{E,n}^s\right)\right] \prod_{t=n}^{T-1} \exp(-2\tilde{a}L\omega^{-1}\eta_t)$$

$$= \exp\left\{-2\tilde{a}L\omega^{-1}\sum_{t=n}^{T-1} \eta_t\right\} \mathbb{E}_{\tau_{1:n}}\left[D_\Omega\left(\pi_n^s \big\| \bar{\pi}_{E,n}^s\right)\right] \quad \forall s \in \mathcal{S}. \tag{36}$$

From the assumption $\pi_E \neq \pi_n$, we have $D_\Omega\left(\pi_n^s \big\| \bar{\pi}_{E,n}^s\right) > 0$ for some states. Therefore, by Eq. (36), the convergence $\lim_{t \to \infty} \mathbb{E}_{\tau_{1:t}}\left[D_\Omega(\pi_t^s \| \bar{\pi}_{E,t}^s)\right] = 0$ for all states implies $\sum_{t=1}^{\infty} \eta_t = \infty$.

*Sufficiency.* We use Eq. (34) in the proof of Lemma 7. In the optimal case, $\|\nabla\Omega(\pi_*) - \Omega(\bar{\pi}_{E,t})\|_* = 0$, so (34) takes the form (we can choose $n = 1$ by Eq. (33))

$$A_{t+1} \leq \frac{\eta_t}{2} A_t, \quad \forall t \in \mathbb{N}, \tag{37}$$

where $A_t = \sup_{s \in \mathcal{S}} \mathbb{E}_{\tau_{1:t}} \big[ D_\Omega(\pi_*^s \| \pi_t^s) \big]$ (and also $A_t = \sup_{s \in \mathcal{S}} \mathbb{E}_{\tau_{1:t}} \big[ D_\Omega(\pi_E^s \| \pi_t^s) \big]$ for the specific parameterization of $\pi_E \in \Pi$). Therefore, for any $0 < h < 1$, there must exist some $t_1 \in \mathbb{N}$ such that $A_t \leq h$ for $t \geq t_1$. Otherwise, $A_t > h$ for every $t \geq t_2$ with $t_2 \geq t_1$, which leads to a contradiction:

$$A_{t+1} \leq A_{t_2} - \frac{h}{2} \sum_{k=t_1}^{t} \eta_k \rightarrow -\infty \quad \text{(as } t \rightarrow \infty\text{).}$$

Eq. (37) also tells us that the sequence $\{A_t\}_{t=1}^\infty$ is monotonically decreasing. Hence $A_t \leq h$ for every $t \geq t_1$, which proves the convergence with respect to the least upper bound of Bregman divergences by combining with Eq. (37)

$$\lim_{t \to \infty} \sup_{s \in \mathcal{S}} \mathbb{E}_{\tau_{1:t}} \big[ D_\Omega(\pi_*^s \| \pi_t^s) \big] = \lim_{t \to \infty} A_t = 0.$$

We now prove the second point in Theorem 2 which is under the special condition of $\eta_t \equiv \eta_1$. It follows from Eq. (35) that $A_T \geq (1 - 2L\omega^{-1}\eta_1)^{T-1} A_1$. Hence, Eq (37) translates to

$$A_{t+1} \leq (1 - \eta_1/2)A_t,$$

from which we find $A_T \leq (1 - \eta_1/2)^{T-1} A_1$ by iteration starting from $t = 1$. Therefore, the second point is verified the theorem with $c_1 = 1 - \frac{2L\eta_1}{\omega}$ and $c_2 = 1 - \frac{\eta_1}{2}$. $\qquad \square$

## A.4 Proof of Proposition 1

The proof of Proposition 1 is based on the Doob's forward convergence theorem.

**Theorem 3** (Doob's forward convergence theorem). *Let $\{X_t\}_{t \in \mathbb{N}}$ be a sequence of nonnegative random variables and let $\{\mathcal{F}_t\}_{t \in \mathbb{N}}$ be a filtration with $\mathcal{F}_t \subset \mathcal{F}_{t+1}$ for every $t \in \mathbb{N}$. Assume that $\mathbb{E}\big[X_{t+1} | \mathcal{F}_t\big] \leq X_t$ almost surely for every $t \in \mathbb{N}$. Then the sequence $\{X_t\}$ converges to a nonnegative random variable $X_\infty$ almost surely.*

We follow the proof of Lemma 7 and apply Eq. (32). Since $\langle \pi_*^s - \pi_t^s, \nabla\Omega(\pi_*^s) - \nabla\Omega(\pi_t^s) \rangle \geq 0$ for all $s \in \mathcal{S}$, Eq. (32) implies: there exists $n \in \mathbb{N}$ that

$$\mathbb{E}_{\tau_t}\big[ D_\Omega(\pi_*^s \| \pi_{t+1}^s) \big] \leq D_\Omega\big(\pi_*^s \big\| \pi_t^s\big) + \frac{\eta_t^2}{\omega} \mathbb{E}\Big[ \big\| \nabla\Omega(\pi_*) - \nabla\Omega(\bar{\pi}_{E,t}) \big\|_*^2 \Big], \quad \forall t \geq n, \forall s \in \mathcal{S}, \quad (38)$$

and since the step size is scheduled as $\lim_{t \to \infty} \eta_t = 0$, the following equation also holds:

$$\mathbb{E}_{\tau_t}\bigg[ \sup_{s \in \mathcal{S}} D_\Omega(\pi_*^s \| \pi_{t+1}^s) \bigg] \leq \sup_{s \in \mathcal{S}} D_\Omega\big(\pi_*^s \big\| \pi_t^s\big) + \frac{\eta_t^2}{\omega} \mathbb{E}\Big[ \big\| \nabla\Omega(\pi_*) - \nabla\Omega(\bar{\pi}_{E,t}) \big\|_*^2 \Big], \quad \forall t \geq n', \quad (39)$$

for some $n' \in \mathbb{N}$. Then, the condition $\sum_{t=1}^\infty \eta_t^2 < \infty$ enables us to define a stochastic process $\{X_t\}$:

$$X_t := \sup_{s \in \mathcal{S}} D_\Omega(\pi_*^s \| \pi_{t+1}^s) + \frac{1}{\omega} \mathbb{E}\Big[ \big\| \nabla\Omega(\pi_*^s) - \nabla\Omega(\bar{\pi}_{E,t}^s) \big\|_*^2 \Big] \sum_{i=t+1}^\infty \eta_i^2.$$

Thus, by Eq. (39), it is straightforwardly derived that there exits $n \in \mathbb{N}$ that $\mathbb{E}_{\tau_t}[X_{t+1}] \leq X_t$ for $t \geq n$. Since $X_t \geq 0$, the stochastic process $\{X_t\}_{t-n+1 \geq 1}$ is a submartingale (equivalently, $\{-X_t\}_{t-n+1 \geq 1}$ is a supermartingale). By Theorem 3, the sequence $\{X_t\}_{t \geq 1}$ converges to a nonnegative random variable $X_\infty$ almost surely. Therefore, $D_\Omega(\pi_*^s \| \pi_t^s)$ converges for every state.

According to Fatou's lemma, and using the convergence of $\lim_t \mathbb{E}_{\tau_{1:t}} \big[ \sum_{i=0}^\infty \gamma^i D_\Omega(\pi_*^s \| \pi_t^s) \big] = 0$ proved by Lemma 7, we obtain

$$\mathbb{E}\bigg[ \lim_{t \to \infty} \sum_{i=0}^\infty \gamma^i D_\Omega\Big( \pi_*\big(\cdot | s_i\big) \Big\| \pi_t\big(\cdot | s_i\big) \Big) \bigg] \leq (1-\gamma)^{-1} \liminf_{t \to \infty} \mathbb{E}_{\tau_{1:t}} \bigg[ \sum_{i=0}^\infty \gamma^i D_\Omega(\pi_*^s \| \pi_t^s) \bigg] = 0.$$

Therefore, it can be concluded that the sequence of costs $\Big\{ \sum_{i=0}^\infty \gamma^i D_\Omega\big( \pi_*(\cdot|s_i) \big\| \pi_t(\cdot|s_i) \big) \Big\}_{t \in \mathbb{N}}$ converges to 0 almost surely. $\qquad \square$

## B Tsallis Entropy and Associated Bregman Divergence Among Full Covariance Multivariate Gaussian Distributions

This appendix **reintroduces** derivations of Bregman divergences and regularized reward functions for tractable computation when $\Omega$ is the Tsallis entropy regularizer, which were previously proposed by Nielsen and Nock [53] and Jeon et al. [18]. And then, we delineate a distinct parameterization used in this paper for modeling Gaussian distribution policies equipped with full covariance matrices.

The standard form of the exponential family is represented as

$$\exp\{\langle\theta, t(x)\rangle - F(\theta) + k(x)\}. \tag{40}$$

The generalized parameterization of the multi-variate Gaussian is defined as follows:

$$\theta = \begin{bmatrix} \Sigma^{-1}\mu \\ -\frac{1}{2}\Sigma^{-1} \end{bmatrix} = \begin{bmatrix} \theta_1 \\ \theta_2 \end{bmatrix},$$

$$t(x) = \begin{bmatrix} x \\ xx^\mathsf{T} \end{bmatrix},$$

$$F(\theta) = -\frac{1}{4}\theta_1^\mathsf{T}\theta_2^{-1}\theta_1 + \frac{1}{2}\ln|-\pi\theta_2^{-1}| = \frac{1}{2}\mu^\mathsf{T}\Sigma^{-1}\mu + \frac{1}{2}\ln(2\pi)^d|\Sigma|,$$

$$k(x) = 0,$$

where we can analytically recover the Gaussian distribution [53]

$$\exp\{\langle\theta, t(x)\rangle - F(\theta) + k(x)\}$$

$$= \exp\left\{\mu^\mathsf{T}\Sigma^{-1}x - \frac{1}{2}\operatorname{tr}(\Sigma^{-1}xx^\mathsf{T}) - \frac{1}{2}\mu^\mathsf{T}\Sigma^{-1}\mu + \frac{1}{2}\ln(2\pi)^d|\Sigma|\right\}$$

$$= \frac{1}{(2\pi)^{d/2}|\Sigma|^{1/2}}\exp\left\{\mu^\mathsf{T}\Sigma^{-1}x - \frac{1}{2}x^\mathsf{T}\Sigma^{-1}x - \frac{1}{2}\mu^\mathsf{T}\Sigma^{-1}\mu\right\} \tag{41}$$

$$= \frac{1}{(2\pi)^{d/2}|\Sigma|^{1/2}}\exp\left\{\frac{1}{2}(x-\mu)^\mathsf{T}\Sigma^{-1}(x-\mu)\right\}.$$

For two distributions $\pi$ and $\hat{\pi}$ with $k(x) = 0$, Nielsen and Nock [53] proposed the function $I(\cdot)$:

$$I(\pi, \hat{\pi}; \alpha, \beta) = \int \pi(x)^\alpha \hat{\pi}(x)^\beta \, dx = \exp\left\{F(\alpha\theta + \beta\hat{\theta}) - \alpha F(\theta) - \beta F(\hat{\theta})\right\}$$

where the detailed derivation is as follows:

$$\int \pi(x)^\alpha \hat{\pi}(x)^\beta \, dx$$

$$= \int \exp\left\{\alpha\langle\theta, t(x)\rangle - \alpha F(\theta) + \beta\langle\hat{\theta}, t(x)\rangle - \beta F(\hat{\theta})\right\} dx$$

$$= \int \exp\left\{\langle\alpha\theta + \beta\hat{\theta}, t(x)\rangle - F(\alpha\theta + \beta\hat{\theta})\right\}\exp\left\{F(\alpha\theta + \beta\hat{\theta}) - \alpha F(\theta) - \beta F(\hat{\theta})\right\} dx$$

$$= \exp\left\{F(\alpha\theta + \beta\hat{\theta}) - \alpha F(\theta) - \beta F(\hat{\theta})\right\}\int \exp\left\{\langle\alpha\theta + \beta\hat{\theta}, t(x)\rangle - F(\alpha\theta + \beta\hat{\theta})\right\} dx$$

$$= \exp\left\{F(\alpha\theta + \beta\hat{\theta}) - \alpha F(\theta) - \beta F(\hat{\theta})\right\}.$$

### B.1 Tsallis entropy of full covariance Gaussian distributions

For $\varphi(x; q) = \frac{1}{q-1}(x^{q-1} - 1)$, the Tsallis entropy can be written as

$$\mathcal{T}_q(\pi) := -\mathbb{E}_{x\sim\pi}\varphi(x; q) = \int \pi(x)\frac{1 - \pi(x)^{q-1}}{q-1} \, dx$$

$$= \frac{1 - \int \pi(x)^q \, dx}{q-1} = \frac{1}{q-1}\left(1 - I(\pi, \pi; q, 0)\right)$$

$$= \frac{1 - \exp(F(q\theta) - qF(\theta))}{q-1}.$$

If $\pi$ is a multivariate Gaussian distribution, we have

$$F(q\theta) = \frac{q}{2}\mu^{\mathsf{T}}\Sigma^{-1}\mu + \frac{1}{2}\ln(2\pi)^d|\Sigma| - \frac{1}{2}\ln q^d.$$

Since a covariance matrix is a symmetric positive semi-definite matrix, the LDL decomposition (a variant of Cholesky decomposition) can be applied, which separates the covariance matrix into $\Sigma = L\,\mathrm{diag}\{\sigma_1^2,\ldots,\sigma_d^2\}L^{\mathsf{T}}$ where $L$ denotes a unit lower triangular matrix and $\mathrm{diag}\{\sigma_1^2,\ldots,\sigma_d^2\}$ denotes a diagonal matrix with positive entries. Then we have

$$
\begin{aligned}
F(q\theta) - qF(\theta) &= (1-q)\left\{\frac{d}{2}\ln 2\pi + \frac{1}{2}\ln|\Sigma| - \frac{d\ln q}{2(1-q)}\right\} \\
&= (1-q)\left\{\frac{d}{2}\ln 2\pi + \frac{1}{2}\ln\prod_{i=1}^{d}\sigma_i^2 - \frac{d\ln q}{2(1-q)}\right\} \\
&= (1-q)\sum_{i=1}^{d}\left\{\frac{\ln 2\pi}{2} + \ln\sigma_i - \frac{\ln q}{2(1-q)}\right\}.
\end{aligned}
$$

## B.2   Tractable Form of $\psi_\pi$

For separable $\Omega$, $\psi_\pi$ is written as [18]

$$\psi_\pi(s,a) = -f'(s,a) + \mathbb{E}_{a\sim\pi}[f'(\pi(a|s)) - \varphi(a|s)]$$

where $\varphi(x) = \frac{k}{q-1}(1 - x^{q-1})$ and accordingly $f(x) = x\varphi(x)$. For the gradient of $f(\cdot)$, we have

$$
\begin{aligned}
f'(x) &= \frac{k}{q-1}(1 - qx^{q-1}) \\
&= \frac{k}{q-1}(q - qx^{q-1} - (q-1)) \\
&= \frac{qk}{q-1}(1 - x^{q-1}) - k \\
&= q\varphi(x) - k.
\end{aligned}
$$

Taking the expectation yields Tsallis entropy as follows.

$$\mathbb{E}_{x\sim\pi}\left[-f'(x;\pi) + \varphi(x)\right] = \mathbb{E}_{x\sim\pi}\left[k - q\varphi(x) + \varphi(x)\right] = (1-q)\mathcal{T}_q^k(\pi) + k.$$

For a multivariate Gaussian distribution $\pi$, the tractable form of $\mathbb{E}_{x\sim\pi}\left[-f'(x) + \varphi(x)\right]$ can be derived by using that of Tsallis entropy $\mathcal{T}_q^k(\pi)$ of $\pi$. Thus $\psi_\pi$ can be rewritten as

$$\psi_\pi(s,a) = q\varphi(s) + (q-1)\mathcal{T}_q^k(\pi)$$

In the special case of $q = 1$ and $k = 1$, we have $\psi_\pi(s,a) = \log\pi(a|s)$.

## B.3   Bregman Divergence with Tsallis Entropy Regularization

We consider the following form of the Bregman divergence:

$$\int \pi(x)\left\{f'(\hat{\pi}(x)) - \omega(\pi(x))\right\}\mathrm{d}x - \int \hat{\pi}(x)\left\{f'(\hat{\pi}(x)) - \omega(\hat{\pi}(x))\right\}\mathrm{d}x$$

For $\omega(x) = \frac{k}{q-1}(1 - x^{q-1})$, $f'(x) = \frac{k}{q-1}(1 - qx^{q-1}) = q\omega(x) - k$, and $k = 1$, the above form is equal to

$$
\begin{aligned}
\int \pi(x)&\left[\frac{1 - q\hat{\pi}(x)^{q-1}}{q-1}\right]\mathrm{d}x - \mathcal{T}_q(\pi) - (q-1)\mathcal{T}_q(\hat{\pi}) + 1 \\
&= \frac{1}{q-1} - \frac{q}{q-1}\int \pi(x)\hat{\pi}(x)^{q-1}\,\mathrm{d}x - \mathcal{T}_q(\pi) - (q-1)\mathcal{T}_q(\hat{\pi}) + 1 \\
&= \frac{q}{q-1} - \frac{q}{q-1}\int \pi(x)\hat{\pi}(x)^{q-1}\,\mathrm{d}x - \mathcal{T}_q(\pi) - (q-1)\mathcal{T}_q(\hat{\pi}).
\end{aligned}
$$

Let us define two multivariate Gaussian distributions as follows:

$$\pi(x) = \mathcal{N}(x; \mu, \Sigma), \mu = [\mu_1, \cdots, \mu_d]^\mathsf{T}, \Sigma = L \operatorname{diag}(\sigma_1^2, \cdots, \sigma_d^2) L^\mathsf{T},$$

$$\hat{\pi}(x) = \mathcal{N}(x; \hat{\mu}, \hat{\Sigma}), \hat{\mu} = [\hat{\mu}_1, \cdots, \hat{\mu}_d]^\mathsf{T}, \hat{\Sigma} = \hat{L} \operatorname{diag}(\hat{\sigma}_1^2, \cdots, \sigma_d^2) \hat{L}^\mathsf{T},$$

where $L$ and $\hat{L}$ denote unit lower triangular matrices. We have

$$\int \pi(x) \hat{\pi}(x)^{q-1} \, \mathrm{d}x = I(\pi, \hat{\pi}; 1, q-1) = \exp\left\{ F(\theta') - F(\theta) - (q-1)F(\hat{\theta}) \right\},$$

where

$$\theta = \begin{bmatrix} \Sigma^{-1} \mu \\ -\frac{1}{2}\Sigma^{-1} \end{bmatrix}$$

$$\hat{\theta} = \begin{bmatrix} \hat{\Sigma}^{-1}\mu \\ -\frac{1}{2}\hat{\Sigma}^{-1} \end{bmatrix}$$

$$\theta' = \theta + (q-1)\hat{\theta} = \begin{bmatrix} \Sigma^{-1}\mu + (q-1)\hat{\Sigma}^{-1}\mu \\ -\frac{1}{2}(\Sigma^{-1} + (q-1)\hat{\Sigma}^{-1}) \end{bmatrix} = \begin{bmatrix} \theta_1' \\ \theta_2' \end{bmatrix}$$

and

$$F(\theta) = \frac{1}{2}\mu^\mathsf{T}\Sigma^{-1}\mu + \frac{1}{2}\ln(2\pi)^d|\Sigma| = \frac{1}{2}(\mu)^\mathsf{T}\Sigma^{-1}\mu + \sum_{i=1}^{d} \frac{\ln 2\pi}{2} + \ln \sigma_i,$$

$$F(\hat{\theta}) = \frac{1}{2}\hat{\mu}^\mathsf{T}\hat{\Sigma}^{-1}\hat{\mu} + \frac{1}{2}\ln(2\pi)^d|\hat{\Sigma}| = \frac{1}{2}(\hat{\mu})^\mathsf{T}\hat{\Sigma}^{-1}\hat{\mu} + \sum_{i=1}^{d} \frac{\ln 2\pi}{2} + \ln \hat{\sigma}_i,$$

$$F(\theta + (q-1)\hat{\theta}) = -\frac{1}{4}(\theta_1')^\mathsf{T}(\theta_2')^{-1}(\theta_1') + \frac{1}{2}\ln|-\pi(\theta_2')^{-1}|$$

### B.4  Parameterization of the full covariance matrix using the LDL decomposition

Computing the Bregman divergence for multi-variate Gaussian distributions is challenging since the derivations involve inverses, determinants, and multiplications regarding $\Sigma$. Previous approaches did not address this issue and typically enforced $\Sigma$ to be a diagonal matrix with positive entries. Motivated by Pourahmadi [54], we propose to mitigate the computations regarding a covariance matrix using the LDL decomposition $\Sigma = L \operatorname{diag}\{\sigma_1^2, \ldots, \sigma_d^2\} L^\mathsf{T}$ at the parameterization level. It enables us to implement relatively simple and numerically safe computations such as

$$\Sigma^{-1} = L^{-1} \operatorname{diag}(1/\sigma)(L^{-1})^\mathsf{T}, \tag{42}$$

$$\ln|\Sigma| = 2 \sum_{i=1}^{d} \ln \sigma_i. \tag{43}$$

where $L$ is a unit lower triangular matrix. Finding an inverse matrix of a unit triangular matrix can be computed by $\mathcal{O}(d^2)$ where the output is always a unit triangular matrix. Using the positive definiteness of $\Sigma$, the parameterization based on LDL decomposition allows a number of efficient computations for dealing with covariance matrices in practice while preserving the symmetry and the positive semi-definite matrix of $\Sigma$ on the parameterization level. We utilized these findings on implementing the full covariance Gaussian policies and regularized reward functions.

## C  Implementation Details

### C.1  Normalizing IRL rewards

Unnormalized rewards of the IRL algorithm often mislead the agent to take unnecessary awareness of *termination* in finite-horizon MDPs [12]. For this point, IRL algorithms need to remove the difference between regarding steps depending on the MDP's time. Doob's optimal stopping theorem formally states that the expected value of a martingale at a stopping time is equal to its initial expectation. Assume a martingale makes the entire procedure a fair game on average, which means nothing can be gained by stopping the play.

**Theorem 4** (Doob's optimal stopping theorem). *Let a process $\{X_t\}_{t=1}^{\infty}$ be a martingale and $\tau$ be a stopping time with respect to filtration $\{\mathcal{F}_t\}_{t \geq 1}$. Assume that one of the conditions holds:*

*(a) $\tau$ is almost surely bounded, i.e., there exists a constant $c \in \mathbb{N}$ such that $\tau \leq c$.*

*(b) $\tau$ has finite expectation and the conditional expectations of the absolute value of the martingale increments almost surely bounded, more precisely, $\mathbb{E}[\tau] < \infty$ and there exists a constant $c$ such that $\mathbb{E}\big[|X_{t+1} - X_t| \,\big|\, \mathcal{F}_t\big] \leq c$ almost surely on the event $\{\tau > t\}$ for all $t \geq 0$.*

*(c) There exists a constant $c$ such that $|X_{\min\{t,\tau\}}| \leq c$ almost surely for all $t \geq 0$. Then $X_\tau$ is an almost surely well-defined random variable and $\mathbb{E}[X_\tau] = \mathbb{E}[X_0]$.*

*Then $X_\infty$ is integrable and $\mathbb{E}[X_\infty] = \mathbb{E}[X_0]$*

Doob's optimal stopping theorem states one of the necessary conditions of IRL reward of normalizing the reward measures and making them a martingale even for finite-horizon benchmarks. In addition to the analyses of [10, 18] regarding reward shaping and normalization, mean-zero rewards for training agents have the additional property of preventing the termination awareness, as stated by the optimal stopping theorem. Therefore, we suggest normalizing with the moving mean of intermediate values of regularized rewards and updating the RL algorithm with mean-subtracted rewards.

### C.2 Transformation between $\pi_\phi$ and $\psi_\phi$

In general, the regularized reward operation $\Psi_\Omega(\Pi)$ (as well as the Bregman divergence) is intractable to be computed. However, some tractable computation methods have been discovered for specific $\Omega$ if the policy is a specific parametric model (e.g., exponential families), thanks to the aforementioned studies. In Section 6, the underlying concept in Eq. (14) is that the bidirectional transformation between $\pi_\phi$ and $\psi_\phi$ implicitly occurs via its shared network parameters $\phi$ without extra computation costs. For example, in our implementation, both $\pi_\phi$ and $\psi_\phi$ are analytically drawn in closed-form expressions using the shared parameter $\phi$ by the following methods, respectively.

**Discrete policies.** Let the agent policy for a state $s \in \mathcal{S}$ be defined as $\pi_\phi(a|s) = p(a)$ for a discrete probability distribution on the action space, typically parameterized by a softmax distribution. Since the cardinality of action space is finite in this case, we can directly compute each output according to Definition 1, i.e., $\psi_\phi(s,a) = \Omega(p) + \nabla_p\Omega(p) - \sum_{a \in \mathcal{A}}$.

**Continuous policies with the Shannon regularizer.** For both discrete and continuous policies, the following equation holds: $\psi_\phi(s,a) = \log \pi_\phi(a|s)$ (pp. 4, Jeon et al. [18]). For multivariate Gaussians, we can analytically compute the log-likelihood. As stated in Appendix B.4, we applied the LDL decomposition on the covariance matrix $\Sigma$, which is a variant of Cholesky decomposition that ensures invertibility and positive-definiteness of the covariance matrix. In our experiments, this particular parameterization usually had significantly low numerical errors, thanks to the TensorFlow linear algebra libraries specialized for variants of the LU decomposition. See the work done by Pourahmadi [54] for more details for the parameterization.

**Continuous policy with the Tsallis regularizer.** Computing the operator $\Psi_\Omega$ for an arbitrary continuous policy is usually intractable when $\Omega$ is a Tsallis entropic regularizer except when the policy is constrained to be specific parametric models. In this work, we assumed the Gaussian policy, and the analytic form of $\Psi_\Omega(\Pi)$ was initially discovered by Nielsen and Nock [53]. The entire portion of Appendix B is dedicated to derivations of $\psi_\phi$ when $\Omega$ is a Tsallis entropic regularizer.

### C.3 Network architectures

For all networks, we used 2-layer MLP with 100 hidden units. We considered the reward model with two separate neural networks $(\psi_\phi, d_\xi)$ for the proposed reward function for $\lambda \in \mathbb{R}^+$:

$$r_\phi(s,a) = \psi_\phi^\lambda(s,a) = \lambda\psi_\phi(s,a) + d_\xi(s),$$

Motivated by RAIRL-DBM, we considered the reward models in Fig. 10. The model outputs reward for proximal updates trained by mirror descent and state-only discriminator network. Discriminating state visitation by $d_\xi(\cdot)$ is required because the reward function needs to consider every state (especially the state that cannot be visited by $\pi_E$) until $D_{\mathrm{KL}}(\rho_\pi \| \rho_{\pi_E}) \approx 0$. Fig. 10 (a) shows logits of the softmax distribution involved when calculating rewards when the action space is discrete. For continuous control (Fig. 10 (b)), the architecture is similar, where the mean and covariance are used to compute a reward for a particular action.

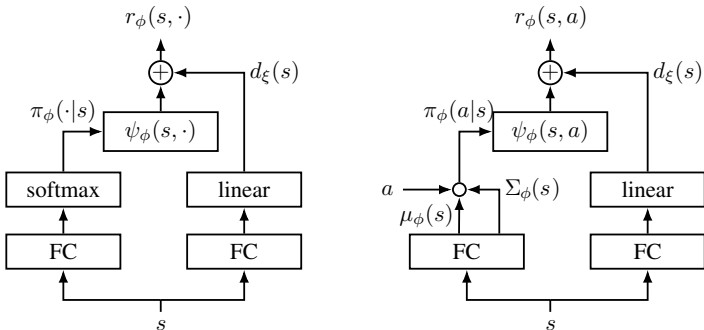

Figure 10: Schematic illustrations of MD-AIRL reward architectures for discrete (left) and continuous control (right)

## C.4 Details on imitation learning data

**The multigoal environment.** Let the 2D coordinate denote the position of a point mass on the environment. In the multigoal environment, the agent, the point mass, is initially located according to the normal distribution $\mathcal{N}(\mathbf{0}, (0.1)^2 \mathbf{I})$. The four goals are located at $(6, 0)$, $(-6, 0)$, $(0, 6)$, and $(0, -6)$, where the agent can move a maximum of 1 unit per time step for each coordinate. The ground-truth reward is given by the difference between successive values of a Gaussian mixture depicted as the contour plot in Fig. 11.

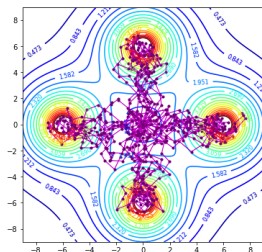

Figure 11: Visualization of the multigoal environment and expert trajectories.

**Collecting expert demonstrations.** For the multigoal environment and as well as MuJoCo benchmarks, we trained an expert policy using the SAC algorithm [55] and the demonstration data of IRL were collected from executing the trained RL expert. Only for the multigoal environment, the trajectories are post-processed to precisely capture optimal behavior (reaching each goal evenly). That is, we set the ratio of trajectories reaching each goal to exactly 25%.

## C.5 Modeling policy with full covariance Gaussian distributions

We used the full covariance Gaussian distribution in this experiment (as well as the toy experiment in Fig. 2). Note that the covariance matrix is positive-definite and symmetric. To achieve numerically stable computation, we applied LDL decomposition in Appendix B.4 to model covariance matrix using unit lower- and upper-triangle matrices, and a diagonal matrix. As a result, the policy network outputs a vector $[\boldsymbol{\mu}(s); \boldsymbol{\sigma}(s); \boldsymbol{l}(s)]^{\mathsf{T}}$ for $s \in \mathcal{S}$ where the additional vector $\boldsymbol{l}(s)$ denotes $\frac{d(d-1)}{2}$ entries of unit lower triangular matrix. Denote $\boldsymbol{L}(s)$ as a unit lower triangular matrix from $\boldsymbol{l}(s)$. For example, the covariance matrix can be reconstructed by

$$\boldsymbol{\Sigma}(s) = \boldsymbol{L}(s)[\mathrm{diag}(\boldsymbol{\sigma}(s))]\boldsymbol{L}(s)^{\mathsf{T}}.$$

In this case, the action samples can be efficiently calculated by

$$a = \boldsymbol{\mu}(s) + \boldsymbol{L}(s)(\boldsymbol{\sigma}(s) \cdot z) \quad z \sim \mathcal{N}(\mathbf{0}, \mathbf{I}) \tag{44}$$

Computing inverses, determinants and multiplications with unit triangular matrices and triangular and diagonal matrices can be efficiently performed by numerical libraries, where we used the accelerated linear algebra library from TensorFlow [56]. Therefore, we can efficiently model the Bregman divergence and reward using neural networks as provided in Appendix B. We clipped the standard deviation as $\sigma_i(s) \in [\ln 0.01, \ln 2]$ using `tanh` for the stability. In MuJoCo experiments, instead of directly using squashed policies proposed in SAC [50], we assumed the application of `tanh` as a part of the environment (known as *hyperbolized* environments of RAIRL [18]). Specifically, after an action $a$ is sampled from the policies, we passed `tanh`$(a/1.01) * 1.01$ to the environment. Then, we additionally clipped the hyperbolized actions to 1, if the given environment is not tolerant to the excessive values of action.

## C.6 Hyperparameters

Table 4: The bandit environments.

| Parameter | Value |
| --- | --- |
| Learning rate (policy) | $1 \cdot 10^{-3}$ |
| Learning rate (reward) | $1 \cdot 10^{-3}$ |
| $\eta_1$ | 2.0 |
| $\eta_T$ | 0.5 |
| $\lambda$ | 1 |
| Discount factor ($\gamma$) | 0.0 |
| Batch size | 16 |
| Steps per update | 50 |
| Total steps | 300,000 |

Table 5: The multigoal environment.

| Parameter | Value |
| --- | --- |
| Learning rate (policy) | $5 \cdot 10^{-4}$ |
| Learning rate (reward) | $5 \cdot 10^{-4}$ |
| Replay size | 10,000 |
| $\eta_1$ | 1.0 |
| $\eta_T$ | 0.1 |
| $\lambda$ | 1 |
| Discount factor ($\gamma$) | 0.5 |
| Batch size | 512 |
| Steps per update | 50 |
| Total steps | 300,000 |

Table 6: The MuJoCo environments.

| Parameter | Hopper-v3 | Walker2d-v3 | HalfCheetah-v3 | Ant-v3 |
| --- | --- | --- | --- | --- |
| Learning rate (policy) | $5 \cdot 10^{-4}$ | $5 \cdot 10^{-4}$ | $5 \cdot 10^{-4}$ | $5 \cdot 10^{-4}$ |
| Learning rate (reward) | $5 \cdot 10^{-4}$ | $5 \cdot 10^{-4}$ | $5 \cdot 10^{-4}$ | $5 \cdot 10^{-4}$ |
| Replay size | 500,000 | 500,000 | 500,000 | 500,000 |
| $\eta_1$ | 1.0 | 1.0 | 1.0 | 1.0 |
| $\eta_T$ | 0.1 | 0.1 | 0.1 | 0.05 |
| $\lambda$ | 0.01 | 0.01 | 0.01 | 0.001 |
| Discount factor ($\gamma$) | 0.99 | 0.99 | 0.99 | 0.99 |
| Batch size | 256 | 256 | 256 | 256 |
| Steps per update | 1,000 | 1,000 | 1,000 | 1,000 |
| Initial exploration | 10,000 | 10,000 | 10,000 | 10,000 |
| Total steps | 1,000,000 | 2,000,000 | 2,000,000 | 2,000,000 |

# D  Supplementary Experimental Results

Guessing the optimal choice of scheduling $\eta_t$ for a short period of time is often challenging. Tab. 7 provides extended results of the experiments depicted in Fig. 2. The table contains the performance of imitation learning varies by series of $\{\eta\}_{t=1}^{100}$ controlled by two hyperparameters $\alpha_1$ and $\alpha_T$. These results substantially helped our hyperparameter choices of step sizes in the learning of MD-AIRL reward functions in Section 7.

Table 7: Bregman divergences $D_\Omega(\pi_t \| \pi_E)$ after the final steps ($T = 100$) with different step size scheduling (10 trials with different seeds).

| $(\eta_1, \eta_T)$ | Shannon ($q = 1$) | Tsallis ($q = 1.1$) | Tsallis ($q = 1.5$) | Tsallis ($q = 2$) |
| --- | --- | --- | --- | --- |
| $(2, 2)$ | - | $1.19502 \pm 0.64091$ | $0.33996 \pm 0.26144$ | $0.68528 \pm 0.46490$ |
| $(1, 1)$ | $0.08601 \pm 0.07951$ | $0.15432 \pm 0.25206$ | $0.22232 \pm 0.31760$ | $0.11193 \pm 0.16801$ |
| $(0.5, 0.5)$ | $0.06707 \pm 0.06042$ | $0.07629 \pm 0.06931$ | $\mathbf{0.03801 \pm 0.04611}$ | $0.06056 \pm 0.05854$ |
| $(0.2, 0.2)$ | $\mathbf{0.01051 \pm 0.00920}$ | $\mathbf{0.03221 \pm 0.03239}$ | $1.21205 \pm 0.00011$ | $\mathbf{0.01805 \pm 0.01587}$ |
| $(10, 1)$ | $0.09861 \pm 0.09546$ | - | $1.09783 \pm 0.53399$ | $0.65887 \pm 0.59846$ |
| $(1, 0.1)$ | $\mathbf{0.00706 \pm 0.00863}$ | $\mathbf{0.00933 \pm 0.01089}$ | $\mathbf{0.01660 \pm 0.01152}$ | $\mathbf{0.02141 \pm 0.00899}$ |
| $(1, 0.01)$ | $0.01500 \pm 0.01510$ | $0.01109 \pm 0.01405$ | $0.02075 \pm 0.02099$ | $0.03348 \pm 0.01901$ |

The results indicate that scheduling $\eta_t$ with a harmonic progression $\eta_1 = 1$ and $\eta_T = 0.1$ shows the overall best results in this experiment. From these results and our theoretical arguments, MD is recommended to gradually lower $\eta_t$, but the rate of change has to be carefully considered, especially when $T$ is not significant. Thus, there are suitable scheduling ways of the step size $\eta_t$ in practice, depending on $\Omega$ and $T$. As a rule of thumb, we recommend setting the initial step size close to 1 and scheduling to $\eta_T \approx 0$ when there is a reasonable amount of time for training.