# OpenReview forum: "Robust Imitation via Mirror Descent Inverse Reinforcement Learning"
_NeurIPS.cc/2022/Conference — NeurIPS 2022 Accept_

### Official Review · Reviewer_4DdJ · 2022-07-07

**Rating:** 6
**Confidence:** 4
**Soundness:** 2 fair
**Presentation:** 2 fair
**Contribution:** 3 good

**Summary:**

The paper derives MD-AIRL, a new algorithm for inverse reinforcement learning as an instance of online mirror descent and proves convergence and bounded regret.
The derived algorithm is closely related to existing methods in the field of adversarial imitation learning, by using a structured discriminator akin to AIRL to estimate the expert densities (but additionally training a state-only discriminatoris to more directy match the state marginals). Furthermore, similar to RAIRL, the algorithm is tested for different Bregman divergences and policy regularizers and uses a density based model for the discriminator. The main algorithmical difference compared to prior methods, is that the discriminator does not directly specify the reward function for the given iteration, but instead is used to estimate the expert policy (which changes in every iteration, motivating the online MD formulation). The update in Eq. 14, hence, corresponds to an additional step that computes the next reward based on the expert policy estimate.
MD-AIRL is compared to RAIRL on a multi-armed bandit problem, a continous point-mass environment and Mujoco environments, and shows slightly improved performance across all settings.

**Questions:**

I only have two questions, but it would be important to answer them thoroughly in the rebuttal:

1. How crucial is the state reward (Eq. 15) for the performance of MD-IRL, how does it relate to the theory?
2. In  Eq. 14, do I understand it correctly, that the objective is minimized with respect to $\pi_{\phi}$, and the optimal policy is then mapped to a reward function using the regularized reward operator? How does this relate to the paths in Fig. 1? If we know $\pi_{t+1}$ already (because it is the policy that we used to compute $\psi$), why do we need to perform reinforcement learning (cf. Eq.4 and the equation after Eq. 15) if we know the (valid) distribution for the current reward? I understand that we need to mitigate covariance shift and take into account the auxiliary state-reward, but how does this relate to the theory?

**Limitations:**

The paper only discussed the limitation that some divergences are not covered by this formulation.

The paper would be much stronger, if it would thoroughly discuss the *mismatch between theory and algorithm* and clearly state that the *effects of estimation errors* have substantial impact in practice, but are not considered in the current work.

**Strengths And Weaknesses:**


## Originality
__Strength__
- The mirror descent formulation seems novel and leads to a slightly different algorithm that can make use of the broad theory around mirror descent.

__Weakness__
- Algorithmically, the changes compared to prior work are not overwhelming.

## Quality
__Strength__
+ The derivations seem sound and the main claims are proven with sufficient rigor.

__Weakness__
- Proofs on convergence do not consider estimation errors (for estimating log-ratios and for policy optimization)
- I think some aspect of the algorithm are not fully consistent with theory and not sufficiently well motivated:
1. The second (state-only) discriminator is motivated based on [45, 46], where the GAN setting is considered. I don't see how these paper motivate the particular use of a state-discriminator.
2. As far as I understood, the regularized reward function (Eq. 15) deviates from the theory. While I understand the motivation behind it (biasing the agent to vist expert states), it is not clear how this additional term affect the theoretical results on convergence and regret bound. I think the algorithm should also be evaluated without the state-reward

## Clarity
__Strength__
- Overall, the paper is well-structured and well-written.

__Weakness__
- Some parts are a bit hard to understand:
1. In Eq.14, it is not clear how $\pi_{\phi}$ is computed from $\psi_{\phi}$. I assume the optimization is actually performed w.r.t. a parameterized policy and then mapped to the reward function (or the policy is directly expressed in terms of $\phi$), however this should be made more explicit. Also, it is not clear how the reinforcement learning relate to the derivations; the MD update in Eq. 7 does not consider time series data.
2. I don't understand how the main change to prior work (Eq. 14) follows from the MD formulation. After the estimation of the expert densities, couldn't we directly perform the MD update by optimizing the policy with respect to Eq. 7 (using expectations over trajectories)?

## Significance
__Strength__
- The work is a solid contribution to the field by providing a mirror descent formulation for inverse reinforcement learning, along with the corresponding algorithm, which preforms reasonably well.

---

> ### Author Response · Authors · 2022-08-02
> **Official Response to Reviewer 4DdJ (1/2)**
>
> Thank you very much for your detailed feedback and time spend reviewing our work. We address your comments below.
>
> **Changes compared to prior work.**
>
> The design choice of the algorithm was intentional; we controlled the amount of the algorithmic changes compared to the RAIRL, since the primary goal of proposing MD-AIRL was to (1) demonstrate that MD in IRL settings can be readily reproduced in modern AIL implementation, and (2) clearly verify our claim by comparative experiments. Nevertheless, we still put a lot of effort into incorporating constrained optimization problem MD to the regularized IRL settings. That is, we believe the changes involve fundamental structural change, and we newly proposed our dual discriminative architecture that is well-grounded by theoretical reasoning.
>
> **About Eqs. (14) and (15).**
>
> 1.  **how $\pi_\phi$ is computed from $\psi_\phi$.**
>
>     The underlying concept in Eq. 14 is that bidirectional transformation between $\pi_\phi$ and $\psi_\phi$ happens via its shared parameters without extra computation cost in our setting. For example, in our experiments, both $\pi_\phi$ and $\psi_\phi$ are analytically drawn in a closed form expression using the shared parameter $\phi$ by the following transformation:
>
>     - **Discrete policy.** Let policy for a state $s\in\mathcal{S}$ be $\pi_\phi(a|s)=p(a)$ for a discrete probability distribution on the action space, typically parameterized by a softmax distribution. Since the number dimension of action space is finite in this case, we can directly compute the Definition 1, i.e $\psi_\phi(s, a)\Omega(p)+\nabla_p\Omega(p) -\sum_{a\in\mathcal{A}}p(a)\nabla\Omega(p)$ for arbitrary $\Omega$.
>     - **Continuous policy (Shannon).** For both discrete and continuous policies the following equation holds: $\psi_\phi(s,a) = \log \pi_\phi(a|s)$ (pp. 4, Jeon et al. 2021 [1]). For multivariate Gaussians, we can analytically compute the log-likelihood. As stated in Appendix B.4, we applied LDL decomposition on $\Sigma$, a variant of Cholelsky decomposition that guarantees invertibility and positive-definiteness of the covariance matrix. For example, in our TensorFlow 2.0 code, this part is actually implemented as
>
>         ```bash
>         import tensorflow as tf
>         import numpy as np
>
>         numact = 3 # The dimension of action space for the Hooper-v3 benchmark
>         log_denom = tf.constant(-numact * 0.5 * np.log(2 * np.pi), tf.float32)
>         perm = tf.expand_dims(tf.range(numact, dtype=tf.int32), 0)
>
>         # Log-likelihood of multivariate Gaussians
>         # https://en.wikipedia.org/wiki/Multivariate_normal_distribution
>         # Cholesky-based multivariate Gaussians:
>         #     https://arxiv.org/pdf/2102.13518.pdf
>         @tf.function
>         def logp_gaussian(x, mu, log_sigma, unit_lower_triangular):
>             return log_denom - \
>                 tf.reduce_sum(log_sigma + \
>                     .5 * tf.square(
>                         tf.linalg.matvec(
>                             tf.linalg.lu_matrix_inverse(
>                                 unit_lower_triangular, perm),
>                             x - mu)) * exp(-2 * log_sigma), -1)
>         ```
>
>         The above Python function `logp_gaussian` requires variables `mu`, `log_sigma`, and `unit_lower_triangular` that represent parametric tensor values for each data computed from a neural network, and these outputs can also be used to produce distribution at the same time since it is essentially a valid set of parameters for a Gaussian distribution. This particular parameterization usually has significantly low numerical errors thanks to the inverse function specialized for LU decompositions (`tf.linalg.lu_matrix_inverse`).
>     - **Continuous policy (Tsallis)**. Computing the operator $\Psi_\Omega$ for an arbitrary continuous policy is usually intractable when $\Omega$ is a Tsallis entropic regularizer except when the policy is constrained to be specific parametric models (e.g., exponential families). In this work, we assumed the Gaussian policy and analytic form that was initially discovered by Nielsen and Nock et al. (2011)[1]. The entire portion of Appendix B is dedicated to derivations of $\psi_\phi$ when $\Omega$ is a Tsallis entropic regularizer.
> 2. **The main change to prior works.**
>     Compared to classical MD, we were able to perform the exact MD in the regularized reward space (Definition 1). Since prior works do not fully commit derivation on the space of conditional distributions, our findings contribute to general optimization studies in sequential decision problems as well as imitation learning problems. To the best of our knowledge, our work is one of the first works to derive a practical MD-based reward function in the IRL settings.
>
> Thanks you for  your detailed comments. Following your suggestion, We carefully rewrote this part in the rebuttal revision.

---

> > ### Author Response · Authors · 2022-08-02
> > **Official Response to Reviewer 4DdJ (2/2)**
> >
> > **The mismatch between theory and algorithm.**
> >
> > This is a good point. The simple experiments in Fig. 2, and Section 7.1 is the example of our approach that works purely and does not rely on state density matching.  In generally, **we could not directly perform the MD update by optimizing the policy with respect to Eq. (7)**, because
> >
> > 1. the data collected from the expert and the agent  in the adversarial framework is not on-policy samples.
> > 2. the data does not fully cover all possible visitable states in an episode, phrased as misaligned state densities.
> >
> > We used state-density in the challenging continuous control benchmarks in Eq. (15), which is similar to $\lambda$ of the RAIRL paper. We respectfully point out that MD itself was not the main cause of this mismatch. The main technical hurdle was how to efficiently provides on-policy state samples covering all expert and agent demonstrations. This is very challenging in practice since our MD formulation in Eq. 14 works on stat densities, and usually, these densities are heavily misaligned. To overcome such issues that are caused by the nature of the off-line setting of IRL, we designed a dual discriminative architecture and utilized the state density discriminator to ensure that the realms of observation are properly aligned as fast as possible. Without matching state densities, MD-AIRL could perform slower, especially in the early phase of learning. We believe that MD-AIRL draws balance this inherits the technical advantage of AIRL in the early learning phase and convergent behavior MD-IRL that is derived from our theoretical analyses.
> >
> > **The effects of estimation errors.**
> >
> > To answer this question, we provide detailed analyses on the noisy demonstration experiments of **Table 3**.  Let the Bregman divergence between agent and expert policies be the error, and we analyze these errors by increasing the given noise level for the expert trajectories. **Table B2**. shows general tendencies of errors in RAIRL and MD-AIRL methods.
> >
> > **Table B2. Additional results on IRL with noisy demonstrations for different regularizers and noise scales (the table consists of changes of noise levels ε: 0.01 → 0.5).**
> >
> > | Settings | Bregman Div. (RAIRL) | Bregman Div. (MD-AIRL) | Average score difference (from Tab. 3) |
> > | --- | --- | --- | --- |
> > | Hoper, Shannon | 1078.19 ± 1885.02  → 1257.60 ± 2144.88  | 697.26 ± 1820.28$  → 781.15 ± 3015.54  | 33.22 → 79.57 |
> > | Hopper, Tsallis | 81.613 ± 63.769 → 84.08 ± 64.361  | 80.593 ± 63.473 → 81.356 ± 63.243 | 59.02 → 125.07 |
> > | Walker2d, Shannon | 2614.029 ± 4113.463  → 4163.818 ± 5771.339 | 2239.664 ± 2893.15  → 3224.986 ± 6439.13  | 529.82 → 801.56 |
> > | Walker2d, Tsallis | 7.142 ± 3.767  → 7.667 ± 4.076  | 6.782 ± 3.727  → 7.109 ± 3.813 | 892.16 → 658.44 |
> > | HalfCheetah, Shannon | 71.545 ± 118.263  → 229.079 ± 287.606 | 65.591 ± 107.82  → 224.707 ± 330.412  | 19.02 → 120.19 |
> > | HalfCheetah, Tsallis | 807.545 ± 1014.531  → 814.094 ± 1035.657 | 807.895 ± 1013.572  → 814.199 ± 1036.893  | 24.74 → 30.77 |
> > | Ant, Shannon | 283.718 ± 1586.597  → 775.331 ± 1915.778  | 108.17 ± 1223.28  → 221.098 ± 1652.558 | 164.55 → 506.6 |
> > | Ant, Tsallis | 6542.044 ± 7022.796 → 6548.301 ± 7026.204  | 6541.958 ± 7022.89 → 6546.35 ± 7027.377   | 345.63 → 467.15 |
> >
> > In the table, we can find an evident correlation between average Bergman divergence and performance since imitation learning converges when the divergence is 0. Therefore, this is another piece of empirical evidence for our theoretical arguments, and we greatly appreciate your suggestions. We will fully reflect these results in the final submission. Meanwhile, displaying various Bregman divergences for different tasks might be overwhelming and not always intuitive at a glance. Hence, we will also make sure to include a new figure with proper visualization of these analyses in the final submission, as authors are allowed to add one more page at that stage.
> >
> > [1] Wonseok Jeon, Chen-Yang Su,Paul Barde,Thang Doan, Derek Nowrouzezahrai, and Joelle Pineau. Regularized inverse reinforcement learning. In 9th International Conference on Learning Representations, 2021.
> >
> > [2] Frank Nielsen and Richard Nock. On Rényi and Tsallis entropies and divergences for exponential families. arXiv preprint arXiv: 1105.3259, 2011.

---

> > > ### Comment · Reviewer_4DdJ · 2022-08-03
> > > **Open Questions**
> > >
> > > Thank you very much for your reply.
> > >
> > > I could not find an answer to the following questions:
> > > 1) How does the update procedure relate to the paths in Fig. 1? Where do we start in each iteration and how does each step relate to a step in the Fig.? While I understand the mechanics of the proposed algorithm, I can not find it in the Fig. For example, where in the Fig. can I find the crucial transition from the estimated expert policy to the reward function?
> > >
> > > 2)
> > > > If we know $\pi_{t+1}$ already (because it is the policy that we used to compute $\psi$), why do we need to perform reinforcement learning (cf. Eq.4 and the equation after Eq. 15) if we know the (valid) distribution for the current reward? I understand that we need to mitigate covariance shift and take into account the auxiliary state-reward, but how does this relate to the theory?

---

> > > > ### Author Response · Authors · 2022-08-06
> > > > **Response to “Open Questions” (Discussion #1, 1/2)**
> > > >
> > > > We highly appreciate the insightful feedback and your dedication to the discussion. We also thank you for raising the questions that were accidentally missed in the official response. We provide the following answers to the questions below.
> > > >
> > > > **How does the update procedure relate to the paths in Fig. 1? … where in the Fig. can I find the crucial transition from the estimated expert policy to the reward function?**
> > > >
> > > > - **Detailed description of Fig. 1.**
> > > > Thank you for your insightful feedback. **Fig. 1** illustrates our general framework: formalizing the regularized reward space and describing MD-based reward updates. We respectfully highlight each procedure in Fig. 1 and its corresponding implementation in the actual algorithm.
> > > >     1. The current $\pi_t$ is the starting point of the optimization step at iteration $t$.
> > > >         - This corresponds to the density model of the current RL network $\pi_\theta$.
> > > >     2. The policy $\pi_t$ creates its isomorphism $\psi_t$ with the regularized operator $\Psi_\Omega$.
> > > >         - We implement this part with the reward function $\psi_\phi \in \Psi_\Omega(\Pi)$ using a dedicated neural network in the actual adversarial algorithm.
> > > >     3. The model $\psi_t$ is updated with the MD objective in Eq. (7),
> > > >         - This corresponds to learning with Eq. (14)
> > > >         - This involves the aforementioned derivations from $\pi_\phi$ to $\psi_\phi$.
> > > >     4. The trained $\psi_{t+1}$ is projected the policy space, yielding $\pi_{t+1}$.
> > > >         - According to [1], the computation of $\nabla \Omega^\ast(\cdot)$ can be performed by regularized RL procedure.
> > > >         - We implemented the regularized actor-critic algorithm to serve this purpose.
> > > > - **Revision for the final version.**
> > > >
> > > >     We fully agree with your comment that detailed computation techniques are crucial. At the same time, there is a minor concern for the presentation: the derivations themselves may not be considered as our contributions since each has previously appeared in one of the references. Therefore, we are working on making additional figures and texts in Appendix C.
> > > >
> > > >     - **For Appendix C:**
> > > >         - On page 26, we will include a new subsection titled “How $\pi_\phi$ is computed from $\psi_\phi$,” which itemizes the detailed computation methods and pseudocode for three different cases: (A) softmax, (B) Gaussian(Shannon), C. Gaussian(Tsallis)).
> > > >         - We will also include a new figure illustrating each computation method on page 26. The figure will consist of subfigures that contain detailed computation blocks for each derivation.
> > > >     - **For the main paper:**
> > > >         - In Line 251, we will include a footnote that explicitly references the new figure and the appendix we are working on.
> > > >
> > > >     **Please note:** these changes are minor and can be finished within only a few days.

---

> > > > > ### Author Response · Authors · 2022-08-06
> > > > > **Response to “Open Questions” (Discussion #1, 2/2)**
> > > > >
> > > > > **Why do we need to perform reinforcement learning if we know the (valid) distribution for the current reward? I understand that we need to mitigate covariance shift and take into account the auxiliary state-reward, but how does this relate to the theory?**
> > > > >
> > > > > Thank you for the comments. We delineate justifications for our RL/IRL procedures below.
> > > > >
> > > > > - **The necessity of Reinforcement Learning.**
> > > > > RL is a canonical method for policy updates in our MD-AIRL for the following reasons.
> > > > >     - **Equations.** We would like to clarify that the many equations in this paper (including Eq. (4), and Eq. after Eq. (15)) define their objectives and costs that inherently require the interaction with the environment by the definition of the expectation $\mathbb{E}_\pi$ (**L113**). Also, this is a restriction of sequential problems where **Eq. (7)** will be typically implemented with offline non-i.i.d data. As stated in **Lines 182-185**, we respectfully point out that the RL process plays an essential role in sequential learning by the induced discounted value measures.
> > > > >     - **Theory.** We respectfully point out our key hypothesis is the online imitation learning and temporal costs. This means the ideal case of objective in **Eq. (7)** certainly is not attainable to the agent. As a result, the temporal cost function in **Eq. (8)** is presented, which involves nonstationary estimations of the expert density and on-policy trajectory samples. Due to our theoretical setting, applying an online policy learning algorithm is the most suitable way of incorporating our theory which is unbiased to minimize the defined temporal costs in a sequential decision problem.
> > > > >     - **Algorithm.** As we mentioned, one of our aims was to verify the excellence of MD updates in the IRL domain. Therefore, the algorithm was designed to be on top of the AIL framework with discriminators and policy functions. Therefore, the trajectory sample in **Eq. (14)** does **not** cover all on-policy data required to update the policy in offline, so RL is fundamentally needed in the algorithm.
> > > > >     - Learning with an on-policy trajectory such as in RL can generally prevent compounding errors [3] in sequential and unexpected problems caused by incomplete trajectories. Consistently fitting the models with RL is important in practice because our algorithm involves neural nets with a limited number of parameters.
> > > > >
> > > > > - **How the auxiliary state-reward fits into the theory?**
> > > > >     We provide the following reasoning below.
> > > > >     1. Let us define the auxiliary temporal cost of  state reward as $h(\pi_t,\tau_t) = \sum_{i=0}^\infty d_\xi(s_i)$.
> > > > >     2. Due to the auxiliary cost, the learning objective in the actual algorithm is separated into the followings.
> > > > >         - Regularized reward function ($\psi_\phi$):
> > > > >             - minimize $f(\pi_t, \tau_t)$ where $\pi_t=\pi_\phi$.
> > > > >             - The convergence of $\psi_\phi$ is achieved at $\pi_\ast$ according to our theoretical analyses in Section 5.
> > > > >         - Policy function ($\pi_\theta$):
> > > > >             - minimize $g(\pi_t, \tau_t) = \lambda f(\pi_t, \tau_t) + h(\pi_t, \tau_t)$  where $\pi_t = \pi_\theta$
> > > > >             - This mitigates the state misalignment issues, which is useful in practice.
> > > > >     3. Suppose all of the states are visited sufficiently many times during training.
> > > > >         - Regularized reward function ($\psi_\phi$):
> > > > >             - The convergence will be found at a unique fixed point of $\Psi_\Omega(\pi_\ast)$
> > > > >         - Policy function ($\pi_\theta$):
> > > > >             - **Ideal case:** $\pi_\theta$ converges to $\pi_E$.
> > > > >             - **General case:** $\pi_\theta$ finds the most reasonable point according to $g(\pi_t, \tau_t)$ that (1) is similar to the convergence point $\pi_\ast$ derived by MD, and (2) covers most of the visited states in the trajectory data. In many benchmarks, both points are important factors of performance measures.
> > > > >             - **For both cases,** MD-AIRL has much lower variances than RAIRL in estimating the expert density since the variance induced by logistic regression for actions is decoupled and replaced with MD updates.
> > > > > - **Revision for the final version.** We will include a new paragraph on these additional justifications in Appendix C.2 on page 25 and will rename the title of Appendix C.2 as “Algorithmic Considerations.”
> > > > >
> > > > > Please let us know if there are any remaining questions.
> > > > >
> > > > > Best Regards,
> > > > >
> > > > > Authors of Submission #4477
> > > > >
> > > > > [1] Matthieu Geist, Bruno Scherrer, and Olivier Pietquin. A theory of regularized Markov decision processes. In *Proceedings of the 36th International Conference on Machine Learning*, volume 97 of *Proceedings of Machine Learning Research*, pages 2160–2169. PMLR, 2019.
> > > > >
> > > > > [2] Wenhao Yang, Xiang Li, and Zhihua Zhang. A regularized approach to sparse optimal policy in reinforcement learning. In *Advances in Neural Information Processing Systems*, pages 5938–5948, 2019.
> > > > >
> > > > > [3] Stéphane Ross, Drew Bagnell: Efficient Reductions for Imitation Learning. AISTATS 2010: 661-668.

---

> > > > > > ### Comment · Reviewer_4DdJ · 2022-08-08
> > > > > > **On the necessity of RL**
> > > > > >
> > > > > > Thank you for the reply.
> > > > > >
> > > > > > Regarding Fig. 1: From what I understand, neither of the two paths is a very clear illustration of the proposed method, since the solid line misses the "detour" of estimating the policy betwenn $\psi_t$ and $\psi_{t+1}$.
> > > > > >
> > > > > > Regarding the necessity of RL:
> > > > > > While I totally understand the practical necessity, I still do not see how the RL step is necessary due to the derivations. For Gaussian policies and Shannon-entropy regularization,  we can solve Eq. 7 and Eq. 14 in closed form, without requiring any samples. And the Max-ent optimal policy that maximizes the expected reward given by $\log \pi(a|s)$ would actually be $\pi(a|s)$ again (due to reward shaping). So at least for this setting (Gaussian, Max-Ent) the reinforcement learning only seems to become necessary due to the additional state reward, which was introduced "in the middle of the derivations" and is not derived from the original problem.

---

> > > > > > > ### Author Response · Authors · 2022-08-08
> > > > > > > **Response to “On the Necessity of RL” (Discussion #2)**
> > > > > > >
> > > > > > > Again, we thank you for your time spent reviewing and outstanding work on this discussion.
> > > > > > >
> > > > > > > **Regarding the necessity of RL:** We do not strongly argue for the necessity of RL **in theory**. Instead, we respectfully emphasize that RL is chosen **in practice** by the following points.
> > > > > > >
> > > > > > > - The derivation of **Eq. (7)** is under the assumptions that the expectation $\mathbb{E}_\pi$ is fully achieved, and all states in $\mathcal{S}$ are taken into consideration. In this case, the closed form solution exactly recovers the next MD iteration based on our theory.
> > > > > > > - However, **Eq. (14)** is quite different in practice. It indicates $\psi_\phi$ is trained with finite stochastic trajectory samples ($s\sim\bar{\tau}$) where the expert trajectories are genuinely limited. Also, since the IRL reward function is actually implemented with neural networks, numerical errors and some overestimations might accumulate in sequential decisions.
> > > > > > > - From the perspective of IRL, RL is (1) an efficient trajectory sampling method for making training trajectory samples seamless and (2) a practical on-policy algorithm that fine-tunes the IRL solution through optimizing with its value measures.
> > > > > > >
> > > > > > > **Regarding Fig. 1:** We agree that the schematic illustration of MD in **Fig. 1** does not fully depict the technical details of MD-AIRL. As we mentioned, we are working on a new figure which is a detailed illustration for **Section 6**.
> > > > > > >
> > > > > > > Overall, we will reflect your comments in the final submission of our paper to improve clarity.
> > > > > > >
> > > > > > > Best Regards,
> > > > > > >
> > > > > > > Authors

---

### Official Review · Reviewer_Mged · 2022-07-08

**Rating:** 2
**Confidence:** 2
**Soundness:** 2 fair
**Presentation:** 1 poor
**Contribution:** 1 poor

**Summary:**

The main idea of this paper is to modify adversarial inverse RL to perform mirror descent policy updates. In practice, this means that the regular RL actor objective is modified by minimizing the divergence between policy iterates. The paper proves that this method converges under reasonable assumptions (e.g., Robbins Monro learning rate), and shows that the method does work roughly on par with one prior methods (RAIRL), for a wide range of Bregman divergences.

**Questions:**

1. To what extent are the results limited to inverse RL, versus any algorithm that performs policy updates? It seems like the idea of doing regularized policy improvement could (and has been?) studied in a wide range of contexts, beyond imitation learning. To what extent are prior results *already* applicable to this setting, and to what extent do the proposed results extend to other settings (e.g., are they stronger than the results in [1]?)?
2. While it seems like the theoretical results are correct, I'm unsure if they motivate the use of mirror descent. What would similar convergence results looks like for (say) standard/Euclidean gradient descent?



[1] https://proceedings.mlr.press/v97/geist19a.html

**Ethics Review Area:**

["I don’t know"]

**Limitations:**

The limitations section is OK. The discussion of extending the analysis to f-divergences is a good point, but not really addressing the limitations of the proposed method. The discussion about "unsafe behavior" and "automation of labor" seem overly broad. I would recommend making more specific limitations about the work (e.g., arbitrary Bregman divergences might not always be useful, and the choice of divergence measure adds an additional hyperparameter for the user to tune).

**Strengths And Weaknesses:**

Strengths
* The proposed method is applicable to a wide range of Bregman divergences.
* Experiments study robustness to varying noise in the dataset and choices of hyperparameters.

Weaknesses
* Poor writing made the paper very difficult to understand. For example, when introducing the theoretical results, it'd be good to explain what question these results are answering before stating the results themselves. Similarly, when introducing the proposed method, it'd be good to explain the problem before introducing the solution (e.g., the dual discriminator is introduced without motivation).
* I'm unsure about some of the claims in the introduction and related works section (see below).
* Limited baselines for the continuous control experiments. E.g., for Fig 7/8/9, I'd recommend comparing to a method like DAC [1], or a more recent+competitive method.

**Summary**: Overall, it seems likely that mirror-descent is the right way to do imitation learning, and that it can outperform vanilla gradient descent in many settings. However, I'm not convinced that this paper effectively makes this point, partially because the writing is very hard to follow, and partially because the experiments do not convincingly show that other Bregman divergences consistently outperform L2 distance (which is effectively what gradient descent uses). So, I think the paper is on the right track, and will eventually make for a strong resubmission after significant revisions to the writing and experiments.


Additional, less important questions/comments:
* I'd recommend running a spelling + grammar checker on the paper.
* L23 -- L30 -- It seems that this paragraph is really about imitation learning, not inverse RL. Indeed, adversarial imitation learning methods do not actually recover the expert reward function, only the expert policy. So, I'm not sure that the claim that these methods "estimate the ... reward function directly" is true.
* Table 1 -- The column about "rewards" seems misleading, for two reasons. First, it's unclear of learning the reward function is actually useful, if the user only cares about acquiring the optimal agent. Second, methods like GAIL do not actually learn the expert's reward function. The column about Bregman divergences seems misleading, too, because BC can be implemented with different Bregman divergences (e.g., the standard BC corresponds to a forward KL divergence). BC also enjoys iterative solutions, and also inherits convergence guarantees from standard supervised learning theory.
* L73 "Pythagorean theorem" -- Is it true that all Bregman divergences obey the pythagoream theorem? I don't think it's true for the KL divergence. The citation [13] doesn't mention Pythagorean theorem.
* Fig 2 is very hard to read. I'd recommend making all subplots a consistent size (e.g., remove the tiny subplots in B) and making the font sizes all larger.
* Eq 14 -- I'd recommend adding a paragraph of explanation before this, explaining the intuition before stating the final objective.
* Eq 15 -- Why is the $\lambda$ parameter needed? From a probabilistic perspective, it seems like $\lambda = 1$ should be optimal
* L263 -- L265 -- What is the motivation for introducing these different Bregman divergences? The proposed method is general, in that it can use any Bregman divergence. But making the claim that this generality is useful requires evidence that different Bregman divergences are useful in different situations.
* Sec 7.1 -- I didn't understand the motivation for this experiment. It seems like the optimal policy is given by the empirical action distribution, which is trivial to compute with gradient descent or iterative updates.
* Fig 5 -- Does the RAIL baseline also decay the learning rate? If the proposed method uses a learning rate schedule, it'd be good to equip the baselines with this schedule, too.
* Fig 6, "goal entropy" -- Why is this a reasonable metric? It seems like some measure of divergence from the expert policy would be more meaningful.

[1] https://arxiv.org/abs/1809.02925

---

> ### Author Response · Authors · 2022-08-02
> **Official Response to Reviewer Mged (1/2)**
>
> We appreciate your helpful feedback and concerns. Please check **the rebuttal revision** of our paper reflecting the majority of the suggestions for improving clarity. Overall, we are working on incorporating all of your helpful comments in the final version. We address your questions below.
>
> **Experiments.**
>
> Thank you for the comment. Following your suggestions, the additional experimental results are provided in the general response (**Table B1**). We respectfully emphasize that we focused on comparative studies with RAIRL for reasons. First, the performance gains attained from MD-AIRL compared to RAIRL directly reflect the effectiveness of our MD-based reward learning schemes, as we controlled most of the algorithmic considerations. In contrast, comparison with other IRL methods is fundamentally limited due to a lack of generality in terms of choice of $\Omega$.
>
> **Questions:**
>
> 1. To what extent are the results limited to inverse RL, versus any algorithm that performs policy updates? It seems like the idea of doing regularized … and to what extent do the proposed results extend to other settings?
> This is a good point. Our approach toward the imitation learning problem is not limited to inverse RL, and our theoretical results can be applied to multiple subfields of machine learning. Also, our fundamental argument was to consider the overall imitation learning process as a combined optimization process between policy and reward functions. We strongly believe this novel perspective brought simplicity over complicated reasoning behind regularized MDPs and reward learning schemes of IRL algorithms. As a result, we proposed MD-AIRL, a robust and pratical adversarial imitation learning algorithm that is based on MD.
> 2. While it seems like the theoretical results are correct, I'm unsure if they motivate the use of mirror descent. What would similar convergence results looks like for (say) standard/Euclidean gradient descent?
> Apparently, the core motivation of our work started from an interest in geometries derived from the regularized MDPs. In machine learning and especially RL, we are familiar with the notion of learning in probability space, such as the Fisher information matrix, policy gradient algorithms, and the information geometry. These theoretical concepts imply that learning parameters in the probability distribution space might differ from convex optimization problems in the standard Euclidean space.
>
> **Limitations of the work.**
>
> We fully agree with your suggestion. We have rewritten **Section 8** to address the following limitations.
>
> - While the Bregman divergence consists of familiar divergences such as the KL divergence, using the divergence might not always be the best choice of cost function to solve a particular task. Currently, the relationship between Bregman divergence and other families of statistical divergence is actively being studied.
> - Starting from Eq. (7), we proposed an "impure" form of Eq. (15) and presented an additional hyperparameter $\lambda$. These are introduced due to a technical hurdle of how to efficiently provides on-policy samples, covering all expert and agent demonstrations. While the algorithm works well in a wide variety of benchmarks, our design choices for MD-AIRL might have some side effects, so it has been addressed as another limitation.

---

> > ### Author Response · Authors · 2022-08-02
> > **Official Response to Reviewer Mged (2/2)**
> >
> > We address other questions and comments:
> >
> > - **[L23 - L30]** We respectfully point out that this paper considers AIL algorithms as a generalized IRL method that involves reward estimation and RL based on estimated reward function. Thus, this paragraph is about a mild introduction to various IRL algorithms that consists of classical and modern IRL algorithms.  Meanwhile, we agree that AIL frameworks do not seek the ground-truth function given by a system. Therefore, we fixed the phrase as follows:
> >
> >     > *estimating the ground-truth reward function “directly,” → learning reward functions of a certain form “directly,"*
> >
> > - **[Tab. 1]** We have the following answers to the questions.
> >     1. **Rewards.** In the scope of this work, we are considering IRL reward functions in a more general sense than exactly recovering the expert’s reward function. In this sense, the discriminative signals of GAIL form a reward function since an RL algorithm can learn these signals, and the learning process recovers the expert density function.
> >     2. **BC (Bregman divergence).** To the best of our knowledge, BC and its theoretical analyses usually assume the KL divergence minimization. While it might be true that other Bregman divergences can be applied as a generalization of the algorithm, we think that the overall approach will be vastly different from the original at that point.
> >     3. **BC (Convergence).** We agree with your point. We fixed this issue and changed the label of the table (”Rate of convergence” *→ “Convergence analyses”*).
> > - **[L73]**  This is a mistake. Our original intention was to address a generalization of the Pythagorean theorem, which is indeed a much weaker condition than the Pythagorean theorem.  -Since we did not intend to make confusion to the reader, we fixed this part as follows:
> >
> >     > satisfies metric-like properties such as the Pythagorean theorem [13]. MD is closely related to… *→* satisfies metric-like properties [13, 25]. MD is also closely related to…
> >
> > - **[Fig. 2]** As suggested, we increased the sizes of fonts and figures and overhauled the arrangement of subfigures.
> > - **[Eqs. 14 & 15]** Thank you for the comments. As suggested, we rewrote the intuitive reasoning in our revised manuscript. Eq. (14) is a direct interpretation of the MD updates of Eq (7), but it has a technical drawback. Therefore, the proposed reward function of Eq (15) incorporates a discriminative signal regarding state densities, which can be adjusted with the hyperparameter $\lambda$.
> > - **[L263 - L265]** We respectfully emphasize that this particular choice of regularized function $\Omega$ in our experiments can be similarly found in previous RL/IRL works such as RAC and RAIRL algorithms. We believe that showing consistent performance gains from regularized IRL methods for various $\Omega$ is essential for verifying the generality of the proposed model.
> > - **[Sec 7. 1]** We would like to point out that the main purpose of this experiment was to measure the performance of IRL for various $\Omega$. Although the randomly generated expert policy is attainable in this particular experiment, we did not expose the expert distribution to the algorithm, a direct method with the ground-truth distribution is not possible in our setting. We clarified this point in the revision.
> >
> >     > We ﬁrst considered multiarmed bandit problems *→ To measure the performance of IRL for various $\Omega$, w*e ﬁrst considered multiarmed bandit problems
> >
> > - **[Fig. 5]** We used a fixed learning rate for the RAIRL and MD-AIRL as similarly reported by the RAIRL paper. Since our primary goal of this paper is to show the effectiveness of MD-based learning rate scheduling, the performance gain explicitly shows the effectiveness of MD. To the best of our knowledge, weight decay is not commonly practiced in AIL. Therefore, applying weight decay requires an exhaustive search and reasoning for a fair comparison. That is, the MD-AIRL algorithm might also have the potential to be more stable by applying such techniques.
> > - **[Fig. 6]**  We respectfully highlight that the goal entropy serves as an intuitive and reasonable performance metric for measuring how evenly an agent travels to multiple goals. Another point for this particular experiment is that the expert policy is a mixture of four different trained RL agents (see Appendix C.3). Computing Bregman divergence for a mixture of continuous distributions is usually intractable to compute.

---

> ### Comment · Reviewer_Mged · 2022-08-02
> **Reviewer Mged Response**
>
> Dear authors,
>
> Thank you for all of the revisions made to the paper, and for the detailed responses to my questions!
>
> My main concern was about whether the theoretical and empirical results sufficiently motivate the use of the (more general) mirror descent framework. That is, why is (Euclidean) gradient descent, a special case, insufficient? The response to question 2 suggests that optimization using mirror descent updates will be "different," but doesn't argue/prove that it will be better. Similarly, the new results in Table B1 seem to suggest that the gains from MD-AIRL are not statistically significant; GAIL (DAC rewards) seems to always be within one standard deviation of MD-AIRL.
>
> **Can the authors make the case for why the more general mirror descent framework is necessary?**

---

> > ### Author Response · Authors · 2022-08-06
> > **Response to “Reviewer Mged Response” (Discussion #1, 1/2)**
> >
> > We highly appreciate the thoughtful feedback and your dedication to the discussion.
> >
> > **Justification on MD-based IRL.** Thank you for the question. We respectfully emphasize the following points.
> >
> > - **The necessity of abstraction.** AIL is one of the most popular topics in the imitation learning domain. However, its mechanism and theoretical analyses on parametric updates are not fully revealed, mainly due to the notorious complexity of adversarial learning combined with sequential decision problems. We propose a novel theoretical framework of IRL that greatly abstracts a policy and an associated reward function into a single point in an optimization process. With this abstraction, our work exhibits a promising direction of interpretability of imitation learning performance, thanks to the rich foundation of optimization studies.
> > - **A good generalization has its own merits.** Previous works have successfully presented many divergence formulations; naturally, these concepts can be compared, distinguished, and categorized. We respectfully point out that this work applied a more general concept based on the well-formulated theory, and it is useful since theoretical or empirical improvements on this generalization can affect multiple derived models.
> > - **Theoretical approach with solid empirical evidence.** Classical IRL studies focused on the problem definition of IRL and convergence analyses with simple linear models. In contrast, newer studies of AIL focused on algorithmic designs and performance. We respectfully point out that this work uses a method of balancing these previous studies. Our empirical evidence sufficiently shows the necessity of MD, and our experiments include challenging problems such as the problem with large-scale bandits and the multi-goal environment.
> > - **Revision for the final submission.** We will carefully include these justifications by adding a new paragraph in Section 8 on page 10, as authors are allowed to add one more page at that stage.
> >
> > **Limitations of AIL.** We highly appreciate the thoughtful feedback. We would like to point out our technical issues regarding AIL. These issues stem from the typical characteristics of adversarial learning.
> >
> > - **Technical Drawbacks on AIL**
> >     - **Mode collapsing.**
> >         - (**+)** AIL shows good performance in single objective tasks such as MuJoCo locomotion benchmarks.
> >         - **(-)** Like GANs, the mode collapse phenomenon is one of the major problems in AIL when the target density is multimodal. One cause of this problem is the unstable reward signals due to limited discriminative capability.
> >         - **(-)** While GAN studies partially solved this problem using large-scale neural networks, this approach would likely be restricted in the RL/IRL setting.
> >         - **(ours)**  When the expert behaves with multiple objectives, MD-AIRL has a clear advantage since the progress of the IRL reward function is governed by constrained updates. This point is empirically verified in the multi-goal experiment.
> >     - **Non convergence.**
> >         - **(+)** AIL formulates its objective with minimax problems and presents tangible solutions when convergence is found.
> >         - **(-)** AIL does not analyze the convergence itself; consequently, it does not deal with multiple issues on unreliable/finite trajectories.
> >         - **(-)** In GAN, sophisticated normalization and regularization techniques have been proposed to solve this issue. However, this does not perfectly translate to AIL models because the imitation learning agent is restricted to certain parameterizations and learning schemes.
> >         - **(ours)** The convergence of our reward learning mechanism is guaranteed for convergence even for challenging problems.
> >     - **Inappropriate design of reward architecture.**
> >         - **(+)** AIL relies on the nonlinearity of neural networks that is flexible and scalable to represent arbitrary reward function in a certain form.
> >         - **(-)** Designing appropriate reward architecture is challenging; inappropriate architecture may result in underfitting or overfitting of the imitation learning agent.
> >         - **(-)** AIL is difficult to progress its architecture because the performance results of adversarial learning are composed of multiple factors specific to tasks.
> >         - **(ours)** The reward model has the same expressiveness as the policy. Also, most hyperparameter selection is grounded in theory.
> > - **Revision for the final submission.** We will also carefully include these justifications in Section 8. We will also carefully revise the supplementary materials by including a new appendix for detailed analyses on page 25.

---

> > > ### Author Response · Authors · 2022-08-06
> > > **Response to “Reviewer Mged Response” (Discussion #1, 2/2)**
> > >
> > > **Limitations of gradient descent for probability distributions**
> > >
> > > The Euclidean gradient descent algorithm produces an "unbiased" update rules, that is useful in updating neural network parameters in practice. More specifically, the algorithm itself can be classified as an MD algorithm where the strongly convex Euclidean ($\ell_2$) norm is the regularizer $\Omega$ and induces the metric of the parametric space. However, as we are dealing with the optimization problems in the space of probability distributions (**not** the underlying neural networks), we respectfully highlight that this metric may not be useful in measuring distance between probability distributions (**L94**). Theoretically, drawing the gradient descent algorithm in a probability space $\Delta_{\mathcal{X}}$ is _possible_ by replacing **Eq. (6)** with the divergence
> > > $$D(p,q) = \sqrt{\int_{\mathcal{X}} \vert p(x) - q(x)\vert^2 \mathrm{d} x}.$$
> > > However, it usually requires solving intractable integration, thus assuming $\Omega$ as an entropic regularizer is generally a better approach for probability densities.
> > >
> > > **Statistical significance of the additional results**
> > >
> > > The DAC reward function is known to remove some biases in GAIL rewarding mechanism and is a highly specialized method to solve MuJoCo benchmarks in terms of performance. Similarly, the GAIL algorithm with the DAC style reward function is the best performing algorithm in our comparison. Since our goal was to verify the overall robustness of the algorithm (**not**
> > >  beating the DAC algorithm by a large margin), we believe the results are meaningful and align with our theory. We respectfully emphasize that MD-AIRL shows better across all other imitation learning algorithms we have tested, and these overall results significantly strengthen our claim.
> > >
> > > Please let us know if there are remaining concerns.
> > >
> > > Best Regards,
> > >
> > > Authors of Submission #4477
> > >
> > > [1] Seyed Kamyar Seyed Ghasemipour, Richard Zemel, and Shixiang Gu. A divergence minimization perspective on imitation learning methods. In Conference on Robot Learning, pages 1259–1277. PMLR, 2020.
> > >
> > > [2] Takeru Miyato, Toshiki Kataoka, Masanori Koyama, Yuichi Yoshida, Spectral Normalization for Generative Adversarial Networks, ICLR 2018

---

> > > ### Comment · Reviewer_Mged · 2022-08-07
> > > **Reviewer response to Discussion 1/2 and 2/2**
> > >
> > > Thank you for providing the detailed response to my question! I want to examine a number of the points more closely.
> > >
> > >  > [AIL has the ] notorious complexity of adversarial learning. ... Novel theoretical framework of IRL that greatly abstracts a policy and an associated reward function into a single point in an optimization process.
> > >
> > > The second part of this claim seems to imply that MD-AIRL does not do adversarial learning. Is this correct? My understanding was that it was still doing adversarial learning, but with a certain additional regularizer applied to the policy update.
> > >
> > > > Our empirical evidence sufficiently shows the necessity of MD. ... MD-AIRL shows better across all other imitation learning algorithms we have tested
> > >
> > > I agree that statistically-significant empirical evidence that MD-AIRL outperforms all prior methods would be a strong argument. I'm a bit confused here, because I interpreted Table B1 as saying that the gains were not statistically significant. Am I mistaken in interpreting these results?
> > >
> > > > Mode collapsing.
> > >
> > > To make sure I understand, the argument is that mode collapse is caused by insufficient discriminator capacity, and that regularizing the discriminator updates might allow the discriminator to allocate it's finite capacity better? I'm a bit confused about this point, for two reasons. **First**, I don't entirely follow why regularizing the discriminator updates would make it allocate it's capacity better. Intuitively, if the network capacity is poorly allocated initially (e.g., it is highly accurate on states that don't matter for the task, but inaccurate on the important states), then regularizing the updates would make the discriminator at the next iteration behave similarly to this initially-bad discriminator. **Second**, my understanding of the method was that it regularized the policy updates, not the reward/discriminator updates (Eq. 7). It's not clear to me why regularizing the policy updates would result in regularizing the discriminator updates.
> > >
> > > > Non convergence.
> > >
> > > To make sure I understand, the concern with AIL methods is that they won't converge because of the adversarial learning dynamics? Would it be possible to elaborate on how the proposed method avoids this? My understanding was that the proposed method was still adversarial in nature, but the theoretical analysis in Section 5 doesn't seem to analyze the adversarial learning dynamics.
> > >
> > > > Inappropriate design of reward architecture. ... most hyperparameter selection is grounded in theory.
> > >
> > > To make sure I understand, the idea is that a poor choice of reward architecture will cause AIL to fail, but that the proposed method automatically suggests how to choose the reward architecture?
> > >
> > > > Limitations of gradient descent for probability distributions
> > >
> > > To clarify, the "Euclidean" update I was referring to was the unregularized policy update typically performed by GAIL/AIL/etc, not a literal Euclidean metric on probability distributions.

---

> > > > ### Author Response · Authors · 2022-08-08
> > > > **Additional Response to Reviewer Mged (Discussion #2, 1/2)**
> > > >
> > > > Again, we thank you very much for your reply and outstanding work on this discussion. We would like to answer your questions in detail; apparently, there have been some misunderstandings of our claims. We address your comments and questions below.
> > > >
> > > > **My understanding of the method was that it regularized the policy updates, not the reward/discriminator updates (Eq. 7).**
> > > >
> > > > 1. This is an important comment that has to be clearly addressed first. We clarify that **Eq. (7)** describes the **reward learning** mechanism of MD-AIRL that is connected to **Eq. (14)**.
> > > >     - In our paper, the updates in **Eq. (7)** is the learning objective for the **reward function** in $\Psi_{\Omega}(\Pi)$ as stated in **Lines 180-181.**
> > > >
> > > >         > … the MD update for the subsequent **reward function** $\psi_{t+1}=\Psi_{\Omega}(\pi_{t+1})$ is derived by solving a problem …
> > > >         >
> > > >     - Note that an alternative expression of the equivalent meaning can be used as  $\eta_t D_{\Omega^\ast}(\bar{\psi}^s_{E,t}\Vert \psi^s) + (1-\eta_t) D_{\Omega^\ast}(\psi^s_t\Vert \psi^s)$ where $\Omega^\ast$ is the Legendre-Fenchel transform of $\Omega$. We omitted this expression since it might be redundant.
> > > > 2. The function $\psi_{t+1}\in\Psi_\Omega(\Pi)$ outputs rewards that a regularized RL algorithm with the reward function $\psi_{t+1}$ converges to the next MD step.
> > > > 3. Since the regularized reward function $\psi_{t+1}$ typically trained via offline trajectory data in practice (IRL), the current policy $\pi_t$ is trained by an RL algorithm with sufficient environmental interactions, yielding a policy of the next MD step $\pi_{t+1}$. Fig. 1 illustrates this entire process.
> > > >
> > > > **The claim seems to imply that MD-AIRL does not do adversarial learning. Is this correct?**
> > > >
> > > > Thank you for the question. To put it simply, MD-AIRL **does do** adversarial learning, but the non-convergence of adversarial learning **does not** affect the convergence of MD. To explain this statement, we would like to highlight that the theoretical framework of MD-AIRL does not specify the estimation algorithm as stated in **Lines 160-163**.
> > > >
> > > > > That is, it is fundamentally uncertain to model global objectives which are not attainable for both RL and IRL. Instead, we hypothesize on existence of a random process where each estimation $\bar{\pi}_{E,t}$ resides in a closed, convex neighborhood of $\pi_E$, generated by an arbitrary estimation algorithm.
> > > > >
> > > >
> > > > Therefore, we respectfully emphasize the following points.
> > > >
> > > > 1. **Agnostic to the estimation process.** Since the online learning hypothesis only requires an erroneous estimation target of $\pi_E$ at each iteration, the estimation process can be arbitrary, e.g., the maximum likelihood estimation algorithm in the toy experiment in **Fig. 2**. In the other experiments, we usually consider the logistic regression of AIL discriminators as an instance of such processes, mainly due to its great popularity in the IRL domain and the fairness of comparison experiments.
> > > > 2. **Learning a specific action (mostly) depends on MD.** In MD-AIRL, we have two neural networks trained by logistic regression of adversarial learning: $\pi_\nu$ and $d_\xi$. For $\pi_\nu$, its only usage is in the MD updates of $\psi_\phi$ in Eq. (14), so $\pi_\nu$ does not affect the actual learning of the policy updates of RL. For $d_\xi$, it reduces the covariance shift on *states* when state visitation is misaligned. Therefore, adversarial learning only has indirect effects on $\pi_\theta$.
> > > >
> > > > **The concern with AIL methods is that they won't converge because of the adversarial learning dynamics? Would it be possible to elaborate on how the proposed method avoids this?**
> > > >
> > > > Yes, this is related to the general arguments in **Lines 185-187**. As illustrated in **Fig. 3**, we claimed that $\pi_t$ finds a certain form of convergence $\pi_\ast$ even if the estimation process $\bar{\pi}_{E,t}$ is not convergent (depicted as the red region). This is because the constrained IRL updates are scheduled with the step size conditions in **Eqs (9)** and **(10)**, which are based on our theoretical analyses. Therefore, we respectfully point out that the estimation errors of adversarial learning’s logistic regression ($D_\nu$ in the algorithm) are clearly decoupled in our anlayses by learning a MD-based reward function ($\psi_\phi$ in the algorithm).
> > > >
> > > > **To make sure I understand, the idea is that a poor choice of reward architecture will cause AIL to fail, but that the proposed method automatically suggests how to choose the reward architecture?**
> > > >
> > > > Yes. The reward architecture of $\psi_\phi\in\Psi_\phi(\Pi)$ is determined by the parameterization of the policy $\Pi$ (such as softmax or Gaussian), so learning actions is inherently optimized to train a policy in $\Pi$. The state reward of $d_\xi(s)$ is auxiliary and only delivers gradual effects on learning to facilitate densities to be matched. Please see **Fig. 9** for the architectures.

---

> > > > > ### Author Response · Authors · 2022-08-08
> > > > > **Additional Response to Reviewer Mged (Discussion #2, 2/2)**
> > > > >
> > > > > **I don't entirely follow why regularizing the discriminator updates would make it allocate it's capacity better.**
> > > > >
> > > > > The argument “a regularized reward function **trained with MD** shows better allocation of its capacity” can be better understood in the context of classical GAN studies for dealing with mode collapsing, especially the **historical averaging** technique proposed by Salimans et al. [1]: a model deviating from its time average is penalized for improving stability.
> > > > >
> > > > > We would like to explain our claim by providing the following a simple intuitive example:
> > > > >
> > > > > - For a iteration $t\in\mathbb{N}$, let $\pi_t\in\Pi$ be a stateless agent Gaussians.
> > > > > - Let the expert policy $\pi_E$ be a mixture of $K$-Gaussians i.e. $\pi_E = \frac{1}{K}\sum_{k=1}^K\pi_{E}^{(k)}$ where $\pi_E^{(k)} = \mathcal{N}(\mu^{(k)}, \Sigma^{(k)})$. Suppoe an estimation process $\bar{\pi}_{E,t} \in \Pi$ does not coverges due to its limited capacity and largely perturbates among local solutions.
> > > > > - Suppose $\bar{\pi}_{E,t} = \tilde{\pi}_E^{(\kappa_t)}$ where $D_\Omega(\bar{\pi}_E^{(k)}\Vert\pi_E^{(k)}) \approx 0$ for $k\in\{1,\dots, K\}$.
> > > > > - Following our analyses in **Section 5**, the distribution $\pi_t$ trained with MD will converge to $\pi_\ast\in\Pi$ such that $\nabla\Omega(\pi_\ast) = \lim_{t\to\infty}\mathbb{E}[\nabla\Omega(\bar{\pi}_{E,t})]$.
> > > > > - Since $\nabla\Omega$ forms an isomorphism, we claim that $\pi_t$ will covers most of modes that the estimation process of $\bar{\pi}_{E,t}$ produces in a long time series (e.g. by having a relatively high entropy), which is similar to the underlying idea of **historical averaging** techniques by Salimans et al. [1].
> > > > >
> > > > > We respectively highlight that one of important theoretical results in **Section 5** is that the regret defined in **Eq. (11)** is bounded by $\mathcal{O}(1/T)$. Therefore, our claim is that our MD updates optimally allocated its limited capacity, embracing discriminator's error of adversarial learning, which is also verified in our experiments.
> > > > >
> > > > > **I'm a bit confused here, because I interpreted Table B1 as saying that the gains were not statistically significant.**
> > > > >
> > > > > We respectively highlight the following points.
> > > > >
> > > > > - For the following reasons, the standard deviations in Table B1 might appear to be relatively high compared to other popular metrics (losses) in machine learning.
> > > > >     1. In the MuJoCo locomotion tasks, the scores indicate the physical distance a robot travels by moving its joint over a long period (≤ 1000 steps). Each environment has a stopping condition, so the trajectory length can be inconsistent.
> > > > >     2. We used stochastic policies for all algorithms. The actions are stochastically sampled for each step, even for evaluation.
> > > > >
> > > > >     Therefore, we would like to point out that consistently showing better performance than various AIL algorithms on average is still pleasing results that align with our claim.
> > > > >
> > > > > - Since RAIRL is the most compatible algorithm in terms of IRL architecture and regularization choice, the extensive comparison experimental results with RAIRL for various $\Omega$ in the main paper should be taken into acount for the statistical sigfnificance.
> > > > >
> > > > > **The "Euclidean" update I was referring to was the unregularized policy update typically performed by GAIL/AIL/etc.**
> > > > >
> > > > > Thank you for the detailed comment. We now understand this point more clearly. As you noticed, there were two levels of “optimization” problem during the discussion. Therefore, we would like to summarize our method on both perspectives as below.
> > > > >
> > > > > - **Optimization for “high-level” probability distributions: the main contributions are mostly in this level.**
> > > > >     - AIL models can be seen as solving *unconstrained* optimization for its discriminator. We claimed that it does not deal with some challenging situations in imtation learning.
> > > > >     - We propose MD-AIRL, which performs *constrained* MD updates utilize the adversarial learning as a close, but unrelaible estimation process of the target densities.
> > > > > - **Optimization for “low-level” neural networks. MD-AIRL also uses (Euclidean) gradients with an optimizer (ADAM) in its training. We do not specifically argue for or against training using gradient-based learning agorithm in neural networks.**
> > > > >     - **MD-AIRL.** The neural network optimizes along with Euclidean gradients using an optimizer, but the proximal local target of learning is the MD-updates in the high level of probabilistic perspective, which makes the policy learning more robust.
> > > > >     - **AIL.** Even if the policy is updated a small learning rate, we claim that optimizing AIL rewards might have practical issues, and these signals can be unreliable depending on the problems.
> > > > >
> > > > > [1] Tim Salimans, Ian J. Goodfellow, Wojciech Zaremba, Vicki Cheung, Alec Radford, and Xi Chen.
> > > > > Improved techniques for training GANs. *CoRR*, abs/1606.03498, 2016. URL http://arxiv.
> > > > > org/abs/1606.03498.

---

> > > > > ### Comment · Reviewer_Mged · 2022-08-08
> > > > > **Reviewer response to Discussion #2 (1/2 and 2/2)**
> > > > >
> > > > > Thanks to the authors for continuing the discussion! This is useful for helping me get an even better understanding of the paper and its contributions.
> > > > >
> > > > > > My understanding of the method was that it regularized the policy updates, not the reward/discriminator updates (Eq. 7).
> > > > >
> > > > > I'm still a bit confused on this point, and would recommend revising the paper to clarify this. Based on this discussion, my rough understanding is that $D_\Omega(\pi \| \pi')$ is defined as a divergence between $r_\pi$ and $r_{\pi'}$. Is this correct? Part of the confusion here stems from the fact that $D_\Omega$ is defined on L106 without reference to a reward function, and the reward function $r$ defined in the preliminaries does not appear in either Eq 7 or 14.
> > > > >
> > > > > > The claim seems to imply that MD-AIRL does not do adversarial learning.
> > > > >
> > > > > OK, I think this makes sense: it's OK that the discriminator learning is noisy because MD-AIRL is an online learning algorithm, and hence can handle noise in the "labels." Are there guarantees that AIL fails to be a "good" online learning algorithm in this same sense?
> > > > >
> > > > > > The concern with AIL methods is that they won't converge because of the adversarial learning dynamics? Would it be possible to elaborate on how the proposed method avoids this?
> > > > >
> > > > > I'm still a bit confused here. The argument that MD-AIRL converges because of decreasing step sizes seems a bit odd because it would be straightforward to apply Robbins-Monro step sizes to AIL. If I understand correctly, this same line of reasoning would apply to any machine learning paper that applies SGD with a fixed step size. Yes, the decreasing step sizes are needed to guarantee convergence, but this seems like a known fact about SGD, rather than a surprising, emergent property of mirror descent.
> > > > >
> > > > > > To make sure I understand, the idea is that a poor choice of reward architecture will cause AIL to fail, but that the proposed method automatically suggests how to choose the reward architecture?
> > > > >
> > > > > Where are the architectures discussed on page 9? The word "architecture" only seems to appear on page 6, but the architectures aren't defined there.
> > > > >
> > > > > > I don't entirely follow why regularizing the discriminator updates would make it allocate it's capacity better.
> > > > >
> > > > > To make sure I understand, the argument here is that the regularized discriminator updates are similar to historical averaging? I'm not sure I fully understand this argument. I agree that historical averaging is necessary to guarantee convergence in two-player games, but it seems like historical averaging could be applied to either standard AIL updates or MD-AIRL updates.
> > > > >
> > > > > > I'm a bit confused here, because I interpreted Table B1 as saying that the gains were not statistically significant.
> > > > >
> > > > > Quick clarification about the "stochastic policies" -- the standard deviation reported in these tables is `StdDev([AverageReturn(pi) for pi in pi_each_seed_list])`, not `StdDev(Return(pi))`, right? I.e., this is measuring the variance across training runs, not the variance across different rollouts of the same policy.
> > > > >
> > > > > > there were two levels of “optimization” problem during the discussion
> > > > >
> > > > > Thanks for clarifying this point! I agree that algorithms can be analyzed at different levels of analysis, and that improving our understanding of any level of analysis can be useful.

---

> > > > > > ### Author Response · Authors · 2022-08-09
> > > > > > **Response to Reviewer Mged (Discussion #3, 1/2)**
> > > > > >
> > > > > > As authors, we really appreciate your time and effort spent reviewing, especially your excellent participation during this discussion period. This has been a genuinely great experience that we have been able to discuss this submission actively and address many of the reviewers’ concerns. We will improve our manuscript based the discussion and your comments for the final submission.
> > > > > >
> > > > > > **My rough understanding is that $D_\Omega(\pi\Vert\pi^\prime)$ is defined as a divergence between between $\psi_\pi$ ($r_\pi$) and $\psi_{\pi^\prime}$ ($r_{\pi^\prime}$). Is this correct?**
> > > > > >
> > > > > > Yes, this is correct. The core idea stems from **Lemma 1**, that a stochastic policy function $\pi\in\Pi$ and an associated reward function $\psi_\pi \coloneqq \Psi_\Omega(\pi)$ (from **Definition 1**) have an “one-to-one relationship” (the regularizer $\Omega$ determines such correspondence). In our setting, the policy space and the reward space are connected and jointly optimized to solve the imitation learning problem. This is a novel perspective in the IRL domain.
> > > > > >
> > > > > > - From this perspective, two mappings are presented in **Fig. 1:** $\Psi_\Omega$ and $\nabla\Omega^\ast$.
> > > > > > - Similarly, $\pi_\phi$ can be used in **Eq. (14)** since the regularized reward function $\psi_\phi$ is analytically drawn in a closed form expression using the shared parameter $\phi$.
> > > > > > - Inspired by previous studies of regularized IRL, the detailed computation methods for the mappings are implemented and addressed in the **appendices** and [Official Response to Reviewer 4DdJ (1/2)](https://openreview.net/forum?id=huT1G2dtSr&noteId=BeVQi2AXyQs).
> > > > > >
> > > > > > Consequently, let us suppose that there are two pairs of functions $(\pi, \psi_{\pi})$ and $(\pi^\prime, \psi_{\pi^\prime})$. We respectfully point out that the two Bregman divergences $D_\Omega(\pi\Vert\pi^\prime)$ and $D_{\Omega^\ast}(\psi_{\pi^\prime}\Vert\psi_\pi)$ are equivalent and can be simply proved for strongly convex $\Omega$. In the paper, we mostly focus on the use of Bregman divergence in the (primal) policy space ($D_\Omega(\cdot \Vert \cdot)$) due to its simplicity for derivations and computations.
> > > > > >
> > > > > > **Are there guarantees that AIL fails to be a "good" online learning algorithm in this same sense?**
> > > > > >
> > > > > > In **Lines 31-45**, we pointed out the two drawbacks of AIL based on our observations:
> > > > > >
> > > > > > > **L37.** We claim that there are two issues leading to unconstrained policy updates: (1) a statistical divergence often cannot be accurately obtained for challenging problems and (2) an immediate divergence between agent and expert densities does not guarantee unbiased learning directions.
> > > > > > >
> > > > > >
> > > > > > As a result, we were able to bring a novel online learning algorithm that utilizes AIL and improves AIL’s overall robustness for imitation learning. Based on the minimax formulation of AIL and our experimental results, it is likely that there are certain situations in that AIL might fail in terms of regrets, and we pointed out three technical issues of AIL during the discussion (mode collapsing, non-convergence, inappropriate architecture). We respectfully remind you that a complete theoretical understanding of the AIL is beyond the scope of our paper and remains as future work.
> > > > > >
> > > > > > **If I understand correctly, this same line of reasoning would apply to any ML paper that applies SGD with a fixed step size.**
> > > > > >
> > > > > > We respectfully point out that adversarial learning and inverse reinforcement learning would be regarded as exceptions to this statement. For both problems, the uncertainty of the intermediate models (discriminators and IRL reward functions) governs and hinders the convergence of primary learning models (generators and policies) and vice versa. To solve these issues, we applied the MD-based algorithm to improve the robustness of an IRL reward function, and it brought pleasing outcomes to the policy optimization in terms of imitation learning performance.
> > > > > >
> > > > > > **I'm still a bit confused here. … but this seems like a known fact about SGD, rather than a surprising, emergent property of mirror descent.**
> > > > > >
> > > > > > We agree since SGD can be considered as an instance of online MD algorithms (**L96**). However, we respectfully point out that theoretical contributions of our work can be better evlauted as an imitation learning study. Our analyses are useful for generalizing statistical divergences that is actively used in the RL and IRL domains. At the same time, our approach have multiple contributions in IRL related problems, such as:
> > > > > >
> > > > > > - we proposed the fundamental relationship between policy and reward function in the imitation learning problem.
> > > > > > - we proposed the RL/IRL methodology as a combined optimization process that theoretically guarantees the robustness of learning in sequential decisions.
> > > > > > - we proposed a novel formalization of the online imitation learning problem and a practical algorithm to solve this problem.

---

> > > > > > > ### Author Response · Authors · 2022-08-09
> > > > > > > **Response to Reviewer Mged (Discussion #3, 2/2)**
> > > > > > >
> > > > > > > … **but it seems like historical averaging could be applied to either standard AIL updates or MD-AIRL updates**
> > > > > > >
> > > > > > > We respectfully point out that MD works much more meaningfully in policy optimization. For example, suppose a parameterized Gaussian policy $\pi_\phi$ and its neural network function $f_\phi$ that outputs parameters of a multivariage Gaussian policy i.e, $f_\phi(s) = (\mu_\phi(s), \Sigma_\phi(s))$ , such that $a \sim \mathcal{N}(\mu_\phi(s), \Sigma_\phi(s))$. Suppose $\phi_t$ as the neural network parameters of $f_\phi$ at the iteration $t$.
> > > > > > >
> > > > > > > - From the example in Discussion #2, we assumed that the expression $\lim_{t\to\infty}\mathbb{E}[\nabla\Omega(\pi_{\phi_t}(\cdot\vert s))]$ resembles a mapping of Guassian mixture (by consistently changing its mode).
> > > > > > > - **Historical averaging on $(\phi_t)_{t=1}^\infty$.**  The technique favors the “average” of historical neural network parameters. It stabilizes the learning process, but analyzing its meaning in the probabilistic space is difficult.
> > > > > > > - **MD of a policy $\pi_\theta$ for the target sequence $(\pi_{\phi_t})^\infty_{t=1}$.** The MD algorithm finds the best policy $\pi_\theta$ that is closest to $\nabla\Omega^\ast(\lim_{t\to\infty}\mathbb{E}[\nabla\Omega(\pi_{\phi_t}(\cdot\vert s))])$, where the Bregman divergence is used measuring distance, e.g. forward KL divergence when $\Omega$ is the negative Shannon entropy. Therefore, we claim that MD brings pleasing outcomes regarding statistical divergence induced by $\Omega$.
> > > > > > >
> > > > > > > **Where are the architectures discussed on page 9? The word "architecture" only seems to appear on page 6, but the architectures aren't defined there.**
> > > > > > >
> > > > > > > We respectfully remind you that we referred to **Fig. 9** on page 25 in the appendix (Supplementary Material: [pdf](https://openreview.net/attachment?id=huT1G2dtSr&name=supplementary_material)), which illustrates detailed reward architecture of **Eq. (15)**.
> > > > > > >
> > > > > > > **Quick clarification about the "stochastic policies" -- the standard deviation reported in these tables is** `StdDev([AverageReturn(pi) for pi in pi_each_seed_list])`,not `StdDev(Return(pi))`**, right?**
> > > > > > >
> > > > > > > Specifically, **we report the results scores on the last $10^5$ steps with five different seeds.** Therefore, the standard deviation is calculated from `StdDev([[AverageReturn(pi, i) for pi in pi_each_seed_list] for i in last_1e5_steps])` with `last_1e5_steps = [0, 5000, ..., 100000]`.
> > > > > > >
> > > > > > > `AverageReturn(pi, i)` indicates the average return score (five episodes) of `pi` at the i-th step starting from the final step. Compared to deterministic policies, the environmental scores of a stochastic policy tend to have high variance by its nature.
> > > > > > >
> > > > > > > **Thanks for clarifying this point! I agree that algorithms can be analyzed at different levels of analysis, and that improving our understanding of any level of analysis can be useful.**
> > > > > > >
> > > > > > > We are very glad that this important concern is clarified.
> > > > > > >
> > > > > > > Since this probably could be the final response considering the situation that the discussion period ends soon, we would like to thank you once again for the constructive feedback and discussion. Overall, we will reflect your comments in the final submission of our paper to improve clarity.
> > > > > > >
> > > > > > > Sincerely yours,
> > > > > > >
> > > > > > > Authors.

---

> > > > > > > > ### Comment · Reviewer_Mged · 2022-08-09
> > > > > > > > **Reviewer response to Discussion #1 (1/2 and 2/2)**
> > > > > > > >
> > > > > > > > Thanks for continuing the discussion! This has been helpful for me to get an even better understanding of the paper. Thanks for responding to all of the questions. Unless mentioned below, the answers have resolved my concerns.
> > > > > > > >
> > > > > > > > > Are there guarantees that AIL fails to be a "good" online learning algorithm in this same sense?
> > > > > > > >
> > > > > > > > I think I might not be asking this question in the same way. At a high-level, the theoretical results of the paper seem to say that MD-AIRL has certain nice properties. The question is whether AIL (i.e., MD-AIRL with an L2 divergence) has those same nice properties. For example, does the analysis fail for the L2 divergence? Or, does MD-AIRL have faster convergence rates than AIL? Said in other words, does the theory give us any reason to prefer MD-AIRL over AIL?
> > > > > > > >
> > > > > > > > > architectures
> > > > > > > >
> > > > > > > > Thanks for the pointer! I had somehow missed this in reading the paper. Do the baselines use the same architectures?
> > > > > > > >
> > > > > > > > > standard deviations
> > > > > > > >
> > > > > > > > I'm not sure `StdDev([[AverageReturn(pi, i) for pi in pi_each_seed_list] for i in last_1e5_steps])` is correct. Written in terms of checkpoints, I think it should be `StdDev([Average([AverageReturn(pi, i) for i in last_1e5_steps]) for pi in pi_each_seed_list])`.

---

> > > > > > > > > ### Author Response · Authors · 2022-08-09
> > > > > > > > > **Response to Reviewer Mged (Discussion #4)**
> > > > > > > > >
> > > > > > > > > We are very grateful that our response cleared many of your concerns. We will do our best to answer further questions until the end of the discussion period.
> > > > > > > > >
> > > > > > > > > **The question is whether AIL (i.e., MD-AIRL with an L2 divergence) has those same nice properties. For example, does the analysis fail for the L2 divergence? Or, does MD-AIRL have faster convergence rates than AIL? Said in other words, does the theory give us any reason to prefer MD-AIRL over AIL?**
> > > > > > > > >
> > > > > > > > > Thank you for the insightful question. We now understand your point better. We would like to answer this question and explain our claims in more detail by following points.
> > > > > > > > >
> > > > > > > > > 1. **MD-AIRL with an L2 divergence.** MD-AIRL with the L2 divergence is possible, and the analyses also hold in the Euclidean space. As we mentioned in Discussion #1, it is difficult to estimate L2 divergence.
> > > > > > > > > 2. **AIL with a good learning rate scheduling mechanism (both a policy and a discriminator).**
> > > > > > > > >
> > > > > > > > >     We think this is what you are referring to. We think this is a good idea, but there are a few concerns.
> > > > > > > > >
> > > > > > > > >     - **Discriminator**
> > > > > > > > >         - **Convergence.** The neural network parameters will converge to a certain point.
> > > > > > > > >     - **Policy**
> > > > > > > > >         - **Convergence.** The neural network parameters will converge to a certain point since the discriminator converges. The policy $\pi_\theta$ finds a most suitable parameter $\theta^\ast$.
> > > > > > > > >         - **Is $\pi_{\theta^\ast}$ similar to $\pi_E$?** While $\theta^\ast$ would be the best solution that incorporates the historical sequence of $\theta_t$ where $t\in[1,\infty)$ (according to our discussion), the relationship between $\pi_{\theta^\ast}$ and $\pi_E$ is unknown due to the nonlinearity of neural networks. The Euclidean distance $\lVert \theta_1-\theta_2\rVert_2$ does not proportional to $\lVert \pi_{\theta_1} - \pi_{\theta_2}\rVert_2$ (the matrix norms induced by the vector 2-norm).
> > > > > > > > > 3. **MD-AIRL with statistical divergences (the proposed model).**
> > > > > > > > >     - **Policy**
> > > > > > > > >         - **Convergence.** The neural network parameters will converge to a certain point since the IRL reward function converges. We claimed that the policy $\pi_\theta\in\Pi$ finds a most sutiable parameter $\pi_\ast \in \Pi$.
> > > > > > > > >         - **Is $\pi_{\ast}$ similar to $\pi_E$?**  We claimed that $\pi_\ast$ would be similar to $\pi_E$ (or the best solution in $\Pi$) in terms of the given Bregman divergence.
> > > > > > > > > 4. **Reason to prefer MD-AIRL over AIL?**
> > > > > > > > >     - $\pi_E\in\Pi$ **and the data are sufficent.** With this ideal situation, both algorithms are preferred.
> > > > > > > > >     - **The expert parameterization is unknown, or the data are insufficient.** Our theoretical analyses are essential to solving this real-world problem. We claimed that MD-AIRL is suitable because (1) it is convergent and (2) the convergence guarantees the best imitation learning performance for $\Pi$ under the regularized MDP assumption.
> > > > > > > > >
> > > > > > > > > **Do the baselines use the same architectures?**
> > > > > > > > >
> > > > > > > > > Yes, the RAIRL-DBM model [1] shares the same reward architecture of $\psi_\phi$, and it also uses an additional neural network; hence, the numbers of neural network parameters are also equal (also, BC does not have a reward architecture). This architecture setting was intentional, hence the difference of the results in experiments are caused by different learning mechanisms.
> > > > > > > > >
> > > > > > > > > **Standard deviation.**
> > > > > > > > >
> > > > > > > > > Thank you for the instructive comment. We are pretty sure that the evaluation is correct; we respectively introduce our intention on the evaluation for `StdDev([[AverageReturn(pi, i) for pi in pi_each_seed_list] for i in last_1e5_steps])`. Let us consider the evaluation matrix:
> > > > > > > > > |  | pi_1 | pi_2  | … | pi_5 |
> > > > > > > > > | --- | --- | --- | --- | --- |
> > > > > > > > > | i=0 | AvgRet(pi_1, 0) | AvgRet(pi_2, 0) |  | AvgRet(pi_1, 0) |
> > > > > > > > > | i=5000 | AvgRet(pi_1, 5000) | AvgRet(pi_2, 5000) |  | AvgRet(pi_2, 0) |
> > > > > > > > > | … | … |  |  | … |
> > > > > > > > > | i=100000 | AvgRet(pi_1, 100000) | AvgRet(pi_2, 100000) | … | AvgRet(pi_5, 100000) |
> > > > > > > > >
> > > > > > > > > We believe that each element of the table can be considered as an independent trial since **(1)** the numbers are evaluated at the end phase of learning **(2)** there is no correlation between pi trained with different random seeds, and **(3)** the considerable amount steps (5000) was put between every evaluation.
> > > > > > > > >
> > > > > > > > > Again, thank you for your great participation in this discussion.
> > > > > > > > >
> > > > > > > > > Sincerely,
> > > > > > > > >
> > > > > > > > > Authors.
> > > > > > > > >
> > > > > > > > > [1] Wonseok Jeon, Chen-Yang Su,Paul Barde,Thang Doan, Derek Nowrouzezahrai, and Joelle Pineau. Regularized inverse reinforcement learning. In 9th International Conference on Learning Representations, 2021.

---

> > > > > > > > > > ### Comment · Reviewer_Mged · 2022-08-09
> > > > > > > > > > **Reviewer response**
> > > > > > > > > >
> > > > > > > > > > Thanks for the clarifications. I will take these into account when discussing the paper with the other reviewers. I don't have any additional questions at this point.

---

> > > > > > > > > > > ### Author Response · Authors · 2022-08-09
> > > > > > > > > > > **Response to Reviewer Mged (after Discussion)**
> > > > > > > > > > >
> > > > > > > > > > > Thank you again for your insightful comments and discussion.
> > > > > > > > > > >
> > > > > > > > > > >
> > > > > > > > > > > Sincerely, Authors.

---

### Official Review · Reviewer_mFTz · 2022-07-11

**Rating:** 6
**Confidence:** 2
**Soundness:** 3 good
**Presentation:** 3 good
**Contribution:** 3 good

**Summary:**

Inspired by mirror descent, the paper proposes MD-IRL, which considers the reward function as an iterative sequence in a proximal method. The paper proves that such a method ensures robust minimization of a Bregman divergence under the mirror descent framework.


**Questions:**

See weaknesses.

**Limitations:**

Yes.

**Strengths And Weaknesses:**

Strengths:
 - It is novel to consider using MD in the reward update. Rigorous regret upper bound is also provided to show that the Bregman divergence is minimized to a local optimum. In the meanwhile, stepsize considerations are also provided, which is good.
 - Thorough empirical studies are provided (though limited, see weaknesses).

Weaknesses:
 - The paper is closely related to RAIRL. I think the authors should make a thorough comparison with RAIRL.
 - In the empirical study, the paper only compares with RAIRL, which is slightly limited.

---

> ### Author Response · Authors · 2022-08-02
> **Official Response to Reviewer mFTz**
>
> We sincerely thank you for your positive feedback and overall support of our submission. We address your comments below.
>
> **Comparison with RAIRL.**
>
> Thank you for the comment. Our theoretical and empirical achievements are built upon the novel perspective of considering iterative RL and IRL algorithms as a combined optimization process with dual aspects. Both RAIRL and MD-AIRL are highly generalized algorithms in terms of multiple options of regularizer $\Omega$. Compared to RAIRL, our work brings more beneficial results in realistic situations with limited training time and unreliable data. Our work is also more aligned with early theoretical IRL studies providing reward learning schemes and convergence guarantees. Most of this explanation can be found in Sections 1 & 8.
>
> **Comparison with other approaches.**
>
> Thank you for the comment. Following your suggestion, we have conduced additional experiments, that is provided in the general response (Table B1). Our primary aim was to demonstrate the excellence of MD-AIRL in various environments, regularizers, and other realistic situations. RAIRL is a competitive AIL algorithm that is highly scalable and general in this sense; thus, it was a good counterpart of our comparative studies, illuminating our claim of two issues in current AIL methods (Section 1). We respectfully emphasize that other IRL algorithms lack compatibility with our experiment settings: most of them only work on a specific regularizer. For example, we can only report with a Shanonnian entropic regularizer for most of the algorithms. This is because applying regularizers other than the Shannon entropy function is not theoretically pleasant for these methods e.g., $\pi_E$ might not be the optimal point. Therefore, comparison with methods other than RAIRL is fundamentally limited for other approaches.

---

### Official Review · Reviewer_Ptuv · 2022-07-11

**Rating:** 6
**Confidence:** 3
**Soundness:** 3 good
**Presentation:** 3 good
**Contribution:** 3 good

**Summary:**

This paper cast the imitation learning problem as an iterative RL-IRL process and proposes a new adversarial imitation learning (AIL) algorithm based on mirror descent. The authors prove the convergence of the learning policy to an optimal policy in $\mathcal{O}(\frac{1}{T})$. The experiments show that the proposed method, MD-AIRL, consistently outperforms RAIRL.

**Questions:**


(1) The current theoretic analyses show the convergence to the optimal policy, which seems to come from the nature of MD. Is it possible to establish some results on the learned reward function? Or is it possible to give some analysis (experimental or theoretic) on the learned reward function?

(2) I noticed the BC results in the MuJoCo tasks are very low, while BC can be a very strong baseline on MuJoCo tasks with only 1 demonstration (please see Figure 4 in [1]). Can you clarify the implementation details of BC?


**Minor:**
In Thm. 2 (Line 220), what is the difference between $A_t$ and $A_T$?


[1] Ziniu, et al., "Rethinking ValueDice - Does It Really Improve Performance?", ICLR Blog Track, 2022.

**Limitations:**

The authors proposed an appealing approach to make the AIL process more robust. The theoretical result on the convergence rate is of great importance and the experiments show MD-AIRL is consistently better than RAIRL. MD-AIRL learns the reward function, which is also important for the IRL problem, while the authors dig little into the learned reward. Besides, comparisons with other recent IL and IRL algorithms will be helpful.

**Strengths And Weaknesses:**

**Strengths**:

Motivated by some practical observations, e.g., the global solution in AIL is hard to obtain by a monolithic estimation process, the authors convert the original AIL formulation to a sequence of policies and associated reward functions, and derive an MD update rule that enjoys a regret bound of $\mathcal{O}(\frac{1}{T})$. This approach is appealing and seems sound, which also provides a linear convergence rate to the optimal policy. The experiments demonstrate MD-AIRL consistently outperforms RAIRL and is more robust to the noise and the number of trajectories.

Overall, this paper is clearly written and easy to follow.

**Weaknesses**:

While the analysis part contributes a lot, the compared baseline only consists of RAIRL and BC. Some recent work, e.g., f-IRL (in terms of IRL) [1] and ValueDice (in terms of imitation learning) [2] can also achieve a more robust learning process than AIL and AIRL, which are missing in the literature or as baselines. Besides, MD-AIRL extends the AIRL framework, thus the learned reward function may be used in the downstream task as done in f-IRL. However, the functionality of the learned reward function has not been revealed.

[1] Tianwei Ni, Harshit S. Sikchi, Yufei Wang, et al. "f-IRL: Inverse Reinforcement Learning via State Marginal Matching." CoRL 2020.
[2] Kostrikov, Ilya, Ofir Nachum, and Jonathan Tompson. "Imitation learning via off-policy distribution matching." ICLR 2020.

---

> ### Author Response · Authors · 2022-08-02
> **Official Response to Reviewer Ptuv (1/2)**
>
> Thank you very much for your insightful feedback and the time spent reviewing our work. We address your comments below.
>
> **Comparison studies.**
>
> Here we note the important reasons this work focused on comparing with RAIRL in this paper.
>
> 1. The reasons for fixing RAIRL as the main comparative algorithm throughout this work are (1) compatibility and (2) significance. A typical IRL algorithm is not designed to work on multiple statistical divergences. To the best of our knowledge, only RAIRL is a suitable algorithm that models the Bregman divergence so that the algorithm can be applied to all sets of our comparative experiments. Second, since RAIRL shares similar reward models and learning schemes with MD-AIRL, consistently showing performance gains from RAIRL was one of the core properties of MD-AIRL that is necessary to validate our claims.
> 2. **F-IRL** [1] works on state densities with asymptotic estimation typically trained with mini-batches. In contrast, MD-AIRL contains the exact form of MD for probabilistic actions, incorporated with an estimated state density discriminator. F-IRL might be a reasonable algorithm if the problem is suitable for a specific divergence. Validating an IRL method that models an f-divergence is beyond the scope of our paper, as addressed in Section 8.
> 3. **ValueDice** [2] focuses on improving offline settings of imitation learning algorithms, such as behavior cloning (BC). The authors assumed (1) environments to be deterministic MDPs, and also (2) parameterization to be deterministic, which is quite different from our settings. These combined factors make a fair comparison of the ValueDice method with the MD-AIRL approach very difficult in this paper. We respectfully point out that our goal for experiments was not state-of-the-art performance, even for these vastly different cases.
>
> **Functionality of the learned reward function.**
>
> We address detailed analyses below.
>
> 1. We presented the strict parameterization for the reward representation that was theoretically pleasing for both MD and regularized that is not standard dual gradients. Compared to classical MD, this work is dealing sequential decision problems of regularized MDPs with unknown dynamics. Therefore, our theories are geared toward rationalizing the regularizer $\Psi$ and the convergence of the cumulative costs.
> 2. The notion of the ground-truth reward requires elaboration since IRL is an ill-posed problem; there can be numerous solutions to the reward function inducing the same optimal policy. Our approach is grounded on a novel perspective of simultaneous learning policies and rewards in RL/IRL methods corresponding to the combined, multi-stage optimization process. Therefore, a particular reward function is tightly coupled with the corresponding policy (regardless of optimality). The proposed learning objective provides the reward function (for?) next iterative steps, constrained to local geometric constraints of the convex regularizer.
>
> We also conducted the multi-goal experiment in Fig. 6, visualizing the basic characteristics of the learned IRL reward function.
>
> **Downstream tasks.**
>
> 1. **Transfer learning.** We would like to point out that the primary purpose of this work is the robustness of imitation learning. While it can reproduce similar results since it retains the functionality of AIRL variants, we believe that a more critical aspect of the practicality of our imitation learning algorithms is preserving the algorithm's robustness even for challenging situations.
> 2. **Reusability of reward functions.** Our key postulation of reward function is rooted in the core argument that a reward function has a one-to-one relationship with a corresponding policy ($\Omega$ determines such correspondence). Based on our theoretical analyses, this claim is a strong argument that is different from other works that directly estimate a reward function of a certain form. From this perspective, we do not see a fixed form of reward function as a reusable object across multiple downstream tasks due to the nature of accumulation of errors in imitation learning in sequential decision problems and the possibility of negative transfer. Instead, we consider a reward function acquired by MD as an instant representation of current policy in a different form.

---

> > ### Author Response · Authors · 2022-08-02
> > **Official Response to Reviewer Ptuv (2/2)**
> >
> > **Implementation details of BC.**
> >
> > Thank you for the question. For this issue, we would like to point out a quote from the appendix of the paper [2] “Rethinking ValueDice - Does It Really improve Performance?” ([https://arxiv.org/pdf/2202.02468.pdf](https://arxiv.org/pdf/2202.02468.pdf), pp. 15)
> >
> > > ***Algorithm Implementation.** Our implementation of ValueDice and DAC follows the public repository https://github.com/google-research/google-research/tree/master/value_dice by Kostrikov et al. [2020]. **Our implementation of BC is different from the one in this repository.** … Instead, we use a simple MLP architecture without the output of the covariance.  **The deterministic policy is trained with mean-squared-error (MSE).***
> >
> > The appendix implies that the authors’ implementation of BC in [2] is different from standard ones for stochastic policies. Our implementation of BC is based on the `OpenAI Baselines` repository, where we applied additional changes to the original code for incorporating the full-covariance Gaussian parameterization and for fixing overall minor compatibility issues with MuJoCo 2.1 and TensorFlow 2.0. Under the close inspection of the paper and code, the main reasons for the performance boost compared to our implementation appear to be as follows:
> >
> > 1. **Parameterizations are different.** A deterministic policy is inherently a strict outlier of our arguments and analyses since $\Omega(\pi)$ is usually undefined. We stress that the scope of this work is configured to dealing the problem in the sequential decision problem in the regularized MDPs of stochastic policies. Studying stochastic policies, in general, has distinct advantages of considering all the possible pathways to reaching multiple goals and the overall stability of small perturbations of environmental configurations.
> > 2. **Loss functions are different.** Notice that we assumed Gaussians for MuJoCo benchmarks, so the loss function of maximum likelihood estimation can be widely different from MSE depending on covariance matrices.
> > 3. **Regularization for weights.** The authors additionally applied $\ell_2$ regularization to standard BC, and we can observe that unregularized BC performs poorly, in Fig. 4 of [2]. In fact, this point is connected to our argument of online updates in Section 2 and step size considerations in our settings that the parameters are treated as a point in constrained convex optimization in the Euclidean space. As mentioned in Section 8, the best (stochastic) regularization for IL/IRL for each specific task is another challenging problem in this domain and remains as future work.
> >
> > **Theorem 2.**
> >
> > The aim of the theoretical argument in Theorem 2 is to show the boundedness of cost that corresponds to conditional Bregman divergences regarding all states $s\in \mathcal{S}$. Therefore, we define $A_t = \sup_{s\in \mathcal{S}} \mathbb{E}_{\tau 1:t } [ D_\Omega(\pi^s_E \Vert \pi_t^s ) ] $ which represents such cost in one term. $T$ represents the finite end of training time, indicating finite training time. Thank you for this question; we enhanced the presentation of Theorem 2 in the rebuttal period, which can be checked in the rebuttal revision.
> >
> > [1] Tianwei Ni, Harshit S. Sikchi, Yufei Wang, Tejus Gupta, Lisa Lee, and Ben Eysenbach. F-IRL: inverse reinforcement learning via state marginal matching. In Conference on Robot Learning, pages 529–551. PMLR, 2020.
> >
> > [2] Ziniu, et al., Rethinking ValueDice - Does It Really improve Performance?, ICLR Blog Track 2022.
> >
> > [3] Ng, A. Y., Harada, D., and Russell, S., Policy invariance under reward transformations: Theory and application to reward shaping. In ICML 1999, Vol. 99, pp. 278-287.

---

> > ### Comment · Reviewer_Ptuv · 2022-08-04
> > **Response to Authors**
> >
> > Thanks very much for the general & personal responses and efforts! These clarifications make it more clear for me to understand the purpose and postulation of this work. The detailed explanations on the functionality of the learned reward function are important.
> >
> > As for the comparisons, I wonder whether the current evaluations on MuJoCo are under the stochastic policies or the deterministic policies (the mean of the Gaussian); if they are stochastic, how about the scores of the deterministic ones? From my experiences, the deterministic policies often achieve higher scores on MuJoCo even though they are parameterized by a Gaussian during the learning.
> >
> > Besides, I am not fully convinced by the authors' implementation of BC: if we can observe that adding a simple $\ell_2$ will help significantly improve the performance, why cannot we adopt this technique? So I suggest the authors add a $\ell_2$ regularized BC as the BC baseline.
> >
> > Minor: BTW, I noticed an extra type in Line 16 ("often described 16 as learning expert behavior fro finite number of demonstrations" [sig]). fro -> for

---

> > > ### Author Response · Authors · 2022-08-06
> > > **Response to Reviewer Ptuv (Discussion, 1/2)**
> > >
> > > We highly appreciate the insightful comments and your dedication to the discussion. We address the comments below.
> > >
> > > **About using deterministic parameterizations for evaluation.**
> > >
> > > Thank you for the insightful and instructive comments. We clarify that this paper assumed stochastic policies evaluating our experiments for the sake of consistency. As suggested, we report the evaluation scores on the deterministic setting below.
> > >
> > > **Table B3.  Performance of MuJoCo with different policy types (100 expert demonstrations).**
> > >
> > > |  | Sto. Policy ($a \sim \pi(a\vert s)$) | Det. Policy ($a = \mu(s)$) | Difference (Det - Sto) |
> > > | --- | --- | --- | --- |
> > > | BC (Hopper-v3) | 0.76 | 0.88 | **0.12** |
> > > | BC (Walker2d-v3) | 0.39 | 0.48  | **0.09** |
> > > | RAIRL (Hopper-v3) | 0.95 | 0.96 | **0.01** |
> > > | RAIRL (Walker2d-v3) | 0.67 | 0.73 | **0.07** |
> > > | MD-AIRL (Hopper-v3) | 0.96 | 0.97 | **0.01** |
> > > | MD-AIRL (Walker2d-v3 | 0.89 | 0.91 | **0.02** |
> > >
> > > - We believe these results are coherent with your experience: in MuJoCo, we can expect some performance gains when a policy behaves deterministically regardless of algorithms. We think these gains are strongly related to the innate characteristics of MuJoCo tasks (combined with Gaussian policies) since similar phenomena have often been reported in previous imitation learning and offline RL papers.
> > > - It is a genuinely good practice to check scores on deterministic versions and to compare them with the expected expert's scores. These results would be helpful as validation metrics, especially when grasping how the modes of actions (i.e., mean actions) deviate from the expert policy.
> > > - That being said, it would also be hard to interpret and analyze these performance differences and consider them as actual performance benefits because inconsistency between training and evaluation settings is intentionally caused by a practitioner's experiment design. We think this criticism could be further validated when solving real-world problems (such as robot hand manipulation and autonomous driving).
> > > - We respectfully emphasize that we presented a parameterization-agnostic algorithm for stochastic policies. The Gaussian parameterization in this paper is much more expressive than diagonal Gaussians when it comes to modeling correlation among dimensions of multivariate actions. Therefore, it is likely that our experiments can successfully determine the performance of tested algorithms for probability distributions. We believe this paper's evaluation strategy is sound and appropriate.

---

> > > > ### Author Response · Authors · 2022-08-06
> > > > **Response to Reviewer Ptuv (Discussion, 2/2)**
> > > >
> > > > **Justifications for the BC implementation**.
> > > >
> > > > Thank you for the insightful comment. First, we will make sure to include ablation of applying $l2$ and other regularization techniques in the final version. At the same time, we provide justifications for our BC implementation as follows.
> > > >
> > > > - **Not Fine-tuned for deterministic tasks.**
> > > >     1. Ziniu, et al. [1] presented some remarkable analyses on the worst-case performance of offline imitation learning algorithms. Personally, their findings and analyses are intriguing as an imitation learning study.
> > > >     2. We respectfully highlight one of the paper's vital observations (https://arxiv.org/pdf/2202.02468.pdf, pp. 7).
> > > >
> > > >         > ***Observation 2.** For deterministic tasks (e.g., MuJoCo locomotion tasks), BC has no compounding errors if the provided expert trajectories are complete.*
> > > >
> > > >         This implies that their formalizations are designed in favor of deterministic tasks, especially MuJoCo benchmarks. We would like to point out that the deterministic BC implementation of [1] is more geared toward MuJoCo benchmarks than ours by utilizing additional information regarding environments.
> > > >
> > > >     3. More importantly, we respectfully highlight one of the drawbacks of deterministic benchmarks explicitly pointed out by the authors (pp. 9) that we fully agree on.
> > > >
> > > >         > **Benchmarks.** *Our study points out several drawbacks of existing MuJoCo locomotion benchmarks: deterministic transitions and limited initial states. … future imitation learning studies could benefit from more challenging benchmarks with stochastic transitions and diverse initial states***.**
> > > >         >
> > > >
> > > >         For standard stochastic MDPs, we believe that the actions of an agent have to be modeled with a stochastic policy. Otherwise, compounding errors will become one of the significant problems. Therefore, we claim that our implementation of BC is appropriate since our MDP definition and theoretical claims are not limited to deterministic settings.
> > > >
> > > >     4. Lastly, we respectfully remind you that this paper also consists of imitation learning in the multi-goal benchmark that heavily favors stochastic policies. Therefore, the general premise of stochastic policy was needed for consistency between continuous action space experiments.
> > > > - **About** $\ell_2$ **Regularization.**
> > > >     1. Following the analyses of Ziniu et al. [1], it is safe to say we can apply the proposed regularization when we are informed that the given MDP is deterministic.
> > > >     2. Applying the $\ell_2$ regularization is closely related to enforcing Lipschitz continuity on the network outputs [2,3]. Specifically, we think this regularization is certainly beneficial to simple deterministic networks in MuJoCo benchmarks because of a few reasons.
> > > >         - Deterministic policy networks process single, direct output of actions
> > > >         - In MuJoCo locomotion tasks, the outputs are real vectors and each action dimension has a relevant meaning for a physical system, such as the magnitude of forces.
> > > >     3. From a technical perspective, a stochastic policy model such as a Gaussian network outputs multiple vectorized outputs, and some of outputs might not be adjusted to the regularization method. Therefore, we are not fully sure this specific regularization can guarantee performance gains for arbitrary stochastic policy parameterization. We believe the relationship between regularization and parameterization needs to be extensively studied.
> > > >     4. Again, Ziniu, et al. [1] left interesting remarks on imitation learning benchmarks and regularization. We respectfully point out that the contributions of our paper are complementary to the work done by Ziniu, et al. [1]. We believe our methodology of MD-IRL and its implementation can benefit from these results, or vice versa.
> > > >
> > > > - **Revision for the final submission.** In Appendix D, we will include ablation studies for regularization techniques. The experiments can be finished in a couple of weeks.
> > > >
> > > > **Minor (Line 16).** Thank you for your effort on this point. We corrected this typo and uploaded a revised version.
> > > >
> > > > Please let us know if there are any remaining questions!
> > > >
> > > > Best Regards,
> > > >
> > > > Authors of Submission #4477
> > > >
> > > > [1] Ziniu, et al., Rethinking ValueDice - Does It Really improve Performance?, ICLR Blog Track 2022.
> > > >
> > > > [2] Takeru Miyato, Toshiki Kataoka, Masanori Koyama, Yuichi Yoshida,
> > > > Spectral Normalization for Generative Adversarial Networks, ICLR 2018
> > > >
> > > > [3] Zhiming Zhou, Jiadong Liang, Yuxuan Song, Lantao Yu, Hongwei Wang, Weinan Zhang, Yong Yu, Zhihua Zhang, Lipschitz Generative Adversarial Nets, ICML 2019.

---

> > > > > ### Comment · Reviewer_Ptuv · 2022-08-07
> > > > > **Response to authors**
> > > > >
> > > > > Thanks again for the efforts. Although the current evaluation is under the stochastic policy setting, I think this work still has its merit.

---

> > > > > > ### Author Response · Authors · 2022-08-08
> > > > > > **Response to Reviewer Ptuv (after Discussion)**
> > > > > >
> > > > > > We thank you for the time spent reviewing the paper and for the discussion. We are glad that most of your concerns have been addressed. Again, we highly appreciate your suggestions for new experiments, as these provide further insights for our paper.
> > > > > >
> > > > > > Best, Authors.

---

### Author Response · Authors · 2022-08-02
**The general response to all reviewers**

We appreciate all the reviewers for their invaluable feedback. We have used them to improve our paper. We also thank encouraging comments from the reviewers, referring to our work as well-written (**`Ptuv`**  & **`4Ddj`**), solid (**`4Ddj`)**, sound (**`Ptuv`**  & **`4Ddj`**), and novel (**`mFTz`)**.  Here we present abridged answers to the essential questions.

- **Limitations.** The design choice of dual discriminative architecture was from the theoretical requirements of Eq. (7) for all possible instances of the on-policy state samples of the environment. Matching state densities with the reward formulation in Eq. (15) has distinct technical benefits in practice. Following the reviewers’ suggestions, we rewrote **Section 8** in the revised version of our manuscript.
- **Experiments.** The main reasons for focusing the comparison studies with RAIRL were (1) compatibility and (2) significance. RAIRL is the essential counterpart for almost all experiments because it inherits the general feature of the modern AIL method, with a wide range of choices of Bregman divergence. Therefore, the consistent performance gains of MD-AIRL from RAIRL are important as they are directly related to the core theoretical claims. As suggested, we clarified the goal of the experiments and the underlying reasoning in the rebuttal revision and included additional experimental results below.

**Additional experiments.** We are pleased to report additional empirical evidence before the author-reviewer discussion period. The following comparison experiment in Table B1 covers several recent IRL methods, where the reviewers suggest many of the algorithms. In particular, GAIL [1] with DAC-style reward function ($r(s,a) := \log D(s, a) - \log(1 - D(s, a))$) [2], FAIRL [3], and F-IRL[4] were additionally tested with noisy demonstrations. The table shows the performance for five different algorithms; we used our RAC implementation with Shannon regularizer except for F-IRL (we tested F-IRL with the official implementation with custom trajectory data).

**Table B1. Imitation learning scores for noisy demonstrations ($\varepsilon = 0.5$, the scores are rescaled by considering the average expert performance as 1).**

|  | Hopper-v3 | Walker2d-v3 | HalfCheetah-v3 | Ant-v3 |
| --- | --- | --- | --- | --- |
| GAIL (DAC-reward) [1,2] | 0.95 ± 0.11 | 0.66 ± 0.24 | 0.97 ± 0.07 | 0.94 ± 0.08 |
| FAIRL [2] | 0.73 ± 0.25 | 0.14 ± 0.06 | 0.96 ± 0.03 | -0.11 ± 0.15 |
| F-IRL [3] | 0.82 ± 0.06 | 0.67 ± 0.2 | 0.94 ± 0.03 | 0.91 ± 0.07 |
| RAIRL | 0.96 ± 0.38 | 0.55 ± 0.32 | 0.96 ± 0.15 | 0.85 ± 0.11 |
| MD-AIRL | 0.98 ± 0.07 | 0.73 ± 0.31 | 0.98 ± 0.02 | 0.96 ± 0.07 |

MD-AIRL clearly outperforms modern IRL algorithms when it comes to the robustness of unreliable demonstrations. These additional results is in alignment with our analyses and supports our claims.

**Rebuttal Revision.** Before the discussion, we posted a current revision of our manuscript. Notably, this version contains new lines in Sections 6 & 8 to reflect the majority of reviewers' important suggestions, as well as a few minor corrections of typos and grammatical errors. We believe the presentation has become much clear. We are currently extending this process further, incorporating all of the reviewers' invaluable comments for the final submission.

Please let us know if there are any remaining questions!

Best Regards,

Authors of Submission #4477

[1] Jonathan Ho and Stefano Ermon. Generative adversarial imitation learning. In Advances in Neural Information Processing Systems, pages 4565–4573, 2016.

[2] Ilya Kostrikov, Kumar Krishna Agrawal, Debidatta Dwibedi, Sergey Levine, Jonathan Tompson, Discriminator-Actor-Critic: Addressing Sample Inefficiency and Reward Bias in Adversarial Imitation Learning, In ICLR 2019

[3] Seyed Kamyar Seyed Ghasemipour, Richard Zemel, and Shixiang Gu. A divergence minimization perspective on imitation learning methods. In Conference on Robot Learning, pages 1259–1277. PMLR, 2020.

[4] Tianwei Ni, Harshit S. Sikchi, Yufei Wang, Tejus Gupta, Lisa Lee, and Ben Eysenbach. F-IRL: inverse reinforcement learning via state marginal matching. In Conference on Robot Learning, pages 529–551. PMLR, 2020.

---

### Meta-Review · Area_Chair_1Hmz · 2022-08-26

**Recommendation:** Accept
**Confidence:** Certain

**Metareview:**

This work proposes imitation learning via the route of mirror descent inverse-rl. Mirror descent is a well-understood optimization algorithm and framing IRL via it is a good theoretical exercise. Using an expert to help schedule learning is a novel theoretical contribution in the context of adversarial imitation learning. This is directly guiding how to design the approach.

The current concern is that the experimental results are not statistically significant and even though the nice theoretical properties of mirror descent are nice to potentially leverage, it is not coming through strongly yet.  One suggestion is to drastically increase the number of random seeds (say 25) and report 2*standard error instead of standard deviation, especially comparing to RAIRL. The promising innovation is the idea of multiple discriminators which can better account for distribution shift. The authors are encouraged to bolster the experiments with this in mind and frame this as the central point of the work with the theory of MD as supporting evidence.

The writing of the paper can also be improved and the idea of estimating experts for curriculum can be highlighted better as this is a significant contribution and is currently a bit buried in text. A significant refactoring of Section 4 and a running example that connects Fig 2 and 3 to the algorithm in Section 5 will greatly help. Lines 130-151 are not the main contribution and can be moved to appendix or cut.

This is also mainly an imitation learning paper and not an IRL paper as the reviewers have noted. While the naming follows the convention of other IL papers like GAIL, RAIRL, it can be a bit misleading. Perhaps the authors can reconsider the name.


**Award:**

No

---

### Decision · Program_Chairs · 2022-09-14

Accept